# Sketching for Convex and Nonconvex Regularized Least Squares with Sharp Guarantees

**Yingzhen Yang**
School of Computing and Augmented Intelligence
Arizona State University, Tempe, AZ 85281, USA
yingzhen.yang@asu.edu

**Ping Li**
VecML Inc., Bellevue, WA 98004, USA
pingli98@gmail.com

## Abstract

Randomized algorithms play a crucial role in efficiently solving large-scale optimization problems. In this paper, we introduce Sketching for Regularized Optimization (SRO), a fast sketching algorithm designed for least squares problems with convex or nonconvex regularization. SRO operates by first creating a sketch of the original data matrix and then solving the sketched problem. We establish minimax optimal rates for sparse signal estimation by addressing the sketched sparse convex and nonconvex learning problems. Furthermore, we propose a novel Iterative SRO algorithm, which reduces the approximation error geometrically for sketched convex regularized problems. To the best of our knowledge, this work is among the first to provide a unified theoretical framework demonstrating minimax rates for convex and nonconvex sparse learning problems via sketching. Experimental results validate the efficiency and effectiveness of both the SRO and Iterative SRO algorithms.

## 1 Introduction

Randomized algorithms for efficient optimization are a critical area of research in machine learning and optimization, with wide-ranging applications in numerical linear algebra, data analysis, and scientific computing. Among these, matrix sketching and random projection techniques have gained significant attention for solving sketched problems at a much smaller scale (Vempala, 2004; Boutsidis & Drineas, 2009; Drineas et al., 2011; Mahoney, 2011; Kane & Nelson, 2014). These methods have been successfully applied to large-scale problems such as least squares regression, robust regression, low-rank approximation, singular value decomposition, and matrix factorization (Halko et al., 2011; Lu et al., 2013; Alaoui & Mahoney, 2015; Raskutti & Mahoney, 2016; Yang et al., 2015; Drineas & Mahoney, 2016; Oymak et al., 2018; Oymak & Tropp, 2017; Tropp et al., 2017). Regularized optimization problems with convex or nonconvex regularization, such as the widely used in regularized least squares such as Lasso and ridge regression, play a fundamental role in machine learning and statistics. While prior research has extensively explored random projection and sketching methods for problems with standard convex regularization (Zhang et al., 2016b) or convex constraints (Pilanci & Wainwright, 2016), there has been limited focus on analyzing regularized problems with general convex or nonconvex regularization frameworks.

We would like to emphasize that while (Yang & Li, 2021) also studies sketching for regularized optimization problem, the focus and results of this work are completely different from that in (Yang & Li, 2021), with a detailed discussion deferred to Section 5. In particular, due to our novel result in the approximation error bound (Theorem D.2-Theorem D.3), the proposed iterative sketching algorithm, Iterative SRO, does not need to sample a new projection matrix and compute the sketched matrix at every iteration, in a strong contrast to (Yang & Li, 2021). Moreover, the focus of this work is to establish minimax optimal rates for sparse convex and nonconvex learning problems by sketching, which has not been addressed by existing works in the literature including (Yang & Li, 2021). While (Yang & Li, 2021) only focuses on the optimization perspective, that is, approximating the solution to the original optimization problem by the solution to the sketched problem, the focus of this work needs much more efforts beyond the efforts made in (Yang & Li, 2021) for optimization only: we need to show that the solution to the sketched problem can still enjoy the minimax optimal

rates for estimation of the sparse parameter vector for both sparse convex and nonconvex learning problems. Such efforts and results in minimax optimal rates for sparse convex and nonconvex learning problems by sketching have not been offered by previous works including (Yang & Li, 2021), which are provided in Section 4 of this paper. Such minimax optimal results are established in a highly non-trivial manner. For example, to the best of our knowledge, Theorem 4.1 is among the first in the literature which uses an iteratively sketching algorithm to achieve the minimax optimal rate for sparse convex learning. Furthermore, Theorem 4.5 shows that sketching can also lead to the minimax optimal rate even for sparse nonconvex problems, while sketching for nonconvex problems is still considered difficult and open in the literature.

In this paper, we study efficient sketching algorithm for the optimization problem of regularized least squares with convex or nonconvex regularization, which is presented as follows:

$$\min_{\boldsymbol{\beta} \in \mathbb{R}^d} f(\boldsymbol{\beta}) = \frac{1}{2} \|\mathbf{y} - \mathbf{X}\boldsymbol{\beta}\|_2^2 + h_\lambda(\boldsymbol{\beta}). \tag{1}$$

Here $\mathbf{X} \in \mathbb{R}^{n \times d}$ is the data matrix or design matrix for regression problems, $h_\lambda : \mathbb{R}^d \to \mathbb{R}$ is a regularizer function and $\lambda$ is a positive regularization weight. When $h_\lambda(\cdot) = \lambda \|\cdot\|_2^2$ or $h_\lambda(\cdot) = \lambda \|\cdot\|_1$, (1) is the optimization problem for ridge regression or $\ell^1$ regularized least square estimation (Lasso). Section 4.2 provides more examples for nonconvex $h_\lambda$. We focus on sparsity-inducing regularizers, and the solution to the sketched problem (1) can provably approximate the true parameter vector for sparse signal estimation to be detailed in Section 4.

We study the regime that $n \gg r = \text{rank}(\mathbf{X})$ where $r$ is the rank of $\mathbf{X}$ in most results of this paper, and it is a popular setting for large-scale problems such as fast least square estimation by sketching (Drineas et al., 2011). For example, matrix $\mathbf{X}$ in sparse linear regression is usually low-rank or approximately low-rank in practice. However, our results for sparse nonconvex learning in Subsection 4.2 hold for $\mathbf{X}$ not necessarily low-rank.

Optimization for (1) is time consuming when $n$ is large, and such large-scale regularized optimization problems are important due to the increasing interest in massive data. To this end, we propose Sketching for Regularized Optimization (SRO) in this paper as an efficient randomized algorithm for problem (1). With $\tilde{n} < n$ where $\tilde{n}$ is the target row number of a sketch of the data matrix $\mathbf{X}$ which is also termed the sketch size, SRO first generates a sketched version of $\mathbf{X}$ by $\tilde{\mathbf{X}} = \mathbf{PX}$, then solves the following sketched problem,

$$\min_{\boldsymbol{\beta} \in \mathbb{R}^d} \tilde{f}(\boldsymbol{\beta}) = \frac{1}{2} \boldsymbol{\beta}^\top \tilde{\mathbf{X}}^\top \tilde{\mathbf{X}} \boldsymbol{\beta} - \langle \mathbf{y}, \mathbf{X}\boldsymbol{\beta} \rangle + h_\lambda(\boldsymbol{\beta}). \tag{2}$$

One hopes that the optimization result of the sketched problem (2), denoted by $\tilde{\boldsymbol{\beta}}^*$, is a good approximation to that of the original problem (1), denoted by $\boldsymbol{\beta}^*$. The optimization and theoretical computer science literature are particulary interested in the solution approximation measure defined as the semi-norm induced by the data matrix $\mathbf{X}$, that is, $\left\|\tilde{\boldsymbol{\beta}}^* - \boldsymbol{\beta}^*\right\|_{\mathbf{X}} = \left\|\mathbf{X}(\tilde{\boldsymbol{\beta}}^* - \boldsymbol{\beta}^*)\right\|_2$ where $\|\mathbf{u}\|_{\mathbf{X}} := \|\mathbf{Xu}\|_2$ for any vector $\mathbf{u}$. Existing research, such as Iterative Hessian Sketch (IHS) (Pilanci & Wainwright, 2016), prefers relative-error approximation to the solution of the original problem in the following form:

$$\left\|\tilde{\boldsymbol{\beta}}^* - \boldsymbol{\beta}^*\right\|_{\mathbf{X}} \le \rho \|\boldsymbol{\beta}^*\|_{\mathbf{X}}, \tag{3}$$

where $0 < \rho < 1$ is a positive constant. With the relative-error approximation (3), IHS proposes an interesting iterative sketching method to reduce the approximation error $\left\|\tilde{\boldsymbol{\beta}}^* - \boldsymbol{\beta}^*\right\|_{\mathbf{X}}$ geometrically in the iteration number. Sketching the matrix $\mathbf{X}$ only in the quadratic term $\|\mathbf{X}\boldsymbol{\beta}\|_2^2$ is proposed in (Pilanci & Wainwright, 2016) for constrained least square problems with convex constraints. SRO adopts this idea for regularized least square problems admitting a broad range of regularizers.

## 1.1 CONTRIBUTIONS AND MAIN RESULTS

We study the sparse signal estimation problem by sketching in Section 4. For sparse convex or nonconvex learning problems where $h_\lambda$ is convex or nonconvex, we obtain the minimax optimal

rate of the order $\mathcal{O}\left(\sqrt{\bar{s}\log d/n}\right)$ for the parameter estimation error in sparse signal estimation by solving the sketched problem (2) where $\bar{s}$ is the support size of the unknown sparse parameter vector to be estimated. To the best of our knowledge, our analysis provides the first unified theoretical result for the minimax optimal error rate for sparse signal estimation using sketching based optimization method.

In order to obtain such minimax optimal rate for sparse signal estimation by solving the sketched sparse convex learning problems, we propose an iterative sketching algorithm termed Iterative SRO in Section 3, which provably reduces the approximation error $\left\|\tilde{\boldsymbol{\beta}}^* - \boldsymbol{\beta}^*\right\|_{\mathbf{X}}$ geometrically in an iterative manner. Albeit being bounded, the approximation error of one-time SRO is still not small enough for various applications. To this end, Iterative SRO iteratively applies SRO to approximate the residual of the last iteration so as to further reduce the approximation error, given the relative-error approximation (3). More details are introduced in Section 3 and Section 7.

There are two key differences between Iterative SRO and IHS (Pilanci & Wainwright, 2016). First, using the subspace embedding as the projection matrix $\mathbf{P}$, Iterative SRO does not need to sample $\mathbf{P}$ and compute the sketched matrix $\tilde{\mathbf{X}} = \mathbf{PX}$ at each iteration, in contrast with IHS where a separate $\mathbf{P}$ is sampled and $\tilde{\mathbf{X}}$ is computed at each iteration. This advantage saves considerable computation and storage for large-scale problems. Second, while IHS is restricted to constrained least-square problems with convex constraints, SRO and Iterative SRO are capable of handling all convex regularization and certain nonconvex regularization in a unified framework. For example, we show that Generalized Lasso (Tibshirani & Taylor, 2011) can be efficiently and effectively solved by Iterative SRO in Section 7.

## 1.2 NOTATIONS

Throughout this paper, we use bold letters for matrices and vectors, regular lower letters for scalars. The bold letter with subscript indicates the corresponding element of a matrix or vector, and the bold letter with superscript indicates the corresponding column of a matrix, i.e. $\mathbf{X}^i$ indicates the $i$-th column of matrix $\mathbf{X}$. $\|\cdot\|_p$ denotes the $\ell^p$-norm of a vector, or the $p$-norm of a matrix. $\sigma_t(\cdot)$ is the $t$-th largest singular value of a matrix, and $\sigma_{\min}(\cdot)$ and $\sigma_{\max}(\cdot)$ indicate the smallest and largest singular value of a matrix respectively. $\mathrm{tr}(\cdot)$ is the trace of a matrix. $f_1(n) = \Theta(f_2(n))$ if there exist constants $k_1, k_2 > 0$ and $n_0$ such that $k_1 f(n) \leq f_1(n) \leq k_2 f_2(n)$. We use $\mathbf{X} \succcurlyeq \mathbf{Y}$ to indicate that $\mathbf{X} - \mathbf{Y}$ is a positive semi-definite matrix, and $\mathbf{I}_d$ indicates the $d \times d$ identity matrix. $\mathrm{rank}(\mathbf{X})$ means the rank of a matrix $\mathbf{X}$. $\mathbb{N}$ denotes the set of all the natural numbers, and we use $[m \ldots n]$ to indicate numbers between $m$ and $n$ inclusively, and $[n]$ denotes the natural numbers between 1 and $n$ inclusively. $\mathsf{nnz}(\mathbf{X})$ indicates the number of nonzero elements of a matrix $\mathbf{X}$. $|\cdot|$ denotes the cardinality of a set, and $\mathrm{supp}(\cdot)$ denotes the set of indices of nonzero elements for a vector.

## 2 THE SRO ALGORITHM

In order to improve the efficiency of optimization for (1), we propose Regularized Optimization by Sketching (SRO) in this section. The key idea is to sketch matrix $\mathbf{X}$ in the quadratic term of (1) by random projection. It consisits of two steps:

Step 1. Project the matrix $\mathbf{X}$ onto a lower dimensional space by a linear transformation $\mathbf{P} \in \mathbb{R}^{\tilde{n} \times n}$ with $\tilde{n} < n$, i.e. $\tilde{\mathbf{X}} = \mathbf{PX}$. $\tilde{n}$ is named the sketch size.

Step 2. Solve the sketched problem (2).

The linear transformation $\mathbf{P}$ is required to be a subspace embedding (Woodruff, 2014) defined in Definition 2.1. The literature (Frankl & Maehara, 1987; Indyk & Motwani, 1998; Zhang et al., 2016a) extensively studies such random transformation which is also closely related to the proof of the Johnson-Lindenstrauss lemma (Dasgupta & Gupta, 2003).

**Definition 2.1.** Suppose $\mathcal{P}$ is a distribution over $\tilde{n} \times n$ matrices, where $\tilde{n}$ is a function of $n$, $d$, $\varepsilon$, and $\delta$. Suppose that with probability at least $1 - \delta$, for any fixed $n \times d$ matrix $\mathbf{X}$, a matrix $\mathbf{P}$ drawn from distribution $\mathcal{P}$ has the property that $\mathbf{P}$ is a $(1 \pm \varepsilon)$ $\ell^2$-subspace embedding for $\mathbf{X}$, that is,

$$(1 - \varepsilon)\|\mathbf{X}\boldsymbol{\beta}\|_2^2 \leq \|\mathbf{PX}\boldsymbol{\beta}\|_2^2 \leq (1 + \varepsilon)\|\mathbf{X}\boldsymbol{\beta}\|_2^2 \tag{4}$$

holds for all $\boldsymbol{\beta} \in \mathbb{R}^d$. Then we call $\mathcal{P}$ an $(\varepsilon, \delta)$ oblivious $\ell^2$-subspace embedding.

**Definition 2.2** (Gaussian Subspace Embedding, (Woodruff, 2014, Theorem 2.3))**.** Let $0 < \varepsilon, \delta < 1$, $\mathbf{P} = \frac{\mathbf{P}'}{\sqrt{\tilde{n}}}$ where $\mathbf{P}' \in \mathbb{R}^{\tilde{n} \times n}$ is a matrix whose elements are i.i.d. samples from the standard Gaussian distribution $\mathcal{N}(0, 1)$. Then if $\tilde{n} = \mathcal{O}((r + \log \frac{1}{\delta})\varepsilon^{-2})$, for any matrix $\mathbf{X} \in \mathbb{R}^{n \times d}$ with $r = \mathrm{rank}(\mathbf{X})$, with probability $1 - \delta$, $\mathbf{P} = \frac{\mathbf{P}'}{\sqrt{\tilde{n}}}$ is a $(1 \pm \varepsilon)$ $\ell^2$-subspace embedding for $\mathbf{X}$. $\mathbf{P}$ is named a Gaussian subspace embedding.

**Definition 2.3** (Sparse Subspace Embedding)**.** Let $\mathbf{P} \in \mathbb{R}^{\tilde{n} \times n}$. For each $i \in [n]$, $h(i) \in [\tilde{n}]$ is uniformly chosen from $[\tilde{n}]$, and $\sigma(i)$ is a uniformly random element of $\{1, -1\}$. We then set $\mathbf{P}_{h(i)i} = \sigma(i)$ and set $\mathbf{P}_{ji} = 0$ for all $j \neq i$. As a result, $\mathbf{P}$ has only a single nonzero element per column, and it is called a sparse subspace embedding.

Lemma 2.1 below, also presented in (Clarkson & Woodruff, 2013), shows that the sparse subspace embedding defined above is indeed a subspace embedding with a high probability.

**Lemma 2.1.** (Clarkson & Woodruff, 2013, Theorem 2.1). Let $\mathbf{P} \in \mathbb{R}^{\tilde{n} \times n}$ be a sparse embedding matrix with $\tilde{n} = \mathcal{O}(r^2/(\delta\varepsilon^2))$ rows. Then for any fixed $n \times d$ matrix $\mathbf{X}$ with $r = \mathrm{rank}(\mathbf{X})$, with probability $1 - \delta$, $\mathbf{P}$ is a $(1 \pm \varepsilon)$ $\ell^2$-subspace embedding for $\mathbf{X}$. Furthermore, $\mathbf{PX}$ can be computed in $\mathcal{O}(\mathrm{nnz}(\mathbf{X}))$ time, where $\mathrm{nnz}(\mathbf{X})$ is the number of nonzero elements of $\mathbf{X}$.

### 2.1 ERROR BOUNDS

The solutions to the original problem (1) and the sketched problem (2) by typical iterative optimization algorithms, such as gradient descent for smooth $h$ or proximal gradient method for non-smooth $h$, are always critical points of the corresponding objective functions under mild conditions (Bolte et al., 2014). Therefore, the analysis in the gap between $\tilde{\boldsymbol{\beta}}^*$ and $\boldsymbol{\beta}^*$ amounts to the analysis in the distance between critical points of the objective functions of (2) and that of (1), which is presented in Section 5. In the sequel, $\tilde{\boldsymbol{\beta}}^*$ is a critical point of the objective function (2) and $\boldsymbol{\beta}^*$ is a critical point of the objective function (1), if no confusion arises. More details about optimization algorithms are deferred to appendix.

## 3 ITERATIVE SRO

---

**Algorithm 1** Iterative SRO

---

Input: Initialize $\boldsymbol{\beta}^{(0)} = \mathbf{0}$, iteration number $N > 0$, $t = 0$.
**for** $t \leftarrow 1$ to $N$
Set

$$\boldsymbol{\beta}^{(t)} = \arg\min_{\boldsymbol{\beta} \in \mathbb{R}^d} \frac{1}{2} \left\| \tilde{\mathbf{X}} \left( \boldsymbol{\beta} - \boldsymbol{\beta}^{(t-1)} \right) \right\|_2^2 - \left\langle \mathbf{y} - \mathbf{X}\boldsymbol{\beta}^{(t-1)}, \mathbf{X}\boldsymbol{\beta} \right\rangle + h_\lambda(\boldsymbol{\beta}) \tag{5}$$

**end for**
Return $\boldsymbol{\beta}^{(N)}$

---

Inspired by Iterative Hessian Sketch (Pilanci & Wainwright, 2016), we introduce an iterative sketching method for SRO, termed Iterative SRO, so that the gap between solutions to the original problem and the sketched problem can be geometrically reduced compared to the one-time SRO. Iterative SRO will be used to solve the sketched Lasso problem with $h_\lambda(\boldsymbol{\beta}) = \lambda\|\boldsymbol{\beta}\|_1$ in the sketched problem (2), and obtain the minimax rate of the order $\mathcal{O}\left(\sqrt{s \log d/n}\right)$ for the parameter estimation error in sparse signal estimation to be detailed in Section 4.

The key idea is to iteratively apply SRO to generate a sequence $\{\boldsymbol{\beta}^{(t)}\}_{t=1}^N$ such that $\boldsymbol{\beta}^{(t)}$ is a more accurate approximation to $\boldsymbol{\beta}^*$, the solution to the original problem (1), than $\boldsymbol{\beta}^{(t-1)}$. Consider the optimization problem

$$\min_{\boldsymbol{\beta} \in \mathbb{R}^d} \frac{1}{2} \left\| \mathbf{X}(\boldsymbol{\beta} + \boldsymbol{\beta}^{(t-1)}) \right\|_2^2 - \mathbf{y}^\top \mathbf{X}\boldsymbol{\beta} + h_\lambda(\boldsymbol{\beta} + \boldsymbol{\beta}^{(t-1)}), \tag{6}$$

then $\boldsymbol{\beta}^* - \boldsymbol{\beta}^{(t-1)}$ is an optimal solution to (6). We apply SRO to problem (6) and suppose $\widehat{\boldsymbol{\beta}}$ is an solution to the sketched problem, i.e.

$$\widehat{\boldsymbol{\beta}} = \arg\min_{\boldsymbol{\beta} \in \mathbb{R}^d} \frac{1}{2} \left\| \tilde{\mathbf{X}} \boldsymbol{\beta} \right\|_2^2 - \left\langle \mathbf{y} - \mathbf{X}\boldsymbol{\beta}^{(t-1)}, \mathbf{X}\boldsymbol{\beta} \right\rangle + h_\lambda(\boldsymbol{\beta} + \boldsymbol{\beta}^{(t-1)}). \tag{7}$$

It is noted that $\boldsymbol{\beta}^{(t-1)}$ is moved from the quadratic term in (6) to the linear term in (7). $\widehat{\boldsymbol{\beta}}$ is supposed to be an approximation to $\boldsymbol{\beta}^* - \boldsymbol{\beta}^{(t-1)}$. If $\widehat{\boldsymbol{\beta}}$ admits the relative-error approximation bound (3), then $\boldsymbol{\beta}^{(t)} = \widehat{\boldsymbol{\beta}} + \boldsymbol{\beta}^{(t-1)}$ becomes a more accurate approximation to $\boldsymbol{\beta}^*$ than $\boldsymbol{\beta}^{(t-1)}$ by a factor of $\rho$. This can be verified by noting that $\left\| \boldsymbol{\beta}^{(t)} - \boldsymbol{\beta}^* \right\|_{\mathbf{X}} = \left\| \widehat{\boldsymbol{\beta}} - (\boldsymbol{\beta}^* - \boldsymbol{\beta}^{(t-1)}) \right\|_{\mathbf{X}} \le \rho \|\boldsymbol{\beta}^* - \boldsymbol{\beta}^{(t-1)}\|_{\mathbf{X}}$. By mathematical induction, we have Theorem 3.1 below showing that the approximation error of Iterative SRO, which is formally described by Algorithm 1, drops geometrically in the iteration number. It should be emphasized that Theorem 3.1 also handles certain nonconvex regularization.

**Theorem 3.1.** Suppose $\tilde{\boldsymbol{\beta}}^*$ is any critical point of the objective function in (2), and $\boldsymbol{\beta}^*$ is any critical point of the objective function in (1). Suppose $0 < \varepsilon < \varepsilon_0 < 1$ where $\varepsilon_0$ is a small positive constant, $0 < \delta < 1$, $\mathbf{P}$ is drawn from an $(\varepsilon, \delta)$ oblivious $\ell^2$-subspace embedding over $\tilde{n} \times n$ matrices. Then with probability at least $1 - \delta$ with $\delta \in (0, 1)$, the output of Iterative SRO described by Algorithm 1 satisfies

$$\left\| \boldsymbol{\beta}^{(N)} - \boldsymbol{\beta}^* \right\|_{\mathbf{X}} \le \rho^N \|\boldsymbol{\beta}^*\|_{\mathbf{X}} \tag{8}$$

for a constant $0 < \rho < 1$ if $h$ is convex, or the Frechet subdifferential of $h$ is $L_h$-smooth and $\mathbf{X}$ has full column rank with $\frac{L_h}{\sigma_{\min}^2(\mathbf{X})} < (1 - \varepsilon)$. Frechet subdifferential of $h$ is $L_h$-smooth if $\sup_{\mathbf{u} \in \tilde{\partial} h(\mathbf{x}), \mathbf{v} \in \tilde{\partial} h(\mathbf{y})} \|\mathbf{u} - \mathbf{v}\|_2 \le L_h \|\mathbf{x} - \mathbf{y}\|_2$ for a positive number $L_h$. In particular, if $\mathbf{P}$ is a Gaussian subspace embedding, then $\tilde{n} = \mathcal{O}\left( \left( r + \log \frac{1}{\delta} \right) \cdot (\rho + 1)^2 / \rho^2 \right)$. If $\mathbf{P}$ is a sparse subspace embedding, then $\tilde{n} = \mathcal{O}\left( r^2 / \delta \cdot (\rho + 1)^2 / \rho^2 \right)$. Here $r = \mathrm{rank}(\mathbf{X})$.

Section 5 presents necessary theoretical results for the proof of Theorem 3.1.

## 4 SKETCHING FOR SPARSE SIGNAL ESTIMATION

We study sparse signal estimation by sketching in this section. We consider the linear model widely used in the sparse signal estimation literature, $\bar{\mathbf{y}} = \bar{\mathbf{X}}\bar{\boldsymbol{\beta}} + \boldsymbol{\varepsilon}$ where $\boldsymbol{\varepsilon}$ is a noise vector of i.i.d. sub-gaussian elements with variance proxy $\sigma^2$, and $\bar{\boldsymbol{\beta}}$ is the sparse parameter vector of interest. Following the standard analysis for parameter estimation in the literature such as (Yang et al., 2016; Zhang, 2010b), we assume $\max_{i \in [d]} \left\| \bar{\mathbf{X}}^i \right\|_2 \le \sqrt{n}$, and it follows that $\max_{i \in [d]} \left\| \mathbf{X}^i \right\|_2 \le 1$. The statistical learning literature has extensively studied the approximation to $\bar{\boldsymbol{\beta}}$ by the M-estimator obtained as a globally or locally optimal solution to problem (1) with $\mathbf{X} = \bar{\mathbf{X}}/\sqrt{n}$, $\mathbf{y} = \bar{\mathbf{y}}/\sqrt{n}$. That is, one hopes to approximate $\bar{\boldsymbol{\beta}}$ by the globally or locally optimal solution to problem (1) with a suitable sparsity-inducing regularizer $h_\lambda$. In Subsection 4.1, we show that the Iterative SRO described in Algorithm 1 achieves the minimax parameter estimation error of the order $\sqrt{\bar{s} \log d / n}$ where $\bar{s} = \left\| \bar{\boldsymbol{\beta}} \right\|_0$. In Subsection 4.2, we prove that SRO achieves the minimax parameter estimation error of the order $\sqrt{\bar{s} \log d / n}$ for sparse nonconvex learning, where the nonconvex regularizer $h_\lambda$ is the sum of a concave penalty function $q_\lambda$ and $\lambda \|\cdot\|_1$. In Section 4.1, $\tilde{\boldsymbol{\beta}}^*$ is obtained by Algorithm 1 through $\tilde{\boldsymbol{\beta}}^* = \boldsymbol{\beta}^{(N)}$. In Section 4.2, $\tilde{\boldsymbol{\beta}}^*$ is the optimization result of the sketched problem (2).

### 4.1 SKETCHING FOR SPARSE CONVEX LEARNING

We define $\mathcal{L}(\boldsymbol{\beta}) := 1/2 \cdot \boldsymbol{\beta}^\top \mathbf{X}^\top \mathbf{X} \boldsymbol{\beta} - \mathbf{y}^\top \mathbf{X} \boldsymbol{\beta}$ and $\tilde{\mathcal{L}}(\boldsymbol{\beta}) := 1/2 \cdot \tilde{\mathbf{X}}^\top \tilde{\mathbf{X}} \boldsymbol{\beta} - \mathbf{y}^\top \mathbf{X} \boldsymbol{\beta}$. We introduce the following definition of sparse eigenvalues widely used in sparse signal estimation literature.

**Definition 4.1.** (Sparse Eigenvalues) Let $s$ be a positive integer. The largest and smallest $s$-sparse eigenvalues of the Hessian matrix $\nabla^2 \mathcal{L}(\boldsymbol{\beta}) = \mathbf{X}^\top \mathbf{X}$ is

$$\rho_{\mathcal{L},+}(s) := \sup \left\{ \mathbf{v}^\top \mathbf{X}^\top \mathbf{X} \mathbf{v} \colon \|\mathbf{v}\|_0 \le s, \|\mathbf{v}\|_2 = 1, \mathbf{v} \in \mathbb{R}^d \right\}, \tag{9}$$

$$\rho_{\mathcal{L},-}(s) := \inf \left\{ \mathbf{v}^\top \mathbf{X}^\top \mathbf{X} \mathbf{v} \colon \|\mathbf{v}\|_0 \le s, \|\mathbf{v}\|_2 = 1, \mathbf{v} \in \mathbb{R}^d \right\}. \tag{10}$$

$\rho_{\tilde{\mathcal{L}},+}(\cdot)$ and $\rho_{\tilde{\mathcal{L}},-}(\cdot)$ are defined in a similar manner with $\mathbf{X}$ replaced by $\tilde{\mathbf{X}}$.

The following assumption is frequently used in the sparse signal estimation literature with the convex sparsity-inducing penalty, $\lambda \|\cdot\|_1$.

**Assumption 1.** (Assumption in (Yang et al., 2016; Zhang, 2010b) for sparse signal estimation) $\rho_{\mathcal{L},+}(s) < \infty, \rho_{\mathcal{L},-}(s) > 0$ are positive constants. Moreover, for $\bar{s} = \|\bar{\boldsymbol{\beta}}\|_0$, there exists a $k^* \in \mathbb{N}$ such that $k^* \geq 2\bar{s}$ and

$$\rho_{\mathcal{L},+}(k^*)/\rho_{\mathcal{L},-}(2k^* + \bar{s}) \leq 1 + 0.5k^*/\bar{s}. \tag{11}$$

We study the sparse signal estimation problem by solving the sketched Lasso problem with $h_\lambda(\boldsymbol{\beta}) = \lambda \|\boldsymbol{\beta}\|_1$ in the original problem (1) and the sketched problem (2) using our Iterative SRO algorithm. We have the following sharp bound for the parameter estimation error.

**Theorem 4.1.** Suppose Assumption 1 holds. Let $\lambda = c\sigma\sqrt{\log d/n}$ where $c$ is a positive constant. Suppose Algorithm 1 returns $\tilde{\boldsymbol{\beta}}^* = \boldsymbol{\beta}^{(N)}$ with $\rho \in (0,1)$ in Theorem 3.1, and the iteration number $N$ is chosen as $N = 1 + \log\left(\|\mathbf{X}\|_2\|\boldsymbol{\beta}^*\|_{\mathbf{X}}/(\lambda\mu)\right)/\log(1/\rho)$. Then with probability at least $1 - \delta - 2/d$ with $\delta \in (0,1)$,

$$\left\|\tilde{\boldsymbol{\beta}}^* - \bar{\boldsymbol{\beta}}\right\|_2 \leq \frac{(1+\gamma)(c + \mu c + 2)\sigma}{\rho_{\mathcal{L},-}(\bar{s} + k^*) \cdot \left(1 - \gamma\sqrt{0.5}\right)}\sqrt{\frac{\bar{s}\log d}{n}}, \tag{12}$$

where $\mu$ is a positive constant, $\gamma = (1 + \mu + 2/c)/(1 - \mu - 2/c)$, and $\mu$ and $c$ are chosen such that $\gamma\sqrt{0.5} < 1$. In particular, if $\mathbf{P}$ is a Gaussian subspace embedding, then $\tilde{n} = \mathcal{O}\left((r + \log\frac{1}{\delta}) \cdot (\rho + 1)^2/\rho^2\right)$. If $\mathbf{P}$ is a sparse subspace embedding, then $\tilde{n} = \mathcal{O}\left(r^2/\delta \cdot (\rho + 1)^2/\rho^2\right)$. Here $r = \text{rank}(\mathbf{X})$.

Theorem 4.1 shows that our Iterative ROS described in Algorithm 1 applied on the sketched problem (2) achieves the parameter estimation error, which is $\left\|\tilde{\boldsymbol{\beta}}^* - \bar{\boldsymbol{\beta}}\right\|_2$, of the order $\sqrt{\bar{s}\log d/n}$. Such estimation error rate is not improvable and it is the minimax error rate for standard Lasso. With the rank $r \ll n$, Iterative ROS obtains the solution $\tilde{\boldsymbol{\beta}}^* = \boldsymbol{\beta}^{(N)}$ efficiently with a small sketch size $\tilde{n}$ and the iteration number $N$ is only of a logarithmic order.

## 4.2 SKETCHING FOR SPARSE NONCONVEX LEARNING

We now study sparse signal estimation by sparse nonconvex learning where the regularizer $h_\lambda$ is nonconvex in the original problem (1) and the sketched problem (2), that is, $h_\lambda(\boldsymbol{\beta}) = \lambda \|\boldsymbol{\beta}\|_1 + Q_\lambda(\boldsymbol{\beta})$ where $Q_\lambda(\boldsymbol{\beta}) := \sum_{j=1}^{d} q_\lambda(\beta_j)$, $q_\lambda$ is a concave function and $\beta_j$ is the $j$-th element of $\boldsymbol{\beta}$. We have $h_\lambda(\boldsymbol{\beta}) = \sum_{j=1}^{d}(\lambda|\beta_j| + q_\lambda(\beta_j))$. Following the analysis of sparse parameter vector recovery in (Wang et al., 2014), $\lambda|\cdot| + q_\lambda(\cdot)$ is a nonconvex function which can be either smoothly clipped absolute deviation (SCAD) (Fan & Li, 2001) or minimax concave penalty (MCP) (Zhang, 2010a). More details about the nonconvex regularizer $h_\lambda$ are deferred to Section A of the appendix. The following regularity conditions on the concave function $q_\lambda$ are used in (Wang et al., 2014).

**Assumption 2.** (Regularity Conditions on Nonconvex Penalty in (Wang et al., 2014) for sparse signal recovery)

(a) $q'_\lambda(\beta_j)$ is monotone and Lipschitz continuous. For $\beta'_j > \beta_j$, there exist two constants $\zeta_- \geq 0, \zeta_+ \geq 0$ such that $-\zeta_- \leq \frac{q'_\lambda(\beta'_j) - q'_\lambda(\beta_j)}{\beta'_j - \beta_j} \leq -\zeta_+$.

(b) $q_\lambda(-\beta_j) = q_\lambda(\beta_j)$ for all $\beta_j \in \mathbb{R}$. Also, $q_\lambda(0) = q'_\lambda(0) = 0$.

(c) $q'_\lambda(\beta_j) \leq \lambda$ for all $\beta_j \in \mathbb{R}$, and $\left|q'_{\lambda_1}(\beta_j) - q'_{\lambda_2}(\beta_j)\right| \leq |\lambda_1 - \lambda_2|$ for all $\lambda_1 > 0, \lambda_2 > 0$.

The following assumption is the standard assumption in (Wang et al., 2014) for sparse signal estimation with the minimax error rate, that is, $\|\boldsymbol{\beta}^* - \bar{\boldsymbol{\beta}}\|_2 \leq \mathcal{O}\left(\sqrt{\bar{s}\log d/n}\right)$.

**Assumption 3.** (Assumption in (Wang et al., 2014) for sparse signal estimation) Let $\bar{s} = \|\bar{\boldsymbol{\beta}}\|_0$. There exist an integer $\tilde{s}$ such that $\tilde{s} > C\bar{s}$ such that $\rho_{\mathcal{L},+}(\bar{s} + 2\tilde{s}) < \infty, \rho_{\mathcal{L},-}(\bar{s} + 2\tilde{s}) > 0$ are two absolute constants. The concavity parameter $\zeta_-$ in Assumption 2 satisfies $\zeta_- \leq C'\rho_{\mathcal{L},-}(\bar{s} + 2\tilde{s})$ with constant $C' \in (0,1)$. Here $C = 144\kappa^2 + 250\kappa$ with $\kappa = (\rho_{\mathcal{L},+}(\bar{s} + 2\tilde{s}) - \zeta_+)/(\rho_{\mathcal{L},-}(\bar{s} + 2\tilde{s}) - \zeta_-)$.

The following corollary shows that when the support size of $\tilde{\boldsymbol{\beta}}^* - \boldsymbol{\beta}^*$ is bounded by $s_0$, then the nonconvex sparse learning problem (1) with $h_\lambda$ being the nonconvex regularizer specified in this subsection still enjoys relative-error approximation. Corollary 4.2 is employed to prove our main result of minimax estimation error rate by sketching in Theorem 4.5.

**Corollary 4.2.** Suppose Assumption 3 holds, and let $\mathbf{P} \in \mathbb{R}^{\tilde{n} \times n}$ be a Gaussian subspace embedding and suppose that $\left| \mathrm{supp}\left( \tilde{\boldsymbol{\beta}}^* - \boldsymbol{\beta}^* \right) \bigcup \mathrm{supp}\left( \boldsymbol{\beta}^* \right) \bigcup \mathrm{supp}\left( \tilde{\boldsymbol{\beta}}^* \right) \right| \le s_0$ for some integer $s_0 \in [d]$. Let $\tilde{n} \ge c_0 \varepsilon^{-2} \left( \log(2/\delta) + s_0 \log d + s_0 \log 5 \right)$ for $\delta \in (0,1)$ and $\varepsilon \in (0, 1 - C')$ where $c_0$ is a positive constant. Then with probability at least $1 - \delta$,

$$\left\| \tilde{\boldsymbol{\beta}}^* - \boldsymbol{\beta}^* \right\|_2 \le \frac{\varepsilon \sqrt{\rho_{\mathcal{L},+}(s_0)}}{(1-\varepsilon)\rho_{\mathcal{L},-}(s_0) - \zeta_-} \|\boldsymbol{\beta}^*\|_{\mathbf{X}}. \tag{13}$$

The following assumption, Assumption 4, is necessary to achieve the minimax parameter estimation error by sketching, and the subsequent Remark 4.3 explains that Assumption 4 is mild. That is, if Assumption 3 holds, then Assumption 4 also holds under mild conditions.

**Assumption 4.** (Assumption in (Wang et al., 2014) for sparse signal estimation) Let $\tilde{s}, C'$ be the parameters specified in Assumption 3 such that Assumption 3 holds. Then it is assumed that $\rho_{\tilde{\mathcal{L}},+}(\bar{s} + 2\tilde{s}) < \infty, \rho_{\tilde{\mathcal{L}},-}(\bar{s} + 2\tilde{s}) > 0$ are two absolute constants. In addition, $\zeta_-$ satisfies $\zeta_- \le C' \rho_{\tilde{\mathcal{L}},-}(\bar{s} + 2\tilde{s})$, and $\tilde{s} > \tilde{C}\bar{s}$ where $\tilde{C} = 144\tilde{\kappa}^2 + 250\tilde{\kappa}$ with $\tilde{\kappa} := (\rho_{\tilde{\mathcal{L}},+}(\bar{s} + 2\tilde{s}) - \zeta_+)/(\rho_{\tilde{\mathcal{L}},-}(\bar{s} + 2\tilde{s}) - \zeta_-)$.

**Remark 4.3** (Assumption 4 is mild)**.** We provide theoretical justification that Assumption 4 is mild. The following theorem, Theorem 4.4, shows that if the standard Assumption 3 holds, then Assumption 4 also holds with high probability under very mild conditions: either $\varepsilon$ is set to the order of $\sqrt{\log d/n}$ with sufficiently large $n$, or the Restricted Isometry Property (RIP) (Candes & Tao, 2005b) holds. It is well known that RIP holds for various choices of the design matrix $\mathbf{X}$, and (Wang et al., 2014) also uses RIP to justify that the standard Assumption 3 is weaker than RIP.

**Theorem 4.4.** Suppose Assumption 3 holds with $\tilde{s}$ and $C'$ specified in Assumption 3, and let $s_0 = \bar{s} + 2\tilde{s}$. Let $0 < \varepsilon, \delta < 1$, $\mathbf{P} \in \mathbb{R}^{\tilde{n} \times n}$ be a Gaussian subspace embedding defined in Definition 2.2, and $\tilde{n} \ge c_0 \varepsilon^{-2} \left( \log(2/\delta) + s_0 \log d + s_0 \log 5 + 1/d^{s_0 - 1} \right)$ where $c_0$ is a positive constant. If $\rho_{\mathcal{L},-}(s_0) > \varepsilon\sqrt{s_0}$, $\zeta_- \le C'\left(\rho_{\mathcal{L},-}(s_0) - \varepsilon\sqrt{s_0}\right)$ and $\left(144\kappa'^2 + 250\kappa'\right)\bar{s} < \tilde{s}$ with $\kappa' = (\rho_{\mathcal{L},+}(s_0) + \varepsilon\sqrt{s_0} - \zeta_+)/(\rho_{\mathcal{L},-}(s_0) - \varepsilon\sqrt{s_0} - \zeta_-)$, then with probability at least $1 - \delta$, Assumption 4 holds. In particular, Assumption 4 holds with probability at least $1 - \delta$ if any one of the following two conditions holds:

(a) $\log d/n \overset{n \to \infty}{\longrightarrow} 0$, $\varepsilon = C_1\sqrt{\log d/n}$ with $C_1$ being a positive constant and $n$ sufficiently large;

(b) There exists $s' \ge s_0$ such that $\mathrm{RIP}(\delta, s')$ holds for $\delta \in (0,1)$, $\zeta_+ = 0$, $\zeta_- = C_2\rho_{\mathcal{L},-}(s_0)$, $\varepsilon\sqrt{s_0} \le C_3\rho_{\mathcal{L},-}(s_0)$, $\tilde{s} > \left(144\kappa_0^2 + 250\kappa_0\right)\bar{s}$ with $\kappa_0 = ((1+C_3)(1+\delta))/((1 - C_2 - C_3)(1-\delta))$. Here the positive constants $C_2, C_3$ satisfy $C_2 + C_3 < 1$ and $C_2 \le C'(1 - C_3)$. $\mathrm{RIP}(\delta, s)$ for $\delta \in (0,1)$ and $s \in \mathbb{N}$ is the Restricted Isometry Property (RIP) (Candes & Tao, 2005b) under which $1 - \delta \le \rho_{\mathcal{L},-}(s) \le \rho_{\mathcal{L},+}(s) \le 1 + \delta$ holds.

It is shown in Section A that $\zeta_+ = 0$ and $\zeta_- = C_2\rho_{\mathcal{L},-}(s_0)$ can be easily achieved by setting the hyperparameter of MCP when MCP is used as the nonconvex regularizer $h_\lambda$. We have the following sharp bound for the parameter estimation error with sparse nonconvex learning by sketching in Theorem 4.5. We note that the approximate path following method described in (Wang et al., 2014, Algorithm 1) is used to solve the original problem (1) and the sketched problem (2) to obtain $\tilde{\boldsymbol{\beta}}^*$ and $\bar{\boldsymbol{\beta}}^*$ such that $\bar{\boldsymbol{\beta}}^*$ is an critical point of problem (1) and $\tilde{\boldsymbol{\beta}}^*$ is an critical point of (2). We use $\lambda = \Theta\left(\sqrt{\bar{s}\log d/n}\right)$ for both (1) and (2) at the final stage of the path following method (Wang et al., 2014, Algorithm 1).

**Theorem 4.5.** Let $\delta \in (0,1)$ and $\mathbf{P} \in \mathbb{R}^{\tilde{n} \times n}$ be a Gaussian subspace embedding defined in Definition 2.2, and $\varepsilon = \min\left\{ C_1\sqrt{\log d/n}, \varepsilon_0 \right\}$ with $C_1, \varepsilon_0$ being positive constants and $\varepsilon \in (0, (1 - C')/2)$. Suppose $\tilde{\boldsymbol{\beta}}^*$ is the optimization result of the sketched problem (2) with sketch size $\tilde{n} \ge n/C_3^2$ for $n \ge \Theta(1)$, Assumption 3 and Assumption 4 hold, $d \ge 5$, and let $s_0 = \bar{s} + 2\tilde{s}$. Then with probability $1 - \delta$,, with probability at least $1 - 4/d$,

$$\left\| \tilde{\boldsymbol{\beta}}^* - \bar{\boldsymbol{\beta}} \right\|_2 \le \frac{C_1\|\bar{\boldsymbol{\beta}}\|_2\sqrt{(1 + \tilde{s}/\bar{s})\rho_{\mathcal{L},+}(s_0)}}{(1 + C')/2 \cdot \rho_{\mathcal{L},-}(s_0) - \zeta_-}\sqrt{\frac{\bar{s}\log d}{n}} + \frac{22\left(C_1\sqrt{\bar{s}}\|\bar{\boldsymbol{\beta}}\|_2 + 2\sigma\right)}{\rho_{\mathcal{L},-}(s_0) - \zeta_-}\sqrt{\frac{\bar{s}\log d}{n}}, \tag{14}$$

where $C_1 = \sqrt{c_0 792 s_0 / 789} C_3$ with $C_3 > 1$ being a positive constant.

It is noted that we can choose $\tilde{s} = \Theta(\bar{s})$. When $\left\|\bar{\boldsymbol{\beta}}\right\|_2$ is a constant, we have the parameter estimation error $\left\|\tilde{\boldsymbol{\beta}}^* - \bar{\boldsymbol{\beta}}\right\|_2 \leq \mathcal{O}\left(\sqrt{\bar{s} \log d / n}\right)$, which is the minimax estimation error according to (Wang et al., 2014). Moreover, with $C_1 = 4\sqrt{792 s_0 / 789} C_3$, we can enjoy a small sketch size $\tilde{n} = n / C_3^2$ with a potentially large $C_3 > 1$, and this is at the expense of having a large constant factor $C_1$ in the parameter estimation error $\mathcal{O}\left(\sqrt{\bar{s} \log d / n}\right)$.

## 5 Approximation Error Bounds for the Proof of Theorem 3.1

In this section, we present an introduction to the theoretical results, Theorem D.2 and Theorem D.3 deferred to Section D.2 of the appendix, which are necessary for the proof of Theorem 3.1. First, a novel universal approximation error bound for SRO on the general problem (1) is given by Theorem D.2. Based on such universal approximation error bound, relative-error approximation bounds for regularized least squares with convex regularization and nonconvex regularization under certain conditions are derived in Theorem D.2. Before introducing these results, we present the definitions of Frechet subdifferential and critical point below, which are crucial for our analysis.

**Definition 5.1.** (Subdifferential and critical points) Given a nonconvex function $f \colon \mathbb{R}^d \to \mathbb{R} \cup \{+\infty\}$ which is a proper and lower semi-continuous function,

- for a given $\mathbf{x} \in \mathrm{dom} f$, its Frechet subdifferential of $f$ at $\mathbf{x}$, denoted by $\tilde{\partial} f(\mathbf{x})$, is the set of all vectors $\mathbf{u} \in \mathbb{R}^d$ which satisfy $\liminf_{\mathbf{y} \neq \mathbf{x}, \mathbf{y} \to \mathbf{x}} \frac{f(\mathbf{y}) - f(\mathbf{x}) - \langle \mathbf{u}, \mathbf{y} - \mathbf{x} \rangle}{\|\mathbf{y} - \mathbf{x}\|_2} \geq 0$.

- The limiting-subdifferential of $f$ at $\mathbf{x} \in \mathbb{R}^d$, denoted by $\partial f(\mathbf{x})$, is defined by $\partial f(\mathbf{x}) = \{\mathbf{u} \in \mathbb{R}^d \colon \exists \mathbf{x}^k \to \mathbf{x}, f(\mathbf{x}^k) \to f(\mathbf{x}), \tilde{\mathbf{u}}^k \in \tilde{\partial} f(\mathbf{x}^k) \to \mathbf{u}\}$. The point $\mathbf{x}$ is a critical point of $f$ if $\mathbf{0} \in \partial f(\mathbf{x})$.

It is noted that the Frechet subdifferential generalizes the notions of Frechet derivative and subdifferential of convex functions. When $f$ is convex, then $\tilde{\partial} f(\mathbf{x})$ is the subdifferential of $f$ at $\mathbf{x}$. Moreover, $\tilde{\partial} f(\mathbf{x}) = \{\nabla f(\mathbf{x})\}$ when $f$ is differentiable. In order to derive the relative-error approximation bounds in this section, we need the following definition of the degree of nonconvexity of a function. The univariate degree of nonconvexity is first introduced in (Zhang & Zhang, 2012) for the analysis of the consistency of nonconvex sparse estimation models with concave regularization, and Definition 5.2 is an extension of such univariate degree of nonconvexity, which is also employed in the analysis of sketching for regularized optimization in (Yang & Li, 2021).

**Definition 5.2.** The degree of nonconvexity of a function $h \colon \mathbb{R}^d \to \mathbb{R}$ at a point $\mathbf{t} \in \mathbb{R}^d$ is defined as

$$\theta_h(\mathbf{t}, \kappa) \coloneqq \sup_{\mathbf{s} \in \mathbb{R}^d, \mathbf{s} \neq \mathbf{t}, \mathbf{u} \in \tilde{\partial} h(\mathbf{s}), \mathbf{v} \in \tilde{\partial} h(\mathbf{t})} \frac{-(\mathbf{s} - \mathbf{t})^\top (\mathbf{u} - \mathbf{v}) - \kappa \|\mathbf{s} - \mathbf{t}\|_2^2}{\|\mathbf{s} - \mathbf{t}\|_2}, \tag{15}$$

where $\kappa \in \mathbb{R}$. We abbreviate (15) as $\theta_h(\mathbf{t}, \kappa) \triangleq \sup_{\mathbf{s} \in \mathbb{R}^d, \mathbf{s} \neq \mathbf{t}} \{-\frac{1}{\|\mathbf{s} - \mathbf{t}\|_2} (\mathbf{s} - \mathbf{t})^\top (\tilde{\partial} h(\mathbf{s}) - \tilde{\partial} h(\mathbf{t})) - \kappa \|\mathbf{s} - \mathbf{t}\|_2\}$ in the following text.

**Remark 5.1.** If $h$ is convex, then $-(\mathbf{s} - \mathbf{t})^\top (\mathbf{u} - \mathbf{v}) \leq 0$, so that the degree of nonconvexity of any convex function $h$ is $\theta_h(\mathbf{t}, \kappa) \leq 0$ for any with $\kappa \geq 0$. Moreover, the degree of nonconvexity of a second-order differentiable function $h$ satisfies $\theta_h(\mathbf{t}, \kappa) \leq 0$ if $h$ is "more PSD" than $-\kappa \|\cdot\|_2^2$, that is, the smallest eigenvalue of its Hessian is not less than $-\kappa$.

Let $\tilde{\boldsymbol{\beta}}^*$ be any critical point of the objective function in (2), $\boldsymbol{\beta}^*$ be any critical point of the objective function in (1), and $0 < \varepsilon < \varepsilon_0 < 1$ where $\varepsilon_0$ is a small positive constant, $0 < \delta < 1$. Suppose $\mathbf{P}$ is drawn from an $(\varepsilon, \delta)$ oblivious $\ell^2$-subspace embedding over $\tilde{n} \times n$ matrices. Then Theorem D.2 asserts that with probability $1 - \delta$,

$$(1 - \varepsilon)\left\|\tilde{\boldsymbol{\beta}}^* - \boldsymbol{\beta}^*\right\|_{\mathbf{X}}^2 - \varepsilon \left\|\tilde{\boldsymbol{\beta}}^* - \boldsymbol{\beta}^*\right\|_{\mathbf{X}} \|\boldsymbol{\beta}^*\|_{\mathbf{X}} \leq \theta_{h_\lambda}(\boldsymbol{\beta}^*, \kappa)\left\|\tilde{\boldsymbol{\beta}}^* - \boldsymbol{\beta}^*\right\|_2 + \kappa \left\|\tilde{\boldsymbol{\beta}}^* - \boldsymbol{\beta}^*\right\|_2^2. \tag{16}$$

In particular, if $\mathbf{P}$ is a Gaussian subspace embedding, then $\tilde{n} = \mathcal{O}((r + \log \frac{1}{\delta})\varepsilon^{-2})$. If $\mathbf{P}$ is a sparse subspace embedding, then $\tilde{n} = \mathcal{O}(r^2/(\delta\varepsilon^2))$.

Furthermore, Theorem D.3 states that if $h_\lambda$ is convex, then with probability $1 - \delta$, then $\left\|\tilde{\boldsymbol{\beta}}^* - \boldsymbol{\beta}^*\right\|_{\mathbf{X}} \leq \frac{\varepsilon}{1-\varepsilon}\|\boldsymbol{\beta}^*\|_{\mathbf{X}}$. If the Frechet subdifferential of $h$ is $L_h$-smooth ($h$ can be nonconvex), then $\left\|\tilde{\boldsymbol{\beta}}^* - \boldsymbol{\beta}^*\right\|_{\mathbf{X}} \leq \frac{\varepsilon}{(1-\varepsilon) - \frac{\lambda L_h}{\sigma_{\min}^2(\mathbf{X})}}\|\boldsymbol{\beta}^*\|_{\mathbf{X}}$.

**Novelty and Our Results and Their Significant Difference from (Yang & Li, 2021) and (Pilanci & Wainwright, 2016).** It is remarked that our results, including the Iterative SRO algorithm in Algorithm 1 and its theoretical guarantee in Theorem 3.1, Theorem D.2-Theorem D.3, and the minimax optimal rates by sketching for sparse convex learning in Theorem 4.1 and sparse nonconvex learning in Theorem 4.5, are all novel and significantly different from (Yang & Li, 2021) in the following two aspects, although (Yang & Li, 2021) also presents an iterative sketching algorithm for regularized optimization problems. First, Iterative SRO does not need to sample a projection matrix $\mathbf{P} \in \mathbb{R}^{\tilde{n} \times n}$ and compute the sketched matrix $\tilde{\mathbf{X}} = \mathbf{P}\mathbf{X}$ at every iteration, while the iterative sketching algorithm in (Yang & Li, 2021) samples a different projection matrix and computes the sketched matrix at every iteration which incurs considerable computational cost for large-scale problem with large data size $n$. Such advantage of Iterative SRO over (Yang & Li, 2021) is attributed to the novel theoretical results in Theorem D.2 and Theorem D.3. In contrast with (Yang & Li, 2021, Theorem 1), the approximation error bound Theorem D.2 is derived for sketching low-rank data matrix by oblivious $\ell^2$-subspace embedding with the sketched size $\tilde{n}$ clearly specified. As a result, Theorem D.3 presents the approximation error bounds for convex and certain nonconvex regularization by sketching with oblivious $\ell^2$-subspace embedding. Based on such results, Theorem 3.1 shows that a single projection matrix suffices for the iterative sketching process. Second, minimax optimal rates for convex and nonconvex sparse learning problems by sketching are established by our results, while there are no such minimax optimal rates by a sketching algorithm in (Yang & Li, 2021). Theorem 4.1, to the best of our knowledge, is among the first in the literature which uses an iteratively sketching algorithm to achieve the minimax optimal rate for sparse convex learning. Furthermore, Theorem 4.5 shows that sketching can also lead to the minimax optimal rate even for sparse nonconvex problems, while sketching for nonconvex problems is still considered difficult and open in the literature.

Our results are also significantly different from those in (Pilanci & Wainwright, 2016). It is remarked that (Pilanci & Wainwright, 2016) only handles convex constrained least square problems of the form $\min_{\mathbf{x} \in \mathcal{C}} \|\mathbf{x}\boldsymbol{\beta} - \mathbf{y}\|_2^2$ where the constraint set $\mathcal{C}$ is a convex set, while our results cover regularized convex and nonconvex problems with minimax optimal rates. It is emphasized that the techniques in (Pilanci & Wainwright, 2016) can never be applied to the regularized problems considered in this paper. (Pilanci & Wainwright, 2016) heavily relies on certain complexity measure of the constraint set $\mathcal{C}$, such as the Gaussian width. It shows that the complexity of such constraint set $\mathcal{C}$ is bounded, so that sketching with such constraint set $\mathcal{C}$ of limited complexity only incurs a relatively small approximation error. However, there is never such constraint set in the original problem (1) or the sketched problem (2), so that such complexity based analysis for sketching can not be applied to this work. Furthermore, as mentioned in Section 1.1, Iterative SRO does not need to sample the projection matrix and compute the sketched matrix at each iteration, in contrast with IHS (Pilanci & Wainwright, 2016) where a separate projection matrix is sampled and the sketched matrix is computed at each iteration. As evidenced by Table 1 in Section 7.1, Iterative SRO is more efficient than its "IHS" counterpart where sketching is performed at every iteration while enjoying comparable approximation error.

## 6 TIME COMPLEXITY

We compare the time complexity of solving the original problem (1) to that of solving the sketched problem (2) with Iterative SRO, which is deferred to Section B of the appendix.

## 7 EXPERIMENTAL RESULTS

We provide empirical results in this section to justify the effectiveness of the proposed SRO and Iterative SRO.

## 7.1 GENERALIZED LASSO

We study the performance of Iterative SRO for Generalized Lasso (GLasso) (Tibshirani & Taylor, 2011) in this subsection. The optimization problem of an instance of GLasso studied here is $\boldsymbol{\beta}^* = \arg\min_{\boldsymbol{\beta} \in \mathbb{R}^d} \frac{1}{2} \|\mathbf{y} - \mathbf{X}\boldsymbol{\beta}\|_2^2 + \lambda \sum_{i=1}^{d-1} |\boldsymbol{\beta}_i - \boldsymbol{\beta}_{i+1}|$, which is solved by Fast Iterative Shrinkage-Thresholding Algorithm (FISTA) (Beck & Teboulle, 2009), an accelerated version of PGD. $\bar{\mathbf{X}} = \sqrt{n}\mathbf{X} \in \mathbb{R}^{n \times d}$ has i.i.d. standard Gaussian entries with $n = 80000$ and $d = 600$, and all the elements of $\bar{\mathbf{y}} = \sqrt{n}\mathbf{y}$ are also i.i.d. Gaussian samples. Figure 4 in Section C.2 of the appendix illustrates the approximation error of SRO and Iterative SRO, which are $\frac{\|\boldsymbol{\beta}^{(1)} - \boldsymbol{\beta}^*\|_{\mathbf{x}}^2}{n}$ and $\frac{\|\boldsymbol{\beta}^{(N)} - \boldsymbol{\beta}^*\|_{\mathbf{x}}^2}{n}$ respectively, for different choices of sketch size $\tilde{n}$ with $\tilde{n} = \bar{\gamma}d$. We set $N = 10$ and employ either sparse subspace embedding or Gaussian subspace embedding. The average approximation errors are reported over 100 trials of data sampling for each $\bar{\gamma}$. It can be observed that Iterative SRO significantly reduces the approximation error and its approximation error is roughly $\frac{1}{3}$ of that of SRO, demonstrating the effectiveness of Iterative SRO. In our experiment, due to the significant reduction in the sample size, for example, $\frac{\bar{\gamma}d}{n} = 0.0225$ when $\bar{\gamma} = 3$, the running time of Iterative SRO is always less than half of that required to solve the original problem with small $\bar{\gamma}$.

We also report the running time of GLasso with sparse subspace embedding in Table 1. Let $M$ be the maximum number of iterations for FISTA, and we set $M = 10000$ for SRO and set $M = 2000$ for Iterative SRO, and the running time is reported for $\bar{\gamma} = 3$ on a CPU of Intel i5-11300H. Iterative SRO-IHS is the "IHS" version of Iterative SRO where a new sketch matrix $\mathbf{P}$ is sampled and the sketched data $\tilde{\mathbf{X}} = \mathbf{P}\mathbf{X}$ is computed at each iteration. We observed that both Iterative SRO-IHS and Iterative SRO achieve the same approximation error, while Iterative SRO is faster than Iterative SRO-IHS because the former only samples the linear transformation $\mathbf{P}$ once and computes the sketched matrix $\tilde{\mathbf{X}}$ once. We also note that the maximum iteration number of FISTA for Iterative SRO is much smaller than that for SRO. This is because Iterative SRO uses an iterative sketching process where the approximation error is geometrically reduced with respect to the iteration number $t$ in Algorithm 1, so that each iteration of Iterative SRO is only required to have a moderate approximation error which can be larger than the approximation error of SRO thus a smaller iteration number of FISTA suffices for Iterative SRO. Such analysis also explains the fact that Iterative SRO is much faster than SRO, and in our experiment the maximum iteration number $N$ for Iterative SRO in Algorithm 1 is always not greater than 5.

Table 1: Running time (in seconds) of SRO, Iterative SRO with $\bar{\gamma} = 3$ and Iterative SRO-IHS for GLasso. The number in the bracket is the approximation error.

| SRO | Iterative SRO | Iterative SRO-IHS |
|---|---|---|
| $11.04s$ | $4.71s(0.036)$ | $5.11s(0.036)$ |

## 7.2 ADDITIONAL EXPERIMENTS

We defer more experimental results to the Section C of the appendix. In particular, experimental results for ridge regression and sparse signal estimation by Lasso are in Section C.1 and Section C.3 respectively, and more details about GLasso are in Section C.2. We further apply SRO to subspace clustering using Lasso in Section C.4, and presents performance of SRO for sparse nonconvex learning with capped-$\ell^1$ regularization in Section C.5.

## 8 CONCLUSION

We present Sketching for Regularized Optimization (SRO) which efficiently solves general regularized optimization problems with convex or nonconvex regularization by sketching. We further propose Iterative SRO to reduce the approximation error of SRO geometrically, and provide a unified theoretical framework under which the minimax rates for sparse signal estimation are obtained for both convex and nonconvex sparse learning problems. Experimental results evidence that Iterative SRO can effectively and efficiently approximate the optimization result of the original problem.

ACKNOWLEDGMENTS

This material is based upon work supported by the U.S. Department of Homeland Security under Grant Award Number 17STQAC00001-07-00. The views and conclusions contained in this document are those of the authors and should not be interpreted as necessarily representing the official policies, either expressed or implied, of the U.S. Department of Homeland Security. This work is also partially supported by the 2023 Mayo Clinic and Arizona State University Alliance for Health Care Collaborative Research Seed Grant Program under Grant Award Number AWD00038846.

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

## A  NONCONVEX PENALTY

We introduce more details about the nonconvex regularizer $h_\lambda$ for sparse nonconvex learning in Subsection 4.2. $h_\lambda = \sum_{j=1}^d p_\lambda(\beta_j)$, and $p_\lambda$ can be either smoothly clipped absolute deviation (SCAD) (Fan & Li, 2001) or minimax concave penalty (MCP) (Zhang, 2010a). When $p_\lambda$ is SCAD, we have

$$p_\lambda(\beta_j) = \lambda \int_0^{|\beta_j|} \left( \mathbb{I}_{\{z \le \lambda\}} + \frac{(a\lambda - z)_+}{(a-1)\lambda} \mathbb{I}_{\{z > \lambda\}} \right) \mathrm{d}z, \quad a > 2.$$

When $p_\lambda$ is SCAD, we have

$$p_\lambda(\beta_j) = \lambda \int_0^{|\beta_j|} \left( 1 - \frac{z}{\lambda b} \right)_+ \mathrm{d}z, \quad b > 0.$$

## B  TIME COMPLEXITY

We compare the time complexity of solving the original problem (1) to that of solving the sketched problem (2) with Iterative SRO. We employ Proximal Gradient Descent (PGD) or Gradient Descent (GD) in our analysis, which are widely used in the machine learning and optimization literature. If $\mathbf{P}$ is a Gaussian subspace embedding in Definition 2.2, it takes $\mathcal{O}(\tilde{n}nd)$ operations to compute the sketched matrix $\tilde{\mathbf{X}} = \mathbf{PX}$ and then form the sketched problem (2). Let $C(\tilde{n}, d)$ be the time complexity of solving the sketched problem (2), and suppose iterative sketching is performed for $N$ iterations, then the overall time complexity of Iterative SRO in Algorithm 1 is $\mathcal{O}\left(\tilde{n}nd + NC(\tilde{n}, d)\right)$. If $\mathbf{P}$ is a sparse subspace embedding in Definition 2.3, then it only takes

$\mathcal{O}\left(\mathsf{nnz}(\mathbf{X})\right)$ operations to compute the sketched matrix $\tilde{\mathbf{X}}$. In this case, the overall time complexity of Iterative SRO is $\mathcal{O}\left(\mathsf{nnz}(\mathbf{X}) + NC(\tilde{n}, d)\right)$. Suppose PGD, such as that analyzed in (Bolte et al., 2014), or GD, is used to solve problem (1) and (2) with maximum number of iterations being $M$. Then $C(\tilde{n}, d) = \mathcal{O}\left(M\tilde{n}d\right)$. If a sparse subspace embedding is used for sketching, then the overall time complexity of Iterative SRO is $\mathcal{O}\left(\mathsf{nnz}(\mathbf{X}) + NM\tilde{n}d\right)$. In contrast, because IHS (Pilanci & Wainwright, 2016) needs to sample an independent sketch matrix at each iteration, the time complexity of IHS using the fast Johnson-Lindenstrauss sketches (that is, the fast Hadamard transform) is $\mathcal{O}\left(Nnd\log\tilde{n} + NM\tilde{n}d\right)$ which is higher than that of Iterative SRO with sparse subspace embedding. Noting that $N \leq \log n$ (Pilanci & Wainwright, 2016) and in many practical cases $N$ is bounded by a constant, and $\tilde{n} \ll n$, the complexity of Iterative SRO, $\mathcal{O}\left(\mathsf{nnz}(\mathbf{X}) + NM\tilde{n}d\right)$, is much lower than that of solving the original problem (1) with the complexity of $\mathcal{O}(Mnd)$.

## C COMPLETE EXPERIMENTAL RESULTS

We demonstrate complete experimental results of SRO and Iterative SRO in this section for three instances of the general optimization problem (1), which include ridge regression where $h(\boldsymbol{\beta}) = \|\boldsymbol{\beta}\|_2^2$, Generalized Lasso where $h(\boldsymbol{\beta}) = \sum_{i=1}^{d-1} |\boldsymbol{\beta}_i - \boldsymbol{\beta}_{i+1}|$, and Lasso where $h(\boldsymbol{\beta}) = \|\boldsymbol{\beta}\|_1$. We further show the application of SRO in subspace clustering.

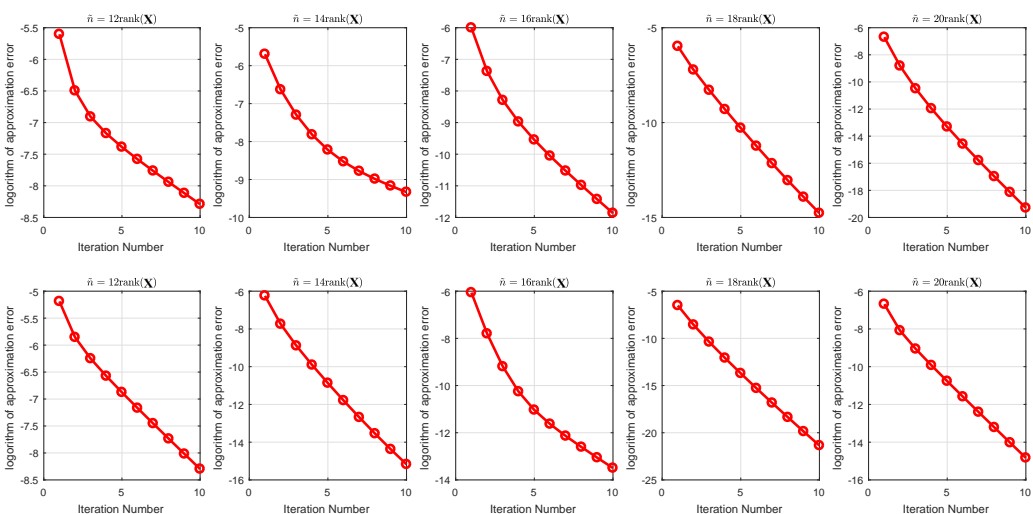

Figure 1: Approximation Error of Iterative SRO with respect to different sketch size $\tilde{n}$ for ridge regression. The first row corresponds to sparse subspace embedding defined in Definition 2.3 and the second row is produced by Gaussian subspace embedding defined in Definition 2.2.

### C.1 RIDGE REGRESSION OF LARGER SCALE

We employ Iterative SRO to approximate solution to Ridge Regression in this subsection, whose optimization problem is $\boldsymbol{\beta}^* = \arg\min_{\boldsymbol{\beta} \in \mathbb{R}^d} \frac{1}{2}\|\mathbf{y} - \mathbf{X}\boldsymbol{\beta}\|_2^2 + \lambda\|\boldsymbol{\beta}\|_2$. We assume a linear model $\bar{\mathbf{y}} = \bar{\mathbf{X}}\bar{\boldsymbol{\beta}} + \mathbf{w}$ where $\mathbf{X} = \bar{\mathbf{X}}/\sqrt{n}$, $\mathbf{y} = \bar{\mathbf{y}}/\sqrt{n}$, $\mathbf{w} \sim \mathcal{N}(\mathbf{0}, \mathbf{I}_n)$ is the Gaussian noise with unit variance. The unknown regression vector $\bar{\boldsymbol{\beta}}$ is sampled according to $\mathcal{N}(\mathbf{0}, \mathbf{I}_d)$. We randomly sample $\bar{\mathbf{X}} \in \mathbb{R}^{n \times d}$ of rank $r = \frac{n}{100}$ with $n = 5000$ and $d = 10000$. Let $\bar{\mathbf{X}} = \mathbf{U}\boldsymbol{\Sigma}\mathbf{V}^\top$ be the Singular Value Decomposition of $\bar{\mathbf{X}}$ where $\boldsymbol{\Sigma} \in \mathbb{R}^{r \times r}$ is a diagonal matrix whose diagonal elements are the singular values of $\bar{\mathbf{X}}$. $\mathbf{U} \in \mathbb{R}^{n \times r}$ is sampled from the uniform distribution over the Stiefel manifold $\mathbb{V}_r(\mathbb{R}^n)$, $\mathbf{V} \in \mathbb{R}^{d \times r}$ is sampled from the uniform distribution over the Stiefel manifold $\mathbb{V}_r(\mathbb{R}^d)$, the diagonal elements of $\boldsymbol{\Sigma}$ are i.i.d. standard Gaussian samples. We set $\lambda = \sqrt{\log d/n}$. Figure 1 illustrates the logarithm of approximation error $\frac{\|\boldsymbol{\beta}^{(i)} - \boldsymbol{\beta}^*\|_{\mathbf{x}}^2}{n}$ with respect to the iteration number $i$ of

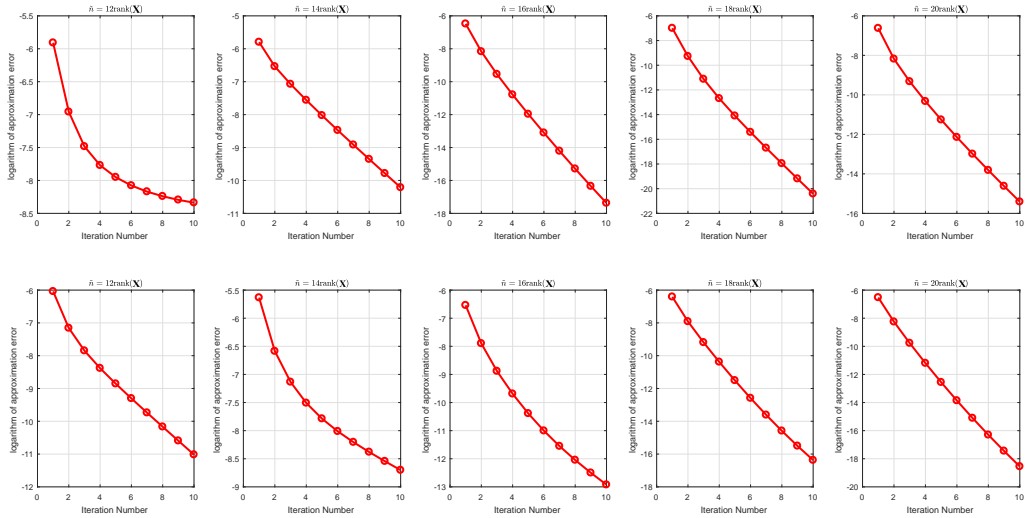

Figure 2: Logarithm of approximation error of Iterative SRO with respect to different sketch size $\tilde{n}$ for ridge regression. The first row corresponds to sparse subspace embedding and the second row is produced by Gaussian subspace embedding.

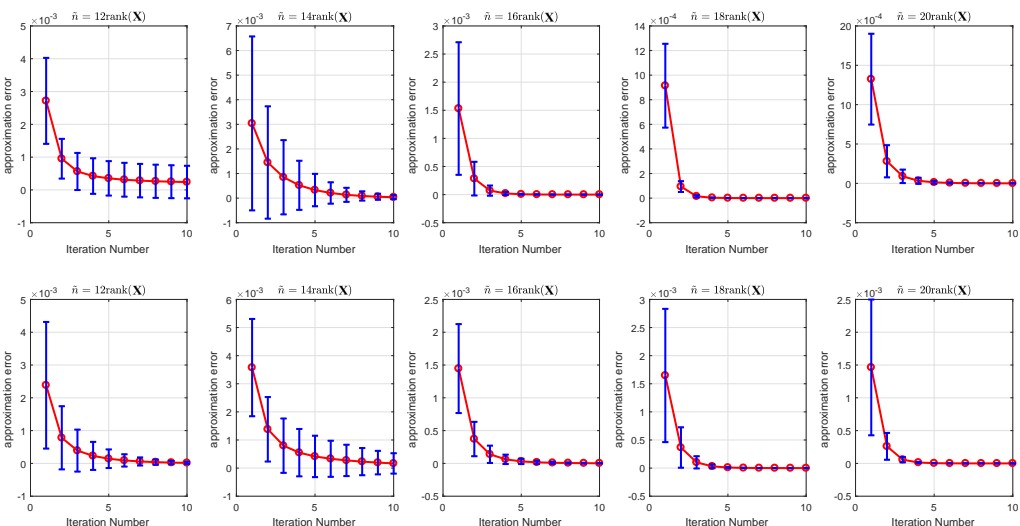

Figure 3: Approximation error of Iterative SRO with respect to different sketch size $\tilde{n}$ for ridge regression. The first row corresponds to sparse subspace embedding and the second row is produced by Gaussian subspace embedding.

Iterative SRO for different choices of sketch size $\tilde{n}$, and the maximum iteration number of Iterative SRO is set to $N = 10$. We let $\tilde{n} = \bar{\gamma}\mathrm{rank}(\mathbf{X})$ where $\bar{\gamma}$ ranges over $\{12, 14, 16, 18, 20\}$, and sample $\bar{\mathbf{X}}, \bar{\mathbf{y}}$ and $\mathbf{w}$ 100 times for each $\bar{\gamma}$. The average approximation errors are illustrated in Figure 1. It can be observed from Figure 1 that the convergence rate of approximation error drops geometrically, or its logarithm drops linearly, evidencing our Theorem 3.1 for Iterative SRO. Moreover, as suggested by Theorem D.2, larger $\tilde{n}$ leads to smaller approximation error.

We present more experimental results for ridge regression with larger-scale data where $n = 10000$ and $d = 100000$, and other settings remain the same. We let $\tilde{n} = \bar{\gamma}\mathrm{rank}(\bar{\mathbf{X}})$ where $\bar{\gamma}$ ranges over

$\{12, 14, 16, 18, 20\}$, and sample $\bar{\mathbf{X}}, \bar{\boldsymbol{\beta}}$ and $\mathbf{w}$ 100 times for each $\bar{\gamma}$. Figure 2 illustrates the logarithm of approximation error, which is $\log \frac{\left\|\boldsymbol{\beta}^{(i)} - \boldsymbol{\beta}^*\right\|_{\mathbf{x}}^2}{n}$, with respect to the iteration number $i$ of Iterative SRO for different choices of sketch size $\tilde{n}$, and the maximum iteration number of Iterative SRO is set to $N = 10$. It can be observed that the logarithm of approximation error drops linearly with respect to the iteration number in most cases, evidencing our theory that the approximation error of Iterative SRO drops geometrically in the iteration number.

Figure 3 illustrates the approximation error of Iterative SRO in red curve for ridge regression, which is $\frac{\left\|\boldsymbol{\beta}^{(i)} - \boldsymbol{\beta}^*\right\|_{\mathbf{x}}^2}{n}$. The blue bar represents standard deviation caused by the random data sampling and the random sketching. A single projection matrix $\mathbf{P}$ is sampled for each sampled data, and this projection matrix is used throughout all the iterations of Iterative SRO, in contrast with Iterative Hessian Sketch (IHS) (Pilanci & Wainwright, 2016) which samples a projection matrix matrix for each iteration of the iterative sketch procedure.

## C.2 GENERALIZED LASSO

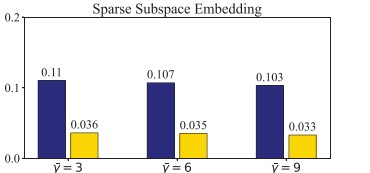
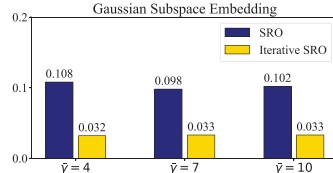

Figure 4: Approximation error of Iterative SRO vs. SRO for GLasso with respect to different sketch size $\tilde{n}$. Iterative SRO and SRO are equipped with either sparse subspace embedding (left) or Gaussian Subspace Embedding (right).

In this subsection, we add more details to Section 7.1 of the paper for Generalized Lasso (GLasso) (Tibshirani & Taylor, 2011). The optimization problem of GLasso studied here is $\boldsymbol{\beta}^* = \arg\min_{\boldsymbol{\beta} \in \mathbb{R}^d} \frac{1}{2} \|\mathbf{y} - \mathbf{X}\boldsymbol{\beta}\|_2^2 + \lambda \sum_{i=1}^{d-1} |\boldsymbol{\beta}_i - \boldsymbol{\beta}_{i+1}|$, which is solved by Fast Iterative Shrinkage-Thresholding Algorithm (FISTA) (Beck & Teboulle, 2009). We construct $\mathbf{D} \in \mathbb{R}^{(d-1) \times d}$ by setting $\mathbf{D}_{i,i} = -1$, $\mathbf{D}_{i,i+1} = 1$ for all $i \in [d-1]$, then $\sum_{i=1}^{d-1} |\boldsymbol{\beta}_i - \boldsymbol{\beta}_{i+1}| = \|\mathbf{D}\boldsymbol{\beta}\|_1$. Let $\mathbf{D}^{\text{ext}} = \begin{bmatrix} \mathbf{D} \\ 0, \dots, 1 \end{bmatrix}$ with $\mathbf{D}_{dd}^{\text{ext}} = 1$. Denote $\mathbf{u} = \mathbf{D}^{\text{ext}} \boldsymbol{\beta}$, then $\boldsymbol{\beta} = (\mathbf{D}^{\text{ext}})^{-1} \boldsymbol{\beta}$ because $\mathbf{D}^{\text{ext}}$ is nonsingular. The instance of GLasso considered above is then rewritten as $\boldsymbol{\beta}^* = \arg\min_{\boldsymbol{\beta} \in \mathbb{R}^d} \frac{1}{2n} \left\| \mathbf{y} - \mathbf{X} (\mathbf{D}^{\text{ext}})^{-1} \mathbf{u} \right\|_2^2 + \lambda \sum_{i=1}^{d-1} |\mathbf{u}_i|$ which can be solved by FISTA.

## C.3 SIGNAL RECOVERY BY LASSO

We present experimental results for signal recovery/approximation by Lasso in this subsection. In this experiment we assume a linear model $\bar{\mathbf{y}} = \bar{\mathbf{X}}\bar{\boldsymbol{\beta}} + \mathbf{w}$ where $\mathbf{X} = \bar{\mathbf{X}}/\sqrt{n}$, $\mathbf{y} = \bar{\mathbf{y}}/\sqrt{n}$, $\mathbf{w} \sim \mathcal{N}(\mathbf{0}, \mathbf{I}_n)$ is the Gaussian noise with unit variance. The optimization problem of Lasso considered here is $\boldsymbol{\beta}^* = \arg\min_{\boldsymbol{\beta} \in \mathbb{R}^d} \frac{1}{2} \|\mathbf{y} - \mathbf{X}\boldsymbol{\beta}\|_2^2 + \lambda \|\boldsymbol{\beta}\|_1$. We set $\lambda = 0.1 \cdot \sqrt{\bar{s} \log d / n}$, sparsity $\bar{s} = \lfloor 3 \log d \rfloor$, and choose the unknown regression vector $\bar{\boldsymbol{\beta}}$ with its support uniformly sampled with entries $\pm \frac{1}{\sqrt{s}}$ with equal probability. We randomly sample $\bar{\mathbf{X}} \in \mathbb{R}^{n \times d}$ of rank not greater than $r = \frac{n}{100}$ with $n = 5000$ and $d = 100000$ using (48) so that $\bar{\mathbf{X}}$ is a low-rank matrix satisfying RIP. That is, $\mathbf{X} = \mathbf{U}\mathbf{U}^\top \boldsymbol{\Omega}$ with $\mathbf{U} \in \mathbb{R}^{n \times r}$ and $\boldsymbol{\Omega} \in \mathbb{R}^{n \times d}$. $\mathbf{U}$ is sampled from the uniform distribution over the Stiefel manifold $\mathbb{V}_r(\mathbb{R}^n)$, the elements of $\boldsymbol{\Omega}$ are i.i.d. Gaussian random variables with $\boldsymbol{\Omega}_{ij} \sim \mathcal{N}(0, 1/r)$ for $i \in [n], j \in [d]$. We let $\tilde{n} = \bar{\gamma}\text{rank}(\bar{\mathbf{X}})$ where $\bar{\gamma}$ ranges over $\{12, 14, 16, 18\}$, and sample $\bar{\mathbf{X}}, \bar{\boldsymbol{\beta}}$ and $\mathbf{w}$ 100 times for each $\gamma$. Figure 5 illustrates the approximation error of SRO and Iterative SRO, i.e. $\frac{\left\|\boldsymbol{\beta}^{(1)} - \boldsymbol{\beta}^*\right\|_{\mathbf{x}}^2}{n}$ and $\frac{\left\|\boldsymbol{\beta}^{(N)} - \boldsymbol{\beta}^*\right\|_{\mathbf{x}}^2}{n}$ respectively, for different choices of sketch

size $\tilde{n}$ with different $\bar{\gamma}$. It can be observed that Iterative SRO significantly and constantly reduces approximation error of SRO.

Table 2 shows the error of approximation to the true unknown regression vector $\bar{\boldsymbol{\beta}}$ by SRO, Iterative SRO and the solution $\boldsymbol{\beta}^*$ to the original problem (1). The error of approximation to $\bar{\boldsymbol{\beta}}$ is the $\ell^2$-distance to $\bar{\boldsymbol{\beta}}$, for example, the error of approximation to $\bar{\boldsymbol{\beta}}$ for $\boldsymbol{\beta}^*$ is $\left\|\boldsymbol{\beta}^* - \bar{\boldsymbol{\beta}}\right\|_2$. Standard deviation is caused by random data sampling and random sketching. It can be seen that Iterative SRO approximates the true regression vector much better than SRO with the sketch size $\tilde{n}$ being a fraction of $n$, especially with Gaussian subspace embedding. More importantly, Iterative SRO and $\boldsymbol{\beta}^*$ have very close error of approximation to $\bar{\boldsymbol{\beta}}$, justifying our theoretical analysis. This is because that our theoretical prediction in Theorem 4.1 for the error of approximation to $\bar{\boldsymbol{\beta}}$ by Iterative SRO is $\Theta\left(\sqrt{\bar{s}\log d/n}\right)$, which is the same order as the error by $\boldsymbol{\beta}^*$.

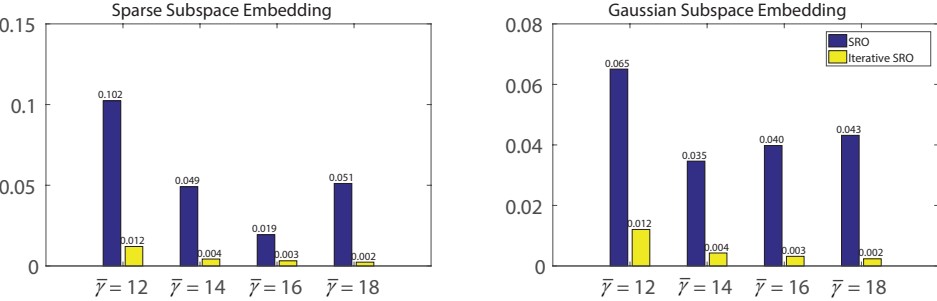

Figure 5: Approximation error of Iterative SRO and SRO for Lasso with respect to different sketch size $\tilde{n} = \gamma \mathrm{rank}(\mathbf{X})$. Iterative SRO and SRO are equipped with either sparse subspace embedding (left) or Gaussian Subspace Embedding (right).

Table 2: Approximation error to the true unknown regression vector $\bar{\boldsymbol{\beta}}$ by SRO, Iterative SRO and the solution $\boldsymbol{\beta}^*$ to the original problem (1) for Lasso.

|  | $\bar{\gamma}$ / Error | 12 | 14 | 16 | 18 |
|---|---|---|---|---|---|
| Sparse Subspace Embedding | SRO | $0.115 \pm 0.149$ | $0.063 \pm 0.023$ | $0.059 \pm 0.010$ | $0.025 \pm 0.054$ |
|  | Iteratie SRO | $0.010 \pm 0.017$ | $0.009 \pm 0.009$ | $0.004 \pm 0.002$ | $0.008 \pm 0.005$ |
|  | $\boldsymbol{\beta}^*$ | $0.008 \pm 0.002$ | $0.007 \pm 0.006$ | $0.003 \pm 0.001$ | $0.006 \pm 0.004$ |
| Gaussian Subspace Embedding | SRO | $0.067 \pm 0.092$ | $0.043 \pm 0.066$ | $0.044 \pm 0.038$ | $0.046 \pm 0.030$ |
|  | Iteratie SRO | $0.008 \pm 0.003$ | $0.008 \pm 0.007$ | $0.004 \pm 0.002$ | $0.003 \pm 0.004$ |
|  | $\boldsymbol{\beta}^*$ | $0.007 \pm 0.001$ | $0.006 \pm 0.003$ | $0.003 \pm 0.001$ | $0.002 \pm 0.002$ |

## C.4 APPLICATION IN LASSO SUBSPACE CLUSTERING

We demonstrate application of SRO in subspace clustering in this subsection. Given a data matrix $\bar{\mathbf{X}} \in \mathbb{R}^{n \times d}$ comprised of $d$ data points in $\mathbb{R}^n$ which lie in a union of subspaces in $\mathbb{R}^n$, classical subspace clustering methods using sparse codes, such as Noisy Sparse Subspace Clustering (Noisy SSC) (Wang & Xu, 2013), recovers the subspace structure by solving the Lasso problem

$$\boldsymbol{\beta}^i = \arg\min_{\boldsymbol{\beta} \in \mathbb{R}^d} \frac{1}{2n}\left\|\bar{\mathbf{X}}^i - \bar{\mathbf{X}}\boldsymbol{\beta}\right\|_2^2 + \lambda\|\boldsymbol{\beta}\|_1, \quad \boldsymbol{\beta}_i = 0, \tag{17}$$

for each $i \in [d]$. $\bar{\mathbf{X}}^i$ is the $i$-th column of $\bar{\mathbf{X}}$, which is also the $i$-th data point, $\lambda > 0$ is the weight for the $\ell^1$ regularization. Under certain conditions on $\bar{\mathbf{X}}$ and the underlying subspaces, it is proved by (Soltanolkotabi & Candés, 2012; Wang & Xu, 2013) that nonzero elements of $\boldsymbol{\beta}^*$ correspond to data points lying in the same subspace as $\bar{\mathbf{X}}^i$, and in this case $\boldsymbol{\beta}^*$ is said to satisfy the Subspace Detection Property (SDP). It has been proved that SDP is crucial for subspace recovery in the subspace clustering literature. By solving (17) for all $i \in [d]$, we have a sparse code matrix $\boldsymbol{\beta} = [\boldsymbol{\beta}^1, \boldsymbol{\beta}^2, \ldots, \boldsymbol{\beta}^d] \in \mathbb{R}^{d \times d}$, and a sparse similarity matrix $\mathbf{W}$ is constructed by $\mathbf{W} = \frac{|\mathbf{X}| + |\mathbf{X}^\top|}{2}$. Spectral clustering is performed on $\mathbf{W}$ to produce the final clustering result of Noisy SSC. Two measures are used to evaluate the performance of different clustering methods, i.e. the Accuracy (AC) and the Normalized Mutual Information (NMI) (Zheng et al., 2004). In this experiment, we employ

SRO to solve the sketched version of problem (17) with $\tilde{n} = \frac{n}{15}$. Note that SRO is equivalent to Iterative SRO with $N = 1$, and we do not incur more iterations by Iterative SRO since SRO produces satisfactory results. Figure 6 and Figure 7 illustrate the accuracy (left) and NMI (right) of sketched Noisy SSC by SRO with respect to various choices of the regularization weight $\lambda$ on the Extended Yale-B Dataset. The Extended Yale-B Dataset contains face images for 38 subjects with about 64 frontal face images of size $32 \times 32$ taken under different illuminations for each subject, and $\bar{\mathbf{X}}$ is of size $1024 \times 2414$. SRO-GSE stands for SRO with Gaussian subspace embedding, and SRO-SSE stands for SRO with sparse subspace embedding. We compare SRO-GSE and SRO-SSE to K-means (KM), Spectral Clustering (SC), Sparse Manifold Clustering and Embedding (SMCE) (Elhamifar & Vidal, 2011) and SSC by Orthogonal Matching Pursuit (SSC-OMP) (Dyer et al., 2013). It can be observed that SRO-GSE and SRO-SSE outperform other competing clustering methods by a notable margin, and they perform even better than the original Noisy SSC for most values of $\lambda$, due to the fact that sketching potentially reduces the adverse effect of noise in the original data.

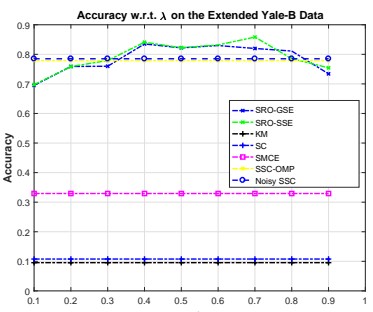 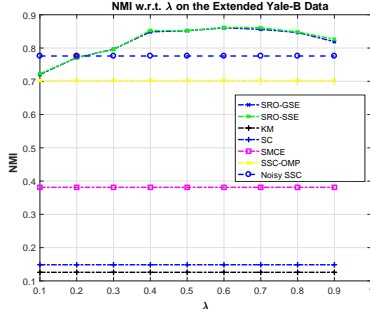

Figure 6: Accuracy with respect to different values of $\lambda$ on the Extended Yale-B Dataset

Figure 7: NMI with respect to different values of $\lambda$ on the Extended Yale-B Dataset

## C.5 SKETCHING FOR SPARSE NONCONVEX LEARNING WITH CAPPED-$\ell^1$ REGULARIZATION

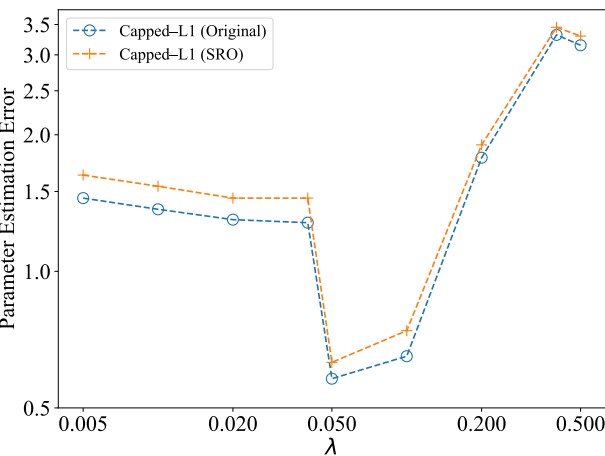

Figure 8: Illustration of the parameter estimation errors of SRO for sparse nonconvex learning with capped-$\ell^1$ regularization.

We study the performance of SRO for sparse nonconvex learning with capped-$\ell^1$ regularization in this subsection. The optimization problem considered here is $\boldsymbol{\beta}^* = \arg\min_{\boldsymbol{\beta} \in \mathbb{R}^d} \frac{1}{2}\|\mathbf{y} - \mathbf{X}\boldsymbol{\beta}\|_2^2 + \lambda \sum_{i=1}^{d-1} \min\{|\boldsymbol{\beta}_i|, \alpha\}$, which is solved by the proximal gradient descent. $\bar{\mathbf{X}} = \sqrt{n}\mathbf{X} \in \mathbb{R}^{n \times d}$ has i.i.d. standard Gaussian entries with $n = 10000$ and $d = 40000$. Following the setup in (Zhang, 2010b), the true parameter vector $\bar{\boldsymbol{\beta}}$ is generated with $\bar{s}$ nonzero elements uniformly distributed

from $[10, 10]$. The response vector $\bar{\mathbf{y}}$ is generated by $\bar{\mathbf{y}} = \bar{\mathbf{X}}\bar{\boldsymbol{\beta}} + \boldsymbol{\varepsilon}$ where $\boldsymbol{\varepsilon}$ is a noise vector of i.i.d. Gaussian elements with variance $\sigma^2 = 1$, and $\bar{\mathbf{y}} = \sqrt{n}\mathbf{y}$. We set $\alpha = 10\lambda$, the sketch size $\tilde{n} = n/5$, and repeat the experiment for 100 times. The $\ell^2$-norm parameter estimation errors for different values of $\lambda$ are illustrated in Figure 8.

# D    PROOFS

We present proofs of theoretical results of the original paper in this section.

## D.1    PROOFS FOR SECTION 5 AND SECTION 3

Before presenting the proof of Theorem D.2, the following lemma is introduced which shows that subspace embedding approximately preserves inner product with high probability.

**Lemma D.1.** Suppose $\mathbf{P}$ is a $(1 \pm \varepsilon)$ $\ell^2$-subspace embedding for $\mathbf{X}$, and let $\mathcal{C}(\mathbf{X})$ denote the column space of $\mathbf{X}$. Then with probability $1 - \delta$, for any two vectors $\mathbf{u} \in \mathcal{C}(\mathbf{X})$, $\mathbf{v} \in \mathcal{C}(\mathbf{X})$,

$$\left| \mathbf{u}^\top \mathbf{P}^\top \mathbf{P} \mathbf{v} - \mathbf{u}^\top \mathbf{v} \right| \leq \|\mathbf{u}\|_2 \|\mathbf{v}\|_2 \varepsilon. \tag{18}$$

*Proof.* If $\mathbf{u} = \mathbf{0}$ or $\mathbf{v} = \mathbf{0}$, then (18) holds trivially. Otherwise, let $\mathbf{u}' = \frac{\mathbf{u}}{\|\mathbf{u}\|_2}$, $\mathbf{v}' = \frac{\mathbf{v}}{\|\mathbf{v}\|_2}$, and $\mathbf{u}', \mathbf{v}' \in \mathcal{C}(\mathbf{X})$. According to the definition of $(1 \pm \varepsilon)$ $\ell^2$-subspace embedding for $\mathbf{X}$, with probability $1 - \delta$,

$$(1 - \varepsilon)\|\mathbf{u}' + \mathbf{v}'\|_2^2 \leq \|\mathbf{P}(\mathbf{u}' + \mathbf{v}')\|_2^2 \leq (1 + \varepsilon)\|\mathbf{u}' + \mathbf{v}'\|_2^2, \tag{19}$$

$$(1 - \varepsilon)\|\mathbf{u}' - \mathbf{v}'\|_2^2 \leq \|\mathbf{P}(\mathbf{u}' - \mathbf{v}')\|_2^2 \leq (1 + \varepsilon)\|\mathbf{u}' - \mathbf{v}'\|_2^2. \tag{20}$$

Subtracting (20) from (19), we have

$$\left| \mathbf{u}'^\top \mathbf{P}^\top \mathbf{P} \mathbf{v}' - \mathbf{u}'^\top \mathbf{v}' \right| \leq \varepsilon, \tag{21}$$

and (18) holds by scaling (21) by $\|\mathbf{u}\|_2 \|\mathbf{v}\|_2$. $\square$

## D.2    GENERAL BOUND

**Theorem D.2.** Suppose $\tilde{\boldsymbol{\beta}}^*$ is any critical point of the objective function in (2), $\boldsymbol{\beta}^*$ is any critical point of the objective function in (1), and $0 < \varepsilon < \varepsilon_0 < 1$ where $\varepsilon_0$ is a small positive constant, $0 < \delta < 1$, $\mathbf{P}$ is drawn from an $(\varepsilon, \delta)$ oblivious $\ell^2$-subspace embedding over $\tilde{n} \times n$ matrices. Then with probability $1 - \delta$,

$$(1 - \varepsilon)\left\| \tilde{\boldsymbol{\beta}}^* - \boldsymbol{\beta}^* \right\|_{\mathbf{X}}^2 - \varepsilon \left\| \tilde{\boldsymbol{\beta}}^* - \boldsymbol{\beta}^* \right\|_{\mathbf{X}} \|\boldsymbol{\beta}^*\|_{\mathbf{X}} \leq \theta_{h_\lambda}(\boldsymbol{\beta}^*, \kappa)\left\| \tilde{\boldsymbol{\beta}}^* - \boldsymbol{\beta}^* \right\|_2 + \kappa\left\| \tilde{\boldsymbol{\beta}}^* - \boldsymbol{\beta}^* \right\|_2^2. \tag{22}$$

In particular, if $\mathbf{P}$ is a Gaussian subspace embedding, then $\tilde{n} = \mathcal{O}((r + \log \frac{1}{\delta})\varepsilon^{-2})$. If $\mathbf{P}$ is a sparse subspace embedding, then $\tilde{n} = \mathcal{O}(r^2/(\delta\varepsilon^2))$.

It can be verified that the degree of nonconvexity vanishes with $\kappa = 0$ when $h$ is convex. As a result, we have relative-error approximation bound for $\left\| \tilde{\boldsymbol{\beta}}^* - \boldsymbol{\beta}^* \right\|_{\mathbf{X}}$ shown in Theorem D.3 below.

**Theorem D.3.** If $h_\lambda$ is convex, then under the conditions of Theorem D.2, with probability $1 - \delta$,

$$\left\| \tilde{\boldsymbol{\beta}}^* - \boldsymbol{\beta}^* \right\|_{\mathbf{X}} \leq \frac{\varepsilon}{1 - \varepsilon}\|\boldsymbol{\beta}^*\|_{\mathbf{X}}. \tag{23}$$

Moreover, if the Frechet subdifferential of $h$ is $L_h$-smooth ($h$ can be nonconvex), then

$$\|\tilde{\boldsymbol{\beta}}^* - \boldsymbol{\beta}^*\|_{\mathbf{X}} \leq \frac{\varepsilon}{(1 - \varepsilon) - \frac{\lambda L_h}{\sigma_{\min}^2(\mathbf{X})}}\|\boldsymbol{\beta}^*\|_{\mathbf{X}}. \tag{24}$$

It can be observed from (23) that one-time SRO renders a (3) relative-error approximation (3). The Iterative SRO algorithm takes advantage of such relative-error approximation, and it applies SRO once at each iteration so as to approximate the residual of the last iteration with relative-error approximation so as to approximate the solution to the original problem geometrically. Detailed proof of the theoretical guarantee of Iterative SRO, Theorem 3.1, along with the proofs of Theorem D.2 and Theorem D.3, are presented in the next parts of this appendix.

**Proof of Theorem D.2.** By the optimality of $\tilde{\boldsymbol{\beta}}^*$, we have

$$\left\|\tilde{\mathbf{X}}^\top\tilde{\mathbf{X}}\tilde{\boldsymbol{\beta}}^* - \mathbf{y}^\top\mathbf{X} + \mathbf{v}\right\|_2 = 0 \tag{25}$$

for some $\mathbf{v} \in \tilde{\partial}h_\lambda(\tilde{\boldsymbol{\beta}}^*)$. In the sequel, we will also use $\tilde{\partial}h(\cdot)$ to indicate an element belonging to $\tilde{\partial}h_\lambda(\cdot)$ if no special note is made.

By the optimality of $\boldsymbol{\beta}^*$, we have

$$\left\|\mathbf{X}^\top\mathbf{X}\boldsymbol{\beta}^* - \mathbf{y}^\top\mathbf{X} + \mathbf{u}\right\|_2 = 0, \tag{26}$$

where $\mathbf{u} \in \tilde{\partial}h_\lambda(\boldsymbol{\beta}^*)$.

Define $\Delta := \tilde{\boldsymbol{\beta}}^* - \boldsymbol{\beta}^*$, $\tilde{\Delta} := \left(\tilde{\partial}h_\lambda(\tilde{\boldsymbol{\beta}}^*) - \tilde{\partial}h_\lambda(\boldsymbol{\beta}^*)\right)$. By (25) and (26), we have

$$\left\|\tilde{\mathbf{X}}^\top\tilde{\mathbf{X}}\tilde{\boldsymbol{\beta}}^* + \mathbf{v} - \mathbf{X}^\top\mathbf{X}\boldsymbol{\beta}^* - \mathbf{u}\right\|_2 = \left\|\tilde{\mathbf{X}}^\top\tilde{\mathbf{X}}\tilde{\boldsymbol{\beta}}^* - \mathbf{X}^\top\mathbf{X}\boldsymbol{\beta}^* + \tilde{\Delta}\right\|_2 = 0. \tag{27}$$

It follows from (27) that

$$\Delta^\top\left(\tilde{\mathbf{X}}^\top\tilde{\mathbf{X}}\tilde{\boldsymbol{\beta}}^* - \mathbf{X}^\top\mathbf{X}\boldsymbol{\beta}^* + \tilde{\Delta}\right) \leq \|\Delta\|_2\left\|\tilde{\mathbf{X}}^\top\tilde{\mathbf{X}}\tilde{\boldsymbol{\beta}}^* - \mathbf{X}^\top\mathbf{X}\boldsymbol{\beta}^* + \tilde{\Delta}\right\|_2 = 0. \tag{28}$$

On the other hand, the LHS of (28) can be written as

$$\Delta^\top\left(\tilde{\mathbf{X}}^\top\tilde{\mathbf{X}}\tilde{\boldsymbol{\beta}}^* - \mathbf{X}^\top\mathbf{X}\boldsymbol{\beta}^* + \tilde{\Delta}\right) = \Delta^\top\left(\tilde{\mathbf{X}}^\top\tilde{\mathbf{X}}\tilde{\boldsymbol{\beta}}^* - \tilde{\mathbf{X}}^\top\tilde{\mathbf{X}}\boldsymbol{\beta}^*\right) + \Delta^\top\left(\tilde{\mathbf{X}}^\top\tilde{\mathbf{X}}\boldsymbol{\beta}^* - \mathbf{X}^\top\mathbf{X}\boldsymbol{\beta}^*\right)$$
$$+ \Delta^\top\tilde{\Delta}. \tag{29}$$

By (28) and (29), we have

$$\Delta^\top\left(\tilde{\mathbf{X}}^\top\tilde{\mathbf{X}}\tilde{\boldsymbol{\beta}}^* - \tilde{\mathbf{X}}^\top\tilde{\mathbf{X}}\boldsymbol{\beta}^*\right) + \Delta^\top\left(\tilde{\mathbf{X}}^\top\tilde{\mathbf{X}}\boldsymbol{\beta}^* - \mathbf{X}^\top\mathbf{X}\boldsymbol{\beta}^*\right) \leq -\Delta^\top\tilde{\Delta} \tag{30}$$

Now we derive lower bounds for the two terms on the LHS of (30). First, we have

$$\Delta^\top\left(\tilde{\mathbf{X}}^\top\tilde{\mathbf{X}}\tilde{\boldsymbol{\beta}}^* - \tilde{\mathbf{X}}^\top\tilde{\mathbf{X}}\boldsymbol{\beta}^*\right) = \Delta^\top\tilde{\mathbf{X}}^\top\tilde{\mathbf{X}}\Delta = \left\|\tilde{\mathbf{X}}\Delta\right\|_2^2 \geq (1-\varepsilon)\|\mathbf{X}\Delta\|_2^2. \tag{31}$$

Moreover,

$$\Delta^\top\left(\tilde{\mathbf{X}}^\top\tilde{\mathbf{X}}\boldsymbol{\beta}^* - \mathbf{X}^\top\mathbf{X}\boldsymbol{\beta}^*\right) = \Delta^\top\left(\tilde{\mathbf{X}}^\top\tilde{\mathbf{X}} - \mathbf{X}^\top\mathbf{X}\right)\boldsymbol{\beta}^* \geq -\varepsilon\|\mathbf{X}\Delta\|_2\|\mathbf{X}\boldsymbol{\beta}^*\|_2. \tag{32}$$

Plugging (31) and (32) in (30), we have

$$(1-\varepsilon)\|\mathbf{X}\Delta\|_2^2 - \varepsilon\|\mathbf{X}\Delta\|_2\|\mathbf{X}\boldsymbol{\beta}^*\|_2 \leq -\Delta^\top\tilde{\Delta}. \tag{33}$$

Moreover, by the definition of degree of nonconvexity in (15),

$$-\Delta^\top\tilde{\Delta} \leq \theta_h(\boldsymbol{\beta}^*, \kappa)\left\|\tilde{\boldsymbol{\beta}}^* - \boldsymbol{\beta}^*\right\|_2 + \kappa\left\|\tilde{\boldsymbol{\beta}}^* - \boldsymbol{\beta}^*\right\|_2^2. \tag{34}$$

(22) then follows by (33) and (34).

$\square$

**Proof of Theorem D.3.** When $h_\lambda$ is convex, then its Frechet differential coincides with its subdifferential. As a result, for any $\mathbf{u} \in \tilde{\partial}h(\mathbf{s})$ and $\mathbf{v} \in \tilde{\partial}h(\mathbf{t})$, we have $(\mathbf{s} - \mathbf{t})^\top(\mathbf{u} - \mathbf{v}) \geq 0$, and it follows that

$$-(\mathbf{s} - \mathbf{t})^\top(\mathbf{u} - \mathbf{v}) \leq 0. \tag{35}$$

With $\kappa = 0$, (35) suggests that

$$\theta_{h_\lambda}(\mathbf{t}, 0) = \sup_{\mathbf{s} \in \mathbb{R}^d, \mathbf{s} \neq \mathbf{t}} \{-\frac{1}{\|\mathbf{s} - \mathbf{t}\|_2}(\mathbf{s} - \mathbf{t})^\top(\tilde{\partial}h_\lambda(\mathbf{s}) - \tilde{\partial}h_\lambda(\mathbf{t}))\} \leq 0, \tag{36}$$

By (22) in Theorem D.2 and (36), setting $\kappa = 0$, we have

$$(1 - \varepsilon)\left\|\tilde{\boldsymbol{\beta}}^* - \boldsymbol{\beta}^*\right\|_{\mathbf{X}}^2 - \varepsilon\left\|\tilde{\boldsymbol{\beta}}^* - \boldsymbol{\beta}^*\right\|_{\mathbf{X}}\|\boldsymbol{\beta}^*\|_{\mathbf{X}} \leq \theta_{h_\lambda}(\boldsymbol{\beta}^*, \kappa)\left\|\tilde{\boldsymbol{\beta}}^* - \boldsymbol{\beta}^*\right\|_2 \leq 0. \tag{37}$$

If $\left\|\tilde{\boldsymbol{\beta}}^* - \boldsymbol{\beta}^*\right\|_{\mathbf{X}} \neq 0$, it follows from (37) that

$$\left\|\tilde{\boldsymbol{\beta}}^* - \boldsymbol{\beta}^*\right\|_{\mathbf{X}} \leq \frac{\varepsilon}{1 - \varepsilon}\|\boldsymbol{\beta}^*\|_{\mathbf{X}}. \tag{38}$$

Because the Frechet subdifferential of $h$ is $L_h$-smooth, namely $\sup_{\mathbf{u} \in \tilde{\partial}h(\mathbf{x}), \mathbf{v} \in \tilde{\partial}h(\mathbf{y})}\|\mathbf{u} - \mathbf{v}\|_2 \leq L_h\|\mathbf{x} - \mathbf{y}\|_2$, it can be veried that $\theta_h(\mathbf{x}^*, \kappa) \leq 0$ with holds with $\kappa = L_h$. This is due to the fact that for any $\mathbf{u} \in \tilde{\partial}h(\mathbf{x})$ and any $\mathbf{v} \in \tilde{\partial}h(\mathbf{y})$,

$$-(\mathbf{s} - \mathbf{t})^\top(\mathbf{u} - \mathbf{v}) - L_h\|\mathbf{s} - \mathbf{t}\|_2^2 \leq \|\mathbf{s} - \mathbf{t}\|_2\|\mathbf{u} - \mathbf{v}\|_2 - L_h\|\mathbf{s} - \mathbf{t}\|_2^2$$
$$\leq L_h\|\mathbf{x} - \mathbf{y}\|_2^2 - L_h\|\mathbf{s} - \mathbf{t}\|_2^2 \leq 0. \tag{39}$$

Therefore,

$$\theta_h(\mathbf{t}, L_h) = \sup_{\mathbf{s} \in \mathbb{R}^d, \mathbf{s} \neq \mathbf{t}, \mathbf{u} \in \tilde{\partial}h(\mathbf{s}), \mathbf{v} \in \tilde{\partial}h(\mathbf{t})} \frac{-(\mathbf{s} - \mathbf{t})^\top(\mathbf{u} - \mathbf{v}) - L_h\|\mathbf{s} - \mathbf{t}\|_2^2}{\|\mathbf{s} - \mathbf{t}\|_2} \leq 0 \tag{40}$$

holds for arbitrary $\mathbf{t}$, and it follows that $\theta_h(\mathbf{x}^*, L_h) \leq 0$. Moreover, since $\mathbf{X}$ is nonsingular, $\|\tilde{\boldsymbol{\beta}}^* - \boldsymbol{\beta}^*\|_2^2 \leq \frac{\|\tilde{\boldsymbol{\beta}}^* - \boldsymbol{\beta}^*\|_{\mathbf{X}}^2}{\sigma_{\min}^2(\mathbf{X})}$. Plugging the above results in (22) of Theorem D.2 and setting $\kappa$ to $L_h$, we have

$$(1 - \varepsilon)\|\tilde{\boldsymbol{\beta}}^* - \boldsymbol{\beta}^*\|_{\mathbf{X}}^2 - \varepsilon\|\tilde{\boldsymbol{\beta}}^* - \boldsymbol{\beta}^*\|_{\mathbf{X}}\|\boldsymbol{\beta}^*\|_{\mathbf{X}} \leq \lambda\theta_h(\boldsymbol{\beta}^*, L_h)\|\tilde{\boldsymbol{\beta}}^* - \boldsymbol{\beta}^*\|_2 + L_h\|\tilde{\boldsymbol{\beta}}^* - \boldsymbol{\beta}^*\|_2^2$$
$$\leq \frac{L_h}{\sigma_{\min}^2(\mathbf{X})} \cdot \|\tilde{\boldsymbol{\beta}}^* - \boldsymbol{\beta}^*\|_{\mathbf{X}}^2, \tag{41}$$

and it follows from (41) that

$$\|\tilde{\boldsymbol{\beta}}^* - \boldsymbol{\beta}^*\|_{\mathbf{X}} \leq \frac{\varepsilon}{(1 - \varepsilon) - \frac{L_h}{\sigma_{\min}^2(\mathbf{X})}}\|\boldsymbol{\beta}^*\|_{\mathbf{X}}.$$

$\square$

**Proof of Theorem 3.1.** This proof mostly follows from the proof of our main Theorem D.2 and Theorem D.3. We first consider the case that $h_\lambda$ is convex.

Let $\tilde{\boldsymbol{\beta}}^* = \boldsymbol{\beta}^{(t)}$. Define $\Delta := \tilde{\boldsymbol{\beta}}^* - \boldsymbol{\beta}^*$, $\tilde{\Delta} := (\tilde{\partial}h_\lambda(\tilde{\boldsymbol{\beta}}^*) - \tilde{\partial}h_\lambda(\boldsymbol{\beta}^*))$. By repeating the proof of Theorem D.2 and Theorem D.3, with probability $1 - \delta$,

$$(1 - \varepsilon)\|\mathbf{X}\Delta\|_2^2 - \varepsilon\|\mathbf{X}\Delta\|_2\left\|\mathbf{X}(\boldsymbol{\beta}^* - \boldsymbol{\beta}^{(t-1)})\right\|_2 \leq -\Delta^\top\tilde{\Delta}. \tag{42}$$

It follows by the proof of Theorem D.3 that $\lambda\Delta^\top\tilde{\Delta} \leq 0$. As a result, it follows by (42) that

$$\left\|\tilde{\boldsymbol{\beta}}^* - \boldsymbol{\beta}^*\right\|_{\mathbf{X}} \leq \frac{\varepsilon}{1 - \varepsilon}\left\|\boldsymbol{\beta}^* - \boldsymbol{\beta}^{(t-1)}\right\|_{\mathbf{X}} \tag{43}$$

For a constant $0 < \rho < 1$, by choosing $\varepsilon \leq \frac{\rho}{\rho+1}$ in (43), we have

$$\left\|\boldsymbol{\beta}^{(t)} - \boldsymbol{\beta}^*\right\|_{\mathbf{X}} \leq \rho\left\|\boldsymbol{\beta}^* - \boldsymbol{\beta}^{(t-1)}\right\|_{\mathbf{X}} \tag{44}$$

for any $t \geq 1$. It follows that

$$\left\|\boldsymbol{\beta}^{(N)} - \boldsymbol{\beta}^*\right\|_{\mathbf{X}} \leq \rho^N\|\boldsymbol{\beta}^*\|_{\mathbf{X}} \tag{45}$$

with $\boldsymbol{\beta}^{(0)} = \mathbf{0}$.

A similar proof is applied for the case that $h_\lambda$ is $L_h$-smooth and $\mathbf{X}$ has full column rank with $\frac{L_h}{\sigma^2_{\min}(\mathbf{X})} < 1 - \varepsilon$. It can be verified that (42) still holds, and it follows from (34) in the proof of Theorem D.2, (39) and (40) in the proof of Theorem D.3 that

$$-\Delta^\top \tilde{\Delta} \le \theta_h(\boldsymbol{\beta}^*, L_h) \left\| \tilde{\boldsymbol{\beta}}^* - \boldsymbol{\beta}^* \right\|_2 + L_h \left\| \tilde{\boldsymbol{\beta}}^* - \boldsymbol{\beta}^* \right\|_2^2 \le \frac{L_h}{\sigma^2_{\min}(\mathbf{X})} \cdot \|\mathbf{X}\Delta\|_2^2. \tag{46}$$

It follows from (42) and (46) that

$$\left\| \tilde{\boldsymbol{\beta}}^* - \boldsymbol{\beta}^* \right\|_{\mathbf{X}} \le \frac{\varepsilon}{1 - \frac{L_h}{\sigma^2_{\min}(\mathbf{X})} - \varepsilon} \left\| \boldsymbol{\beta}^* - \boldsymbol{\beta}^{(t-1)} \right\|_{\mathbf{X}}. \tag{47}$$

For a constant $0 < \rho < 1$, by choosing $\varepsilon \le \frac{\rho(1 - L_h/\sigma^2_{\min}(\mathbf{X}))}{\rho+1}$ in (47), we still have $\left\| \boldsymbol{\beta}^{(t)} - \boldsymbol{\beta}^* \right\|_{\mathbf{X}} \le \rho \left\| \boldsymbol{\beta}^* - \boldsymbol{\beta}^{(t-1)} \right\|_{\mathbf{X}}$ for any $t \ge 1$. It follows that $\left\| \boldsymbol{\beta}^{(N)} - \boldsymbol{\beta}^* \right\|_{\mathbf{X}} \le \rho^N \|\boldsymbol{\beta}^*\|_{\mathbf{X}}$.

$\square$

### D.3 PROOFS FOR SECTION 4: SKETCHING FOR SPARSE CONVEX LEARNING

In Theorem 4.1, we showed that for a low-rank data matrix $\mathbf{X}$ with ran $r \ll n$, then the Iterative ROS described in Algorithm 1 applied on the sketched problem (2) achieves the parameter estimation error of the order $\mathcal{O}\left(\sqrt{\bar{s}\log d/n}\right)$ under Assumption 1. As explained in (Yang et al., 2016), Assumption 1 is weaker than the Restricted Isometry Property (RIP) in compressed sensing (Candes & Tao, 2005a). We show that **there exists low-rank data matrix X satisfying Assumption 1**. We first show by Theorem D.4 that there exists a low-rank data matrix $\mathbf{X}$ which satisfies $\text{RIP}(\delta, s)$ for $\delta \in (0,1)$ and $s \in \mathbb{N}$ with $\text{RIP}(\delta, s)$ defined in Theorem 4.4(b).

We define the necessary notations in the following text. Let $\mathbf{A}$ be a matrix and $\mathcal{S}$ be a set, we denote by $\mathbf{A}_\mathcal{S}$ the submatrix of $\mathbf{A}$ with columns indexed by $\mathcal{S}$. Similarly, for a vector $\mathbf{v}$, $\mathbf{v}_\mathcal{S}$ is a vector formed by elements of $\mathbf{v}$ indexed by $\mathcal{S}$.

**Theorem D.4.** Let $\delta \in (0,1)$ and $s \in \mathbb{N}$. There exists a data matrix $\mathbf{X} \in \mathbb{R}^{n \times d}$ with rank not greater than $c_1 n$ for $c_1 \in (0,1)$ such that $\text{RIP}(\delta, s)$ holds with probability at least $1 - 2d^s \left(1 + \frac{8}{\delta}\right)^s \exp\left(-nc_{1,2}(\delta)\right) - 2d^s \left(1 + \frac{8}{\delta}\right)^s \exp\left(-n^{\alpha_0}\right)$, where $\alpha_0 \in (0,1)$ is a arbitrary positive constant, $c_{1,2}(\delta)$ is a positive constant depending on $c_1$ and $\delta$. Here $n \ge (\Theta(s \log d))^{1/\alpha_0}$. $\text{RIP}(\delta, s)$ means that for all $\mathbf{v} \in \mathbb{R}^d, \|\mathbf{v}\|_0 \le s$,

$$(1 - \delta)\|\mathbf{v}\|_2^2 \le \mathbf{v}\mathbf{X}^\top \mathbf{X}\mathbf{v} \le (1 + \delta)\|\mathbf{v}\|_2^2.$$

*Proof.* We construct the data matrix by

$$\mathbf{X} = \mathbf{U}\mathbf{U}^\top \Omega, \quad \mathbf{U} \in \mathbb{R}^{n \times r}, \Omega \in \mathbb{R}^{n \times d}, r = c_1 n, \tag{48}$$

where $\mathbf{U}$ is an orthogonal matrix and $\mathbf{U}^\top \mathbf{U} = \mathbf{I}_r$, all the elements of $\Omega$ are i.i.d. Gaussian random variables with $\Omega_{ij} \sim \mathcal{N}(0, 1/r)$ for $i \in [n], j \in [d]$. It is clear that the rank of $\mathbf{X}$ is bounded by $c_1 n$.

We define the set

$$\mathcal{F} := \bigcup_{\mathbf{S} \subseteq [d], |\mathbf{S}| = s} \mathcal{F}_{\mathbf{S}}, \quad \mathcal{F}_{\mathbf{S}} := \left\{ \mathbf{u} \in \mathbb{R}^d \mid \|\mathbf{u}\|_2 = 1, \text{supp}(\mathbf{u}) \subseteq \mathbf{S} \right\}.$$

Given a set $\mathbf{S} \subseteq [d], |\mathbf{S}| = s$, let $\mathbf{v} \in \mathcal{F}_{\mathbf{S}}$, we define functions

$$F(\mathbf{v}) := \frac{1}{c_1 \|\mathbf{A}\mathbf{v}\|_2^2} \left\| \mathbf{U}\mathbf{U}^\top \mathbf{A}\mathbf{v} \right\|_2^2, \quad g(\mathbf{x}) := \frac{1}{c_1} \left\| \mathbf{U}\mathbf{U}^\top \mathbf{x} \right\|_2^2.$$

where $\mathbf{A} = \sqrt{r}\Omega$ and the elements of $\mathbf{A}$ are i.i.d. standard Gaussian random variables. It is clear that $F(\mathbf{v}) = g(\mathbf{A}\mathbf{v}/\|\mathbf{A}\mathbf{v}\|_2)$, and $g$ is a $2/c_1$-Lipschitz function. Moreover,

$$\mathbb{E}_{\mathbf{x} \sim \text{Unif}(\mathbb{S}^{n-1})}[g(\mathbf{x})] = \frac{1}{c_1} \mathbb{E}_{\mathbf{x} \sim \text{Unif}(\mathbb{S}^{n-1})} \left[ \mathbf{U}\mathbf{U}^\top \mathbf{x}\mathbf{x}^\top \mathbf{U}\mathbf{U}^\top \right]$$

$$= \frac{c_1 n}{c_1 n} = 1.$$

Let $\mathbf{x} = \mathbf{A}\mathbf{v}/\|\mathbf{A}\mathbf{v}\|_2$, then $\mathbf{x} \sim \text{Unif}(\mathbb{S}^{n-1})$, and $F(\mathbf{v}) = g(\mathbf{x})$. It follows by Lemma D.9 that

$$\Pr\left[|g(\mathbf{x}) - \mathbb{E}\left[g(\mathbf{x})\right]| > t\right] \le 2\exp\left(-nc_1^2 t^2/8\right), \tag{49}$$

It follows by (49) that for a given $\mathbf{v} \in \mathcal{F}_\mathbf{S}$, $\Pr\left[|F(\mathbf{v}) - 1| \le t\right] \ge 1 - 2\exp\left(-nc_1^2 t^2/8\right)$.

On the other hand, $\mathbf{v}^\top \mathbf{X}^\top \mathbf{X}\mathbf{v} = 1/r \cdot \left\|\mathbf{U}\mathbf{U}^\top \mathbf{A}\mathbf{v}\right\|_2^2$ and $\mathbf{v}^\top \mathbf{X}^\top \mathbf{X}\mathbf{v} = \|\mathbf{A}\mathbf{v}\|_2^2/n \cdot F(\mathbf{v})$. Noting that $\|\mathbf{A}\mathbf{v}\|_2^2$ is $\chi^2$, it follows by standard concentration on $\chi^2$ that $\left|\|\mathbf{A}\mathbf{v}\|_2^2/n - 1\right| \le \Theta(\delta^2)$ when $n \ge n_0(\delta, \alpha_0)$, where $n_0(\delta, \alpha_0)$ is a positive constant depending on $\delta$ and $\alpha_0$. Let $t = \Theta(\delta^2)$ in (49) w.h.p. (with a high probability), then $|F(\mathbf{v}) - 1| \le \Theta(\delta^2)$ w.h.p. By choosing proper constants in $\Theta(\delta^2)$ and choosing the constant $n_0(\delta, \alpha_0)$ accordingly, we have $\left|\|\mathbf{A}\mathbf{v}\|_2^2/n \cdot F(\mathbf{v}) - 1\right| \le \delta^2/4$ w.h.p. In particular, for a given $\mathbf{v} \in \mathcal{F}_\mathbf{S}$, with probability at least $1 - 2\exp\left(-nc_{1,2}(\delta)\right) - 2\exp\left(-n^{\alpha_0}\right)$, we have $\left|\mathbf{v}^\top \mathbf{X}^\top \mathbf{X}\mathbf{v} - 1\right| \le \delta^2/4$, and $\left|\|\mathbf{X}\mathbf{v}\|_2 - 1\right| \le \delta/2$.

It follows from (Vershynin, 2012, Lemma 5.2) that there exists an $\delta/4$-net $N_{\delta/4}(\mathcal{F}_\mathbf{S}, \|\cdot\|_2) \subseteq \mathcal{F}_\mathbf{S}$ of $\mathcal{F}_\mathbf{S}$ such that $\left|N_{\delta/4}(\mathcal{F}_\mathbf{S}, \|\cdot\|_2)\right| \le \left(1 + \frac{8}{\delta}\right)^s$. Using the union bound, with probability at least $1 - 2\left(1 + \frac{8}{\delta}\right)^s \exp\left(-nc_{1,2}(\delta)\right) - 2\left(1 + \frac{8}{\delta}\right)^s \exp\left(-n^{\alpha_0}\right)$, the event that $\left|\|\mathbf{X}\mathbf{v}\|_2 - 1\right| \le \delta/2$ holds for all $\mathbf{v} \in N_{\delta/4}(\mathcal{F}_\mathbf{S}, \|\cdot\|_2)$ happens. Define this event as $\mathcal{A}_\mathbf{S}$, and the following argument is conditioned on $\mathcal{A}_\mathbf{S}$.

We define $A > 0$ as the smallest number such that $\|\mathbf{X}\mathbf{v}\|_2 \le 1 + A$ for all $\mathbf{v} \in \mathcal{F}_\mathbf{S}$. For any $\mathbf{v} \in \mathcal{F}_\mathbf{S}$, there exists $\mathbf{v}' \in N_{\delta/4}(\mathcal{F}_\mathbf{S}, \|\cdot\|_2)$ such that $\|\mathbf{v} - \mathbf{v}'\|_2 \le \delta/4$. As a result,

$$\|\mathbf{X}\mathbf{v}\|_2 \le \|\mathbf{X}\mathbf{v}'\|_2 + \|\mathbf{X}(\mathbf{v} - \mathbf{v}')\|_2 \le 1 + \frac{\delta}{2} + (1 + A)\frac{\delta}{4}.$$

It follows by the definition of $A$ that $A \le \delta/2 + (1 + A)\delta/4$, so $A \le \frac{3\delta/4}{1 - \delta/4} \le \delta$. We then have $\|\mathbf{X}\mathbf{v}\|_2 \le 1 + \delta$ for all $\mathbf{v} \in \mathcal{F}_\mathbf{S}$. On the other hand,

$$\|\mathbf{X}\mathbf{v}\|_2 \ge \|\mathbf{X}\mathbf{v}'\|_2 - \|\mathbf{X}(\mathbf{v} - \mathbf{v}')\|_2 \ge 1 - \frac{\delta}{2} - (1 + \delta)\frac{\delta}{4} \ge 1 - \delta, \quad \forall \mathbf{v} \in \mathcal{F}_\mathbf{S}.$$

As a result, conditioned on the event $\mathcal{A}_\mathbf{S}$ for a set $\mathbf{S} \subseteq [d], |\mathbf{S}| = s, \left|\|\mathbf{X}\mathbf{v}\|_2 - 1\right| \le \delta$ holds for all $\mathbf{v} \in \mathcal{F}_\mathbf{S}$. Because $|\mathcal{F}| \le d^s$, using the union bound, with probability specified in this theorem, $\left|\|\mathbf{X}\mathbf{v}\|_2 - 1\right| \le \delta$ holds for all $\mathbf{v} \in \mathbb{R}^d, \|\mathbf{v}\|_0 \le s$.

$\square$

**Low-Rank Matrix X Satisfying Assumption 1.** It is proved in Theorem D.4 that the low-rank data matrix $\mathbf{X}$ constructed by (48) satisfies RIP$(\delta, s)$ for $\delta \in (0, 1)$ and $s \in \mathbb{N}$. By making $\alpha_0 \to 1$, the lower bound for $n$ is $(\Theta(s \log d))^{1/\alpha_0}$ which can be close to the sample optimal $\Theta(s \log d)$. As indicated by (Yang et al., 2016), The condition (11) in Assumption 1 is weaker than RIP. To see this, (11) holds with $k^* = (s - \bar{s})/2$ if RIP$(\delta, s)$ holds with $s = 5\bar{s}$ and $\delta = 1/3$.

**An Alternative Construction of A Low-Rank Matrix Satisfying RIP$(\delta, s)$.** It is remarked that one can also construct the low-rank matrix $\mathbf{X} = \mathbf{U}\mathbf{A}$ where $\mathbf{U} \in \mathbb{R}^{n \times m}$ is sampled from the Stiefel manifold $V_m(\mathbb{R}^n)$ comprising all the $n \times m$ matrices of orthogonal columns with $n \ge m$, and all the elements of $\mathbf{A} \in \mathbb{R}^{m \times d}$ are i.i.d. Gaussian random variables with $\mathbf{A}_{ij} \sim \mathcal{N}(0, 1/m)$ for $i \in [m], j \in [d]$. Then rank$(\mathbf{X}) \le m$, and it follows by (Baraniuk et al., 2008, Theorem 5.2) that when $m \ge \Theta(s \log d)$, then $\mathbf{A}$ satisfies RIP$(\delta, s)$. Since $\mathbf{X}^\top \mathbf{X} = \mathbf{A}^\top \mathbf{A}$, it follows that w.h.p. $\mathbf{X}$ also satisfies RIP$(\delta, s)$. The latter construction can admit the optimal size $n \asymp \Theta(s \log d)$. On the other hand, rank$(\mathbf{X}) \to \infty$ as $d \to \infty$ in latter construction, while the construction in Theorem D.4 allows for arbitrarily specified rank of the constructed $\mathbf{X}$.

We need the following lemma for the proof of Theorem 4.1.

**Lemma D.5** ((Yang et al., 2016, Lemma 5)). For any $\mathbf{v} \in \mathbb{R}^d$ and any index set $\mathcal{S} \subseteq [d]$ with $|\mathcal{S}| = \bar{s}$. Let $\mathcal{J}$ be the set of indices of the largest $k^*$ elements of $\mathbf{v}_{\mathcal{S}^c}$ in absolute value and let

$\mathcal{I} = \mathcal{J} \bigcup \mathcal{S}$. Here $\bar{s}$ and $k^*$ are the same as those in Assumption 1. Assume that $\|\mathbf{v}_{\mathcal{S}^c}\|_1 \leq \gamma \|\mathbf{v}_{\mathcal{S}}\|_1$ for some $\gamma > 0$. Then we have $\|\mathbf{v}\|_2 \leq (1+\gamma)\|\mathbf{v}_{\mathcal{I}}\|_2$, and

$$\mathbf{v}\mathbf{X}^\top\mathbf{X}\mathbf{v} \geq \rho_{\mathcal{L},-}(\bar{s}+k^*) \cdot \left( \|\mathbf{v}_{\mathcal{I}}\|_2 - \gamma\sqrt{\bar{s}/k^*}\sqrt{\rho_{\mathcal{L},+}(k^*)/\rho_{\mathcal{L},-}(\bar{s}+2k^*)-1}\|\mathbf{v}_{\mathcal{S}}\|_2 \right) \|\mathbf{v}_{\mathcal{I}}\|_2. \tag{50}$$

**Proof of Theorem 4.1.** We consider the upper bound for the quadratic form $\left\langle \nabla\mathcal{L}(\tilde{\boldsymbol{\beta}}^*) - \nabla\mathcal{L}(\bar{\boldsymbol{\beta}}), \tilde{\boldsymbol{\beta}}^* - \bar{\boldsymbol{\beta}} \right\rangle = \Delta^\top\mathbf{X}^\top\mathbf{X}\Delta$, where $\tilde{\boldsymbol{\beta}}^* = \beta^{(N)}$ and $\Delta := \tilde{\boldsymbol{\beta}}^* - \bar{\boldsymbol{\beta}}$. We define $\mathcal{S} = \text{supp}(\bar{\boldsymbol{\beta}})$. Let $\mathbf{v}$ be a vector, we denote by $\mathbf{v}_{\mathcal{S}}$ the vector formed by elements of $\mathbf{v}$ in the set $\mathcal{S}$.

We have

$$\begin{aligned}
\Delta^\top\mathbf{X}^\top\mathbf{X}\Delta &= \left\langle \nabla\mathcal{L}(\tilde{\boldsymbol{\beta}}^*) - \nabla\mathcal{L}(\bar{\boldsymbol{\beta}}), \tilde{\boldsymbol{\beta}}^* - \bar{\boldsymbol{\beta}} \right\rangle \\
&\leq \left\langle \nabla\mathcal{L}(\tilde{\boldsymbol{\beta}}^*), \tilde{\boldsymbol{\beta}}^* - \bar{\boldsymbol{\beta}} \right\rangle + \left\| \nabla\mathcal{L}(\bar{\boldsymbol{\beta}}) \right\|_\infty \|\Delta\|_1 \\
&\overset{\text{\textcircled{1}}}{\leq} \left\langle \nabla\mathcal{L}(\tilde{\boldsymbol{\beta}}^*) - \nabla\mathcal{L}(\boldsymbol{\beta}^*), \tilde{\boldsymbol{\beta}}^* - \bar{\boldsymbol{\beta}} \right\rangle + \left\langle \nabla\mathcal{L}(\boldsymbol{\beta}^*), \tilde{\boldsymbol{\beta}}^* - \bar{\boldsymbol{\beta}} \right\rangle + 2\sigma\sqrt{\frac{\log d}{n}}\|\Delta\|_1 \\
&\leq \left\| \nabla\mathcal{L}(\tilde{\boldsymbol{\beta}}^*) - \nabla\mathcal{L}(\boldsymbol{\beta}^*) \right\|_\infty \|\Delta\|_1 - \lambda\|\Delta_{\mathcal{S}^c}\|_1 + \lambda\|\Delta_{\mathcal{S}}\|_1 + \frac{2}{c}\lambda\|\Delta\|_1 \\
&\overset{\text{\textcircled{2}}}{\leq} \lambda\mu\|\Delta\|_1 - \lambda\|\Delta_{\mathcal{S}^c}\|_1 + \lambda\|\Delta_{\mathcal{S}}\|_1 + \frac{2}{c}\lambda\|\Delta\|_1 \\
&= -\lambda\left(1 - \mu - \frac{2}{c}\right)\|\Delta_{\mathcal{S}^c}\|_1 + \lambda\left(1 + \mu + \frac{2}{c}\right)\|\Delta_{\mathcal{S}}\|_1. \tag{51}
\end{aligned}$$

Here ① follows by Lemma D.8, and $\rho^N\|\mathbf{X}\|_2\|\boldsymbol{\beta}^*\|_{\mathbf{X}} \leq \lambda\mu$ in ②.

It follows by (52) that $\|\Delta_{\mathcal{S}^c}\|_1 \leq ((1+\mu+2/c)/(1-\mu-2/c))\|\Delta_{\mathcal{S}}\|_1$.

We now apply Lemma D.5 with $\mathbf{v} = \Delta$ and $\gamma = (1+\mu+2/c)/(1-\mu-2/c)$. It follows by Lemma D.5 that $\|\Delta\|_2 \leq (1+\gamma)\|\Delta_{\mathcal{I}}\|_2$. Moreover, it follows by Assumption 1 that $\rho_{\mathcal{L},+}(k^*)/\rho_{\mathcal{L},-}(2k^*+\bar{s}) \leq 1 + 0.5k^*/\bar{s}$. Plugging this inequality in (50), we have

$$\begin{aligned}
\Delta\mathbf{X}^\top\mathbf{X}\Delta &\geq \rho_{\mathcal{L},-}(\bar{s}+k^*) \cdot \left( \|\Delta_{\mathcal{I}}\|_2 - \gamma\sqrt{\bar{s}/k^*}\sqrt{\rho_{\mathcal{L},+}(k^*)/\rho_{\mathcal{L},-}(\bar{s}+2k^*)-1}\|\Delta_{\mathcal{S}}\|_2 \right) \|\Delta_{\mathcal{I}}\|_2 \\
&\geq \rho_{\mathcal{L},-}(\bar{s}+k^*) \cdot \left( 1 - \gamma\sqrt{0.5} \right) \|\Delta_{\mathcal{I}}\|_2^2. \tag{52}
\end{aligned}$$

It follows from (51) that $\Delta^\top\mathbf{X}^\top\mathbf{X}\Delta \leq \lambda\left(1 + \mu + \frac{2}{c}\right)\|\Delta_{\mathcal{S}}\|_1 \leq \lambda\left(1 + \mu + \frac{2}{c}\right)\sqrt{\bar{s}}\|\Delta_{\mathcal{I}}\|_2$. It follows by this inequality and (52) that

$$\|\Delta\|_2 \leq (1+\gamma)\|\Delta_{\mathcal{I}}\|_2 \leq \frac{(1+\gamma)\left(1+\mu+\frac{2}{c}\right)\sqrt{\bar{s}}\lambda}{\rho_{\mathcal{L},-}(\bar{s}+k^*) \cdot \left(1 - \gamma\sqrt{0.5}\right)},$$

which proves (12).

$\square$

## D.4 Sketching for Sparse Nonconvex Learning

**Proof of Corollary 4.2.** We have $h_\lambda(\boldsymbol{\beta}) = \lambda\|\boldsymbol{\beta}\|_1 + Q_\lambda(\boldsymbol{\beta})$. For any $\mathbf{s} \in \mathbb{R}^d$, $\mathbf{t} \in \mathbb{R}^d$, and let $\mathbf{u} \in \tilde{\partial}h_\lambda(\mathbf{s})$, $\mathbf{v} \in \tilde{\partial}h_\lambda(\mathbf{t})$, then $\mathbf{u} = \xi_1 + \nabla Q_\lambda(\mathbf{s})$ and $\mathbf{v} = \xi_2 + \nabla Q_\lambda(\mathbf{t})$ with $\mathbf{x}_1 \in \partial(\lambda\|\mathbf{s}\|_1)$ and $\mathbf{x}_2 \in \partial(\lambda\|\mathbf{t}\|_1)$. We have

$$\begin{aligned}
-(\mathbf{s}-\mathbf{t})^\top(\mathbf{u}-\mathbf{v}) &= -(\mathbf{s}-\mathbf{t})^\top(\xi_1-\xi_2) - (\mathbf{s}-\mathbf{t})^\top(\nabla Q_\lambda(\mathbf{s}) - \nabla Q_\lambda(\mathbf{t})) \\
&\overset{\text{\textcircled{1}}}{\leq} -(\mathbf{s}-\mathbf{t})^\top(\nabla Q_\lambda(\mathbf{s}) - \nabla Q_\lambda(\mathbf{t}))
\end{aligned}$$

$$\overset{②}{\leq} \zeta_- \|\mathbf{s} - \mathbf{t}\|_2^2, \tag{53}$$

where ① follows from the convexity of $\lambda\|\cdot\|_1$, and ② follows from the regularity condition (b) in Assumption 2.

It follows by (53) that the degree of nonconvexity $\theta_{h_\lambda}(\mathbf{t}, \zeta_-) \leq 0$. Using the upper bound for $\tilde{n}$ in the given condition, plugging $\theta_{h_\lambda}(\mathbf{t}, \zeta_-) \leq 0$ in (22) and setting $\kappa$ to $\zeta_-$, we have

$$(1 - \varepsilon)\left\|\tilde{\boldsymbol{\beta}}^* - \boldsymbol{\beta}^*\right\|_{\mathbf{X}}^2 - \varepsilon\left\|\tilde{\boldsymbol{\beta}}^* - \boldsymbol{\beta}^*\right\|_{\mathbf{X}}\|\boldsymbol{\beta}^*\|_{\mathbf{X}} \leq \theta_h(\boldsymbol{\beta}^*, L_h)\left\|\tilde{\boldsymbol{\beta}}^* - \boldsymbol{\beta}^*\right\|_2 + \zeta_-\left\|\tilde{\boldsymbol{\beta}}^* - \boldsymbol{\beta}^*\right\|_2^2$$

$$\leq \zeta_-\left\|\tilde{\boldsymbol{\beta}}^* - \boldsymbol{\beta}^*\right\|_2^2. \tag{54}$$

By the definition of sparse eigenvalues in Definition 4.1, we have

$$\left\|\tilde{\boldsymbol{\beta}}^* - \boldsymbol{\beta}^*\right\|_{\mathbf{X}}^2 \geq \rho_{\mathcal{L},-}(s)\left\|\tilde{\boldsymbol{\beta}}^* - \boldsymbol{\beta}^*\right\|_2^2,$$

$$\left\|\tilde{\boldsymbol{\beta}}^* - \boldsymbol{\beta}^*\right\|_{\mathbf{X}}^2 \leq \rho_{\mathcal{L},+}(s)\left\|\tilde{\boldsymbol{\beta}}^* - \boldsymbol{\beta}^*\right\|_2^2.$$

Applying the above inequalities in (54), we have

$$(1 - \varepsilon)\rho_{\mathcal{L},-}(s)\left\|\tilde{\boldsymbol{\beta}}^* - \boldsymbol{\beta}^*\right\|_2^2 \leq \varepsilon\sqrt{\rho_{\mathcal{L},+}(s)}\left\|\tilde{\boldsymbol{\beta}}^* - \boldsymbol{\beta}^*\right\|_2\|\boldsymbol{\beta}^*\|_{\mathbf{X}} + \zeta_-\left\|\tilde{\boldsymbol{\beta}}^* - \boldsymbol{\beta}^*\right\|_2^2$$

which leads to (13). $\qquad\square$

**Definition D.1.** ($\varepsilon$-net) Let $(X, d)$ be a metric space and let $\varepsilon > 0$. A subset $N_\varepsilon(X, d)$ is called an $\varepsilon$-net of $X$ if for every point $x \in X$, there exists some point $y \in N_\varepsilon(X, d)$ such that $d(x, y) \leq \varepsilon$. The minimal cardinality of an $\varepsilon$-net of $X$, if finite, is called the covering number of $X$ at scale $\varepsilon$.

**Theorem D.6** (revised version of (Woodruff, 2014, Theorem 2.1) with explicit constant in the bound for $\tilde{n}$). Let $0 < \varepsilon, \delta < 1$ and $\mathbf{P} \in \mathbb{R}^{\tilde{n} \times n}$ be a Gaussian subspace embedding defined in Definition 2.2. Then if $\tilde{n} \geq 32\varepsilon^{-2}\log(2f_0/\delta)$, $\mathbf{P}$ is a Johnson–Lindenstrauss Transform, or JLT($\varepsilon, \delta, f_0$). That is, for any set $\mathcal{V}$ with $f_0$ elements and all $\mathbf{v}, \mathbf{v}' \in \mathcal{V}$, with probability at least $1 - \delta$, it holds that $|\langle \mathbf{Pv}, \mathbf{Pv}' \rangle - \langle \mathbf{v}, \mathbf{v}' \rangle| \leq \varepsilon\|\mathbf{v}\|_2\|\mathbf{v}'\|_2$.

**Lemma D.7.** Let $s \in [d]$, $s \geq 2$, $0 < \varepsilon, \delta < 1$, and $\mathbf{P} \in \mathbb{R}^{\tilde{n} \times n}$ be a Gaussian subspace embedding defined in Definition 2.2. If $\tilde{n} \geq c_0\varepsilon^{-2}\left(\log(2/\delta) + s\log d + s\log 5 + 1/d^{s-1}\right)$ where $c_0$ is a positive constant, then with probability at least $1 - \delta$, the following inequalities hold:

$$\rho_{\tilde{\mathcal{L}},+}(s) \leq \rho_{\mathcal{L},+}(s) + \varepsilon\sqrt{s}, \rho_{\tilde{\mathcal{L}},-}(s) \geq \rho_{\mathcal{L},-}(s) - \varepsilon\sqrt{s}, \tag{55}$$

$$\left\|\nabla\tilde{\mathcal{L}}(\bar{\boldsymbol{\beta}})\right\|_\infty \leq \left\|\nabla\mathcal{L}(\bar{\boldsymbol{\beta}})\right\|_\infty + \varepsilon\sqrt{s}\|\bar{\boldsymbol{\beta}}\|_2. \tag{56}$$

**Proof of Lemma D.7**. Define the set

$$\mathcal{F} := \bigcup_{\mathbf{S} \subseteq [d], |\mathbf{S}| = s} \mathcal{F}_{\mathbf{S}}, \quad \mathcal{F}_{\mathbf{S}} := \left\{\mathbf{u} \in \mathbb{R}^d \mid \text{supp}(\mathbf{u}) \subseteq \mathbf{S}\right\}.$$

Then $|\mathcal{F}| = \binom{d}{s} < d^s$. We now consider the subspace

$$\mathcal{U}_{\mathbf{S}} := \left\{\mathbf{u} = \mathbf{Xv} \mid \|\mathbf{u}\|_2 = 1, \mathbf{u} = \mathbf{Xv} \text{ for some } \mathbf{v} \in \mathcal{F}_{\mathbf{S}}\right\}, \quad \mathbf{S} \subseteq [d], |\mathcal{S}| = s.$$

It can be verified that the dimension of $\mathcal{U}_{\mathbf{S}}$ is not greater than $s$. Let $\gamma > 0$, it follows by (Vershynin, 2012, Lemma 5.2), there exists an $\gamma$-net $N_\gamma(\mathcal{U}_{\mathbf{S}}, \|\cdot\|_2) \subseteq \mathcal{U}_{\mathbf{S}}$ of $\mathcal{U}_{\mathbf{S}}$ such that $|N_\gamma(\mathcal{U}_{\mathbf{S}}, \|\cdot\|_2)| \leq \left(1 + \frac{2}{\gamma}\right)^s$.

Define the set

$$\mathcal{V}_s := \bigcup_{\mathbf{S} \subseteq [d], |\mathbf{S}| = s} N_{1/2}(\mathcal{U}_{\mathbf{S}}, \|\cdot\|_2) \bigcup \left(\bigcup_{i \in [d]} N_{1/2}(\mathbf{e}^{(i)}, \|\cdot\|_2)\right),$$

where $\mathbf{e}^{(i)}$ is the subspace spanned by $\{\mathbf{e}^i, \bar{\boldsymbol{\beta}}\}$ where $\{\mathbf{e}^i\}_{i=1}^d$ is the standard basis of $\mathbb{R}^d$. As a result of the above argument, $f_0 := |\mathcal{V}_s| \leq \binom{d}{s} \cdot 5^s + 25d$. It follows by standard calculation that $\log(2/\delta) + s \log d + s \log 5 + 1/d^{s-1} \geq \log(2f_0/\delta)$. It then follows by Theorem D.6 that when $\tilde{n} \geq c_0 \varepsilon^{-2} \left( \log(2/\delta) + s \log d + s \log 5 + 1/d^{s-1} \right) \geq c_0 \varepsilon^{-2} \log(2f_0/\delta)$ with $c_0 = 512$, $\mathbf{P}$ is a JLT$(\varepsilon/4, \delta, f_0)$. Therefore, with probability at least $1 - \delta$, $\mathbf{P}$ is a $(1 \pm \varepsilon/4)$ $\ell^2$-embedding for $\mathcal{V}_s$.

For any $\mathbf{u} \in \mathcal{U}_\mathbf{S}$, there exists a $\mathbf{u}' \in N_\gamma(\mathcal{U}_\mathbf{S}, \|\cdot\|_2)$ such that $\|\mathbf{u} - \mathbf{u}'\|_2 \leq \gamma$. We now use the simpler notation $\langle \mathbf{Pu}', \mathbf{Pv}' \rangle = (1 \pm \varepsilon) \langle \mathbf{u}', \mathbf{v}' \rangle$ to indicate $|\langle \mathbf{Pu}', \mathbf{Pv}' \rangle - \langle \mathbf{u}', \mathbf{v}' \rangle| \leq \varepsilon \|\mathbf{u}'\|_2 \|\mathbf{v}'\|_2$ for any two numbers $\mathbf{u}', \mathbf{v}'$ in the following text.

We follow the construction in the proof of (Woodruff, 2014, Theorem 2.3), that is, we build a $1/2$-net $N_{1/2}(\mathcal{U}_\mathbf{S}, \|\cdot\|_2) \subseteq \mathcal{U}_\mathbf{S}$ of $\mathcal{U}_\mathbf{S}$. Given any $\mathbf{S} \subseteq [d], |\mathcal{S}| = s$, we now show in the sequel that if $\mathbf{P}$ is a $(1 \pm \varepsilon/4)$ $\ell^2$-embedding for $N_{1/2}(\mathcal{U}_\mathbf{S}, \|\cdot\|_2)$, that is, $\langle \mathbf{Pu}', \mathbf{Pv}' \rangle = (1 \pm \varepsilon/4) \langle \mathbf{u}', \mathbf{v}' \rangle$ for all $\mathbf{u}', \mathbf{v}' \in N_{1/2}(\mathcal{U}_\mathbf{S}, \|\cdot\|_2)$, then $\mathbf{P}$ is also a $(1 \pm \varepsilon)$ $\ell^2$-embedding for $\mathcal{U}_\mathbf{S}$, that is, $\langle \mathbf{Pu}, \mathbf{Pv} \rangle = (1 \pm \varepsilon) \langle \mathbf{u}, \mathbf{v} \rangle$ for all $\mathbf{u}, \mathbf{v} \in \mathcal{U}_\mathbf{S}$. To see this, any $\mathbf{u} \in \mathcal{U}_\mathbf{S}$ can be expressed by

$$\mathbf{u} = \sum_{j=0}^{\infty} \mathbf{u}^j, \quad \|\mathbf{u}^j\|_2 \leq \frac{1}{2^j}, \mathbf{u}^j \in N_{1/2}(\mathcal{U}_\mathbf{S}, \|\cdot\|_2). \tag{57}$$

(57) can be proved by induction. By the definition of $1/2$-net, there exists $\mathbf{u}^0 \in N_{1/2}(\mathcal{U}_\mathbf{S}, \|\cdot\|_2)$ such that $\|\mathbf{u} - \mathbf{u}^0\|_2 \leq 1/2$ and $\mathbf{u} = \mathbf{u}^0 + (\mathbf{u} - \mathbf{u}^0)$. Also, there exists $\mathbf{u}'^1 \in N_{1/2}(\mathcal{U}_\mathbf{S}, \|\cdot\|_2)$ such that $\|((\mathbf{u} - \mathbf{u}^0))/\|(\mathbf{u} - \mathbf{u}^0)\|_2 - \mathbf{u}'^1\|_2 \leq 1/2$, so that $\mathbf{u} = \mathbf{u}^0 + \mathbf{u}^1 + (\mathbf{u} - \mathbf{u}^0 - \mathbf{u}^1)$ with $\mathbf{u}^1 = \|(\mathbf{u} - \mathbf{u}^0)\|_2 \mathbf{u}'^1$, $\|\mathbf{u}^1\|_2 \leq 1/2$, and $\|\mathbf{u} - \mathbf{u}^0 - \mathbf{u}^1\|_2 \leq 1/2^2$. As the induction step, if $\mathbf{u} = \sum_{j=0}^k \mathbf{u}^j + \left( \mathbf{u} - \sum_{j=1}^k \mathbf{u}^j \right)$ for $k \geq 1$ such that $\|\mathbf{u}^j\|_2 \leq \frac{1}{2^j}$ for all $j \in [0, k]$ and $\left\| \mathbf{u} - \sum_{j=1}^k \mathbf{u}^j \right\|_2 \leq 1/2^{k+1}$. Then there exists $\mathbf{u}'^{k+1} \in N_{1/2}(\mathcal{U}_\mathbf{S}, \|\cdot\|_2)$ such that $\left\| \left( \mathbf{u} - \sum_{j=1}^k \mathbf{u}^j \right) / \left\| \mathbf{u} - \sum_{j=1}^k \mathbf{u}^j \right\|_2 - \mathbf{u}'^{k+1} \right\|_2 \leq 1/2$. As a result, $\mathbf{u} = \sum_{j=0}^{k+1} \mathbf{u}^{k+1} + \left( \mathbf{u} - \sum_{j=1}^{k+1} \mathbf{u}^j \right)$ where $\mathbf{u}^{k+1} = \left\| \mathbf{u} - \sum_{j=1}^k \mathbf{u}^j \right\|_2 \mathbf{u}'^{k+1}$, $\|\mathbf{u}^{k+1}\|_2 \leq \left\| \mathbf{u} - \sum_{j=1}^k \mathbf{u}^j \right\|_2 \leq 1/2^{k+1}$ and $\left\| \mathbf{u} - \sum_{j=1}^{k+1} \mathbf{u}^j \right\|_2 \leq 1/2^{k+2}$.

It follows by (57) that

$$
\begin{aligned}
\|\mathbf{Pu}\|_2^2 &= \left\| \mathbf{P} \left( \sum_{j=0}^{\infty} \mathbf{u}^j \right) \right\|_2^2 \\
&= \sum_{j=0}^{\infty} \|\mathbf{Pu}^j\|_2^2 + 2 \sum_{i<j} \langle \mathbf{Pu}^i, \mathbf{Pu}^j \rangle \\
&= \sum_{j=0}^{\infty} (1 \pm c) \|\mathbf{u}^j\|_2^2 + 2 \sum_{i<j} \langle \mathbf{u}^i, \mathbf{u}^j \rangle \pm \frac{\varepsilon}{4} \sum_{i<j} \|\mathbf{u}^i\|_2 \|\mathbf{u}^j\|_2 \\
&= \left( \sum_{j=0}^{\infty} \|\mathbf{u}^j\|_2^2 + 2 \sum_{i<j} \langle \mathbf{u}^i, \mathbf{u}^j \rangle \right) \pm \frac{\varepsilon}{4} \left( \sum_{j=0}^{\infty} \|\mathbf{u}^j\|_2^2 + 2 \sum_{i<j} \|\mathbf{u}^i\|_2 \|\mathbf{u}^j\|_2 \right) \\
&= \|\mathbf{u}\|_2^2 \pm \varepsilon \|\mathbf{u}\|_2^2 = 1 \pm \varepsilon.
\end{aligned}
\tag{58}
$$

It follows by (58) that $\mathbf{P}$ is a $(1 \pm \varepsilon)$ $\ell^2$-embedding for $\mathcal{U}_\mathbf{S}$. Recall that $\mathbf{P}$ is a $(1 \pm \varepsilon/4)$ $\ell^2$-embedding for $\mathcal{V}_s \supseteq \bigcup_{\mathbf{S} \subseteq [d], |\mathcal{S}|=s} N_{1/2}(\mathcal{U}_\mathbf{S}, \|\cdot\|_2)$. By the above argument, $\mathbf{P}$ is also a $(1 \pm \varepsilon)$ $\ell^2$-embedding for $\mathcal{U}_\mathbf{S}$ for any $\mathbf{S} \subseteq [d], |\mathbf{S}| = s$.

For any $\mathbf{v} \in \mathbb{R}^d$ such that $\|\mathbf{v}\|_0 \leq s, \|\mathbf{v}\|_2 = 1$, there must exists a set $\mathbf{S} \subseteq [d], |\mathbf{S}| = s$ such that $\mathbf{X}\mathbf{v}/\|\mathbf{X}\mathbf{v}\|_2 \in \mathcal{U}_\mathbf{S}$. As a result,

$$\left| \mathbf{v}^\top \mathbf{X}^\top \mathbf{X} \mathbf{v} - \mathbf{v}^\top \tilde{\mathbf{X}}^\top \tilde{\mathbf{X}} \mathbf{v} \right| \leq \varepsilon \|\mathbf{X}\mathbf{v}\|_2^2 \leq \varepsilon\sqrt{s}, \tag{59}$$

where the last inequality follows by $\max_{i \in [d]} \|\mathbf{X}^i\|_2 \leq 1$. Therefore, $\rho_{\tilde{\mathcal{L}},+}(s) \leq \rho_{\mathcal{L},+}(s) + \varepsilon\sqrt{s}$, and $\rho_{\tilde{\mathcal{L}},-}(s) \geq \rho_{\mathcal{L},-}(s) - \varepsilon\sqrt{s}$.

Because $\mathbf{P}$ is a $(1 \pm \varepsilon/4)$ $\ell^2$-embedding for $\mathcal{V}_s \supseteq \bigcup_{i \in [d]} N_{1/2}(\mathbf{e}^{(i)}, \|\cdot\|_2)$, $\mathbf{P}$ is also $(1 \pm \varepsilon)$ $\ell^2$-embedding for $\mathbf{e}^{(i)}$ for all $i \in [d]$. Therefore, we have

$$\sup_{i \in [d]} \mathbf{e}^{i^\top} \left( \mathbf{X}^\top \mathbf{X} \bar{\boldsymbol{\beta}} - \tilde{\mathbf{X}}^\top \tilde{\mathbf{X}} \right) \bar{\boldsymbol{\beta}} \leq \varepsilon \|\mathbf{X}\mathbf{e}_i\|_2 \|\mathbf{X}\bar{\boldsymbol{\beta}}\|_2,$$

so that

$$\left\| \nabla\tilde{\mathcal{L}}(\bar{\boldsymbol{\beta}}) - \nabla\tilde{\mathcal{L}}(\bar{\boldsymbol{\beta}}) \right\|_\infty = \left\| \mathbf{X}^\top \mathbf{X} \bar{\boldsymbol{\beta}} - \tilde{\mathbf{X}}^\top \tilde{\mathbf{X}} \bar{\boldsymbol{\beta}} \right\|_\infty = \sup_{i \in [d]} \mathbf{e}^{i^\top} \left( \mathbf{X}^\top \mathbf{X} \bar{\boldsymbol{\beta}} - \tilde{\mathbf{X}}^\top \tilde{\mathbf{X}} \right) \bar{\boldsymbol{\beta}}$$

$$\leq \varepsilon \|\mathbf{X}\mathbf{e}^i\|_2 \|\mathbf{X}\bar{\boldsymbol{\beta}}\|_2 \leq \varepsilon\sqrt{\tilde{s}} \|\bar{\boldsymbol{\beta}}\|_2.$$

$\square$

**Proof of Theorem 4.4**. It follows by (55) in Lemma D.7 that

$$\rho_{\tilde{\mathcal{L}},-}(s) \geq \rho_{\mathcal{L},-}(s) - \varepsilon\sqrt{s} > 0$$

with $s = s_0 = \bar{s} + 2\tilde{s}$. Moreover, it also follows by (55) that

$$C'\rho_{\tilde{\mathcal{L}},-}(\bar{s} + 2\tilde{s}) \geq C' \left( \rho_{\mathcal{L},-}(\bar{s} + 2\tilde{s}) - \varepsilon\sqrt{\bar{s} + 2\tilde{s}} \right) \geq \zeta_-.$$

In addition, by (55) we have

$$\kappa' = \frac{\rho_{\mathcal{L},+}(s_0) + \varepsilon\sqrt{s_0} - \zeta_+}{\rho_{\mathcal{L},-}(s_0) - \varepsilon\sqrt{s_0} - \zeta_-} \geq \frac{\rho_{\tilde{\mathcal{L}},+}(s_0) - \zeta_+}{\rho_{\tilde{\mathcal{L}},+}(s_0) - \zeta_-} = \tilde{\kappa}$$

with $\tilde{\kappa}$ specified in Assumption 4. Therefore, if $\left( 144\kappa'^2 + 250\kappa' \right) \bar{s} < \tilde{s}$, we have $\left( 144\tilde{\kappa}^2 + 250\tilde{\kappa} \right) \bar{s} = \tilde{C}\bar{s} < \tilde{s}$ which completes the first part of the claim.

Now we need to verify that either condition (a) or condition (b) can lead to the following conditions:

$$\rho_{\mathcal{L},-}(s_0) > \varepsilon\sqrt{s_0}, \tag{60}$$

$$\zeta_- \leq C' \left( \rho_{\mathcal{L},-}(s_0) - \varepsilon\sqrt{s_0} \right), \tag{61}$$

$$\left( 144\kappa'^2 + 250\kappa' \right) \bar{s} < \tilde{s}, \quad \kappa' = (\rho_{\mathcal{L},+}(s_0) + \varepsilon\sqrt{s_0} - \zeta_+)/(\rho_{\mathcal{L},-}(s_0) - \varepsilon\sqrt{s_0} - \zeta_-). \tag{62}$$

For condition (a), with $\log d/n \overset{n \to \infty}{\longrightarrow} 0$, $\varepsilon = C_1\sqrt{\log d/n}$, we have $\varepsilon \overset{n \to \infty}{\longrightarrow} 0$. It can be verified that $\kappa' \overset{n \to \infty}{\longrightarrow} \kappa$, and (60)-(62) holds with sufficiently large $n$ when Assumption 3 holds.

For condition (b), with $\zeta_- = C_2\rho_{\mathcal{L},-}(s_0), \varepsilon\sqrt{s_0} \leq C_3\rho_{\mathcal{L},-}(s_0), C_2 + C_3 < 1$ and $C_2 \leq C'(1 - C_3)$, (60)-(61) hold. Moreover, we have

$$\kappa' = \frac{\rho_{\mathcal{L},+}(s_0) + \varepsilon\sqrt{s_0} - \zeta_+}{\rho_{\mathcal{L},-}(s_0) - \varepsilon\sqrt{s_0} - \zeta_-} \leq \frac{(1 + C_3)\rho_{\mathcal{L},+}(s_0)}{(1 - C_2 - C_3)\rho_{\mathcal{L},-}(s_0)} \leq \frac{(1 + C_3)(1 + \delta)}{(1 - C_2 - C_3)(1 - \delta)} = \kappa_0.$$

As a result. $\tilde{s} > \left( 144\kappa_0^2 + 250\kappa_0 \right) \bar{s}$ leads to (62).

$\square$

**Proof of Theorem 4.5**. It follows by (56) in Lemma D.7 and Lemma D.8 that

$$\left\| \nabla\tilde{\mathcal{L}}(\bar{\boldsymbol{\beta}}) \right\|_\infty \leq \left\| \nabla\mathcal{L}(\bar{\boldsymbol{\beta}}) \right\|_\infty + \varepsilon\sqrt{\tilde{s}} \|\bar{\boldsymbol{\beta}}\|_2 \leq \left( C_1\sqrt{\tilde{s}} \|\bar{\boldsymbol{\beta}}\|_2 + 2\sigma \right) \sqrt{\frac{\log d}{n}}. \tag{63}$$

Let $\lambda_{\text{tgt}} = 8\left(C_1\sqrt{\bar{s}}\|\bar{\beta}\|_2 + 2\sigma\right)\sqrt{\frac{\log d}{n}}$. Because Assumption 4 holds, we can apply the approximate path following method described in (Wang et al., 2014, Algorithm 1) to solve the original problem (1) and the sketched problem (2) to obtain $\tilde{\beta}^*$ and $\tilde{\beta}^*$ such that $\beta^*$ is an critical point of problem (1) and $\tilde{\beta}^*$ is a critical point of (2) with $\tilde{\mathbf{X}} = \mathbf{PX}$ and nonconvex regularizer $h_{\lambda_{\text{tgt}}}$. Let $\mathcal{S} = \text{supp}\left(\bar{\beta}\right)$. We can repeat the proof for (Wang et al., 2014, Theorem 5.5) and conclude that

$$\|\beta^*_{\mathcal{S}^c}\|_0 \leq \tilde{s}, \quad \left\|\tilde{\beta}^*_{\mathcal{S}^c}\right\|_0 \leq \tilde{s}.$$

The above inequalities show that $\left|\text{supp}\left(\tilde{\beta}^* - \beta^*\right)\bigcup\text{supp}\left(\beta^*\right)\bigcup\text{supp}\left(\tilde{\beta}^*\right)\right| \leq s_0 = \bar{s} + 2\tilde{s}$. It then follows by Corollary 4.2 that

$$\left\|\tilde{\beta}^* - \beta^*\right\|_2 \leq \frac{\varepsilon\sqrt{\rho_{\mathcal{L},+}(s_0)}}{(1-\varepsilon)\rho_{\mathcal{L},-}(s_0) - \zeta_-}\|\beta^*\|_{\mathbf{X}}. \tag{64}$$

In addition, it follows by (Wang et al., 2014, Theorem 4.7, Theorem 4.8) that the parameter estimation error of $\beta^*$ satisfies

$$\|\beta^* - \bar{\beta}\|_2 \leq (21/8)/\left(\rho_{\mathcal{L},-}(s_0) - \zeta_-\right)\sqrt{\bar{s}}\lambda_{\text{tgt}}. \tag{65}$$

It follows by (65) and (64) that

$$
\begin{aligned}
\left\|\tilde{\beta}^* - \beta^*\right\|_2 &\leq \frac{\varepsilon\sqrt{(\bar{s}+\tilde{s})\rho_{\mathcal{L},+}(s_0)}}{(1+C')/2 \cdot \rho_{\mathcal{L},-}(s_0) - \zeta_-}\|\beta^*\|_2 \\
&\leq \frac{\varepsilon\sqrt{(\bar{s}+\tilde{s})\rho_{\mathcal{L},+}(s_0)}}{(1+C')/2 \cdot \rho_{\mathcal{L},-}(s_0) - \zeta_-}\|\bar{\beta}\|_2 + \frac{\varepsilon\sqrt{(\bar{s}+\tilde{s})\rho_{\mathcal{L},+}(s_0)}}{(1+C')/2 \cdot \rho_{\mathcal{L},-}(s_0) - \zeta_-} \cdot (21/8)/\left(\rho_{\mathcal{L},-}(s_0) - \zeta_-\right)\sqrt{\bar{s}}\lambda_{\text{tgt}} \\
&\leq \frac{\varepsilon\sqrt{(\bar{s}+\tilde{s})\rho_{\mathcal{L},+}(s_0)}}{(1+C')/2 \cdot \rho_{\mathcal{L},-}(s_0) - \zeta_-}\|\bar{\beta}\|_2 + \frac{\sqrt{\bar{s}}\lambda_{\text{tgt}}}{8\left(\rho_{\mathcal{L},-}(s_0) - \zeta_-\right)} \\
&\leq \frac{C_1\|\bar{\beta}\|_2\sqrt{(1+\tilde{s}/\bar{s})\rho_{\mathcal{L},+}(s_0)}}{(1+C')/2 \cdot \rho_{\mathcal{L},-}(s_0) - \zeta_-}\sqrt{\frac{\bar{s}\log d}{n}} + \frac{\sqrt{\bar{s}}\lambda_{\text{tgt}}}{8\left(\rho_{\mathcal{L},-}(s_0) - \zeta_-\right)}
\end{aligned} \tag{66}
$$

(14) then follows by (65) and (66).

$\square$

**Lemma D.8** ((Zhang, 2010c, Lemma 5)). Under the linear model in Section 4, that is, $\mathbf{y} = \bar{\mathbf{X}}\bar{\beta} + \varepsilon$ where $\varepsilon$ is a noise vector of i.i.d. sub-gaussian elements with variance proxy $\sigma^2$. Let $\mathbf{X} = \bar{\mathbf{X}}/\sqrt{n}$, $\mathbf{y} = \bar{\mathbf{y}}/\sqrt{n}$, and $\mathcal{L}(\beta) = 1/2 \cdot \beta^\top\mathbf{X}^\top\mathbf{X}\beta - \mathbf{y}^\top\mathbf{X}\beta$. Then with probability at least $1 - \eta$ for any $\eta \in (0, 1)$,

$$\left\|\nabla L(\bar{\beta})\right\|_\infty \leq \sqrt{2}\sigma\sqrt{\frac{\log(2d/\eta)}{n}}. \tag{67}$$

**Lemma D.9** ((Aubrun & Szarek, 2017, Proposition 5.20)). Let $d > 2$. If $f\colon \mathbb{S}^{d-1} \to \mathbb{R}$ is a 1-Lipschitz function, then for any $t > 0$,

$$\Pr\left[|f(\mathbf{x}) - \mathbb{E}\left[f(\mathbf{x})\right]| > t\right] \leq 2\exp\left(-dt^2/2\right), \tag{68}$$

where $\mathbf{x}$ is drawn uniformly from the unit sphere $\mathbb{S}^{d-1}$ in $\mathbb{R}^d$.

