# OpenReview forum: "Sketching for Convex and Nonconvex Regularized Least Squares with Sharp Guarantees"
_ICLR.cc/2025/Conference — ICLR 2025 Poster_

### Official Review · Reviewer_HrLh · 2024-10-29

**Soundness:** 3
**Presentation:** 3
**Contribution:** 3
**Rating:** 6
**Confidence:** 3

**Summary:**

This paper proposes a sketching method for convex and nonconvex regularized least squares and sharp theoretical guarantees are provided for the algorithms.

**Strengths:**

1. The introduction of the method is clear and easy to follow.
2. The motivation is clearly conveyed.

**Weaknesses:**

Please see following questions.

**Questions:**

1. This work appears to be a straightforward extension of (Pilanci & Wainwright, 2016) with the addition of a regularization term. Authors should clarify the unique difficulties and challenges introduced by this regularization term. It does mention some differences like sampling the projection matrix only once. However, can this be easily adjusted in (Pilanci & Wainwright, 2016)? In addition, what are the primary obstacles to applying traditional analyses of convex and nonconvex regularizations in a least squares setting without sketching? What are the difficulties by extending the analysis in (Pilanci & Wainwright, 2016)? What are the new techniques introduced in this paper to study the error bound?

2. The notations are not very clear. In the paper, $\tilde{\beta}^*$ represents the critical point of the objective function (2). When studying the theoretical properties, is it the solution of SRO algorithm or the solution to iterative SRO algorithm? What are the errors bound for solution of SRO and what are the error bounds for iterative SRO?

3. In Section 3 about iterative SRO, it only concerns the convex regularization for $h_\lambda(\beta)$. Is there any particular reason that it cannot be applied to nonvex regularization? How about the error bound for the nonconvex regularization with iterative SRO?

4. The organization of the theoretical guarantees and proof sketch is not very clear. For instance, Corollary 4.2 is introduced before introducing Theorem 5.2, while Corollary 4.2 needs the conditions in Theorem 5.2.

5. In Table 1, why the running time for SRO is longer than those iterative algorithm? The first step of iterative SRO is somehow equivalent to SRO?

---

> ### Author Response · Authors · 2024-11-29
> **Response to Reviewer HrLh Part 1**
>
> We appreciate the review and the suggestions in this review. The raised issues are addressed below. In the following text, the line numbers are for the revised paper.
>
> **(1) Novelty of Our Results and Their Significant Difference from [Pilanci2016]**
>
> **We respectfully point out that the claim “This work appears to be a straightforward extension of [Pilanci2016] with the addition of a regularization term…” in this review is a factual misunderstanding**.  Our results are novel and significantly different from those in [Pilanci2016], which is detailed in line 437-459 of the revied paper and copied below for your convenience.
> It is remarked that [Pilanci2016] only handles convex constrained least square problems of the form $\min _ {\mathbf X \in \mathcal C} || \mathbf X \mathbf \beta-\mathbf y ||^2$ where the constraint set $\mathcal C$ is a convex set, while our results cover regularized convex and nonconvex problems with minimax optimal rates. It is emphasized that the techniques in [Pilanci2016] can never be applied to the regularized problems considered in this paper. [Pilanci2016] heavily relies on certain complexity measure of the constraint set $\mathcal C$, such as the Gaussian width. It shows that the complexity of such constraint set $\mathcal C$ is bounded, so that sketching with such constraint set $\mathcal C$ of limited complexity only incurs a relatively small approximation error. However, there is never such constraint set in the original problem (Eq. (1)) or the sketched problem (Eq. (2)), so that such complexity based analysis for sketching cannot be applied to this work. Furthermore, as mentioned in Section 1.1, Iterative SRO does not need to sample the projection matrix and compute the sketched matrix at each iteration, in contrast with IHS [Pilanci2016] where a separate projection matrix is sampled and the sketched matrix is computed at each iteration. As evidenced by Table 1 in Section 7.1, Iterative SRO is more efficient than its “HIS” counterpart where sketching is performed at every iteration while enjoying comparable approximation error.
>
> **(2) Improved Presentation**
>
> **”The notations are not very clear. …represents the critical point of the objective function (2) ..."**.
>
> The meaning of …$\tilde {\mathbf \beta}^*$ has been clearly indicated in every theoretical result in the revised paper. In particular, when using Iterative SRO for sparse convex learning in Section 4.1, it is clearly stated in Theorem 4.1 that $\tilde {\mathbf \beta}^* = \mathbf \beta^{(N)}$ (line 262-263 of the revised paper or line 254-255 of the original submission). When using SRO for sparse nonconvex learning in Section 4.2, it is clearly stated in Theorem 4.2 that $\tilde {\mathbf \beta}^*$ is the optimization result of the sketched problem (Eq. (2)) in line 368-369 of the revised paper.  To make the meaning of $\tilde {\mathbf \beta}^*$ even clearer, it is stated in line 237-238 of the revised paper that “In Section 4.1, $\tilde {\mathbf \beta}^*$ is obtained by Algorithm 1 through $\tilde {\mathbf \beta}^* = \mathbf \beta^{(N)}$. In Section 4.2, $\tilde {\mathbf \beta}^*$ is the optimization result of the sketched problem (2)” (Section 4.1 in line 238 should be Section 4.2).
>
> **(3) "…iterative SRO, it only concerns the convex regularization for $h_{\lambda}(\mathbf \beta)$. Is there any particular reason that it cannot be applied to nonvex regularization?..."**
>
> We respectfully point out that it is a factual misunderstanding that Iterative SRO cannot be applied to nonconvex regularizer. Theorem 3.1 (in both original submission and the revised paper) shows that iterative SRO can handle nonconvex regularizer $h = h_{\lambda}(\cdot)$ as long as the Frechet subdifferential of $h$ is $L_h$-smooth, and please refer to line 215-216 for the definition of $L_h$-smoothness. It is remarked that a $L_h$-smooth function $h$ can definitely be nonconvex.
>
>
> **(4) "organization…is not very clear. For instance, Corollary 4.2 is introduced before introducing Theorem 5.2, while Corollary 4.2 needs the conditions in Theorem 5.2."**
>
> We have improved the organization of the theoretical results. In particular, Theorem 5.2 and Theorem 5.4 (which now become Theorem D.2 and Theorem D.3 of the revised paper), as the intermediate steps for the proof of Theorem 3.1, have been moved to Section D of the revised paper, following the suggestion of Reviewer 2raj for the improved clarity of this paper. Now all the results presented before Theorem D.2 in the revised paper, including Corollary 4.2, do not depend on the conditions of the original Theorem 5.2 (or Theorem D.2 of the revised paper).

---

> ### Author Response · Authors · 2024-11-29
> **Response to Reviewer HrLh Part 2**
>
> **(5) "…why the running time for SRO is longer than those iterative algorithm?..."**
>
> The reason that the running time of SRO is longer than that of Iterative SRO is explained in line 511-518 of the revised paper, which is copied below for your convenience. In summary, the Fast Iterative Shrinkage-Thresholding Algorithm (FISTA) is used as the optimization algorithm to solve the original and the sketched problems, and Iterative SRO can use much less iterations of FISTA compared to SRO, explaining that Iterative SRO takes less running time than SRO in Table 1.
>
> The maximum iteration number of FISTA for Iterative SRO ($2000$) is much smaller than that for SRO ($10000$). This is because Iterative SRO uses an iterative sketching process where the approximation error is geometrically reduced with respect to the iteration number $t$ in Algorithm 1, so that each iteration of Iterative SRO is only required to have a moderate approximation error which can be larger than the approximation error of SRO thus a smaller iteration number of FISTA suffices for each iteration of Iterative SRO. Such analysis also explains the fact that Iterative SRO is much faster than SRO, and in our experiment the maximum iteration number $N$ for Iterative SRO described in Algorithm 1 is always not greater than $5$.
>
> **References**
>
> [Pilanci2016] Pilanci et al. Iterative hessian sketch: Fast and accurate solution approximation for constrained least-squares. Journal of Machine Learning Research, 2016.

---

> > ### Comment · Reviewer_HrLh · 2024-12-02
> > **Thanks for the response**
> >
> > Thanks authors for the effort in addressing my misunderstandings and confusion. I have raised the score for the paper.

---

### Official Review · Reviewer_Y2Sn · 2024-11-01

**Soundness:** 3
**Presentation:** 1
**Contribution:** 3
**Rating:** 6
**Confidence:** 4

**Summary:**

The article is concerned with the solution of sketch-based solution of regularized least squares problems, i.e. problems of the form $\min 1/2 \Vert X\beta-y \Vert^2 + f(\beta)$, where $X$ is a matrix and $f$ is a regularization term. If $X$ is of low rank, it can be well approximated via a sketch, i.e a matrix $\widetilde{X}=PX$, where $P$ is a random projector. The idea of sketching is to use this fact to reduce the complexity of the optimization problems.

Concretely, the authors propose the iterative SRO method. In SRO, the quadratic term $\langle X\beta, X\beta\rangle$ of the loss function is exchanged with a term based on a sketch, i.e $\langle \widetilde{X}\beta, \widetilde{X}\beta\rangle$. Since $\widetilde{X}$ is a matrix of smaller dimension than $X$, the latter problem is less expensice to solve. In iterative SRO, this procedure is repeated to obtain a better and better estimate of the true solution $\beta_*$.

The main result of the paper says that as soon as $P$ is an $\ell_2$-subspace embedding for the random matrix $X$, the iterative SRO will converge at a linear rate towards the true solution $\beta_*$. The authors also prove a result about applying the method to do sparse regularization - they show that with a moderate amount of iterations, the iterative SRO achieves the minimax sampling rate when solving the LASSO problem.

**Strengths:**

This article has many positive sides. The research questions of this article are sound, as is the algorithm it proposes. Its results are relevant and the proofs are modulo typos correct. The experimental verification is also reasonable.
I would like to highlight the generality of the results (there is much freedom to choose both the regularization term and the sketching matrix $P$). I am also fond of the fact that the authors take the time to prove that there indeed are matrices $X$ that are both of low rank (making their sketching approach viable) but still has the RIP.

**Weaknesses:**

A big problem with this article is its presentation. Let me address some of the problems, in order of appearance.
- It takes some effort to sort out the relations between the problems (2), (5) and (6) in Section 3 (they are all equivalent - but it is non trivial to realize this due to the notation). Some ironing out of this would be good, possibly with an explicit calculation in an appendix.
- Many of the arguments in the beginning of proof of Theorem 5.2 are written down in an unnecessarily complicated: Since the vector in (25) is zero, there is no need to apply the Cauchy-Schwarz inequality to arrive at the inequality (26)
- In equation 35, $\kappa$ is still present although $\kappa$ has been set to 0 just before.
- There are many steps in the proof of the main result that are only sketched, in particular for $L_h$-smooth $h$ -- I think that I can reproduce the steps, but I really think that the proof of the main result of the paper deserves to have this step written out. *This is in my opinion the most pressing issue*.
- The sentence "We show that RIP($\delta,s$)" has appearantly lost its ending.
- Reference to equation (45) in the proof of Theorem 4.1 seems to be spurious.
- The term JLT in Theorem D.4 is undefined - I suppose it has something to do with a Johnson Lindenstrauss embedding. The meaning of $f_0$ is also unclear.
- Theorem D.6 has no statement.

Another weakness of the paper, in my opinion, is Theorem D.2. Due to the appearance of the term $\alpha_0$ in its current form, one needs to use $n\geq C(s\log(d))^{1/\alpha_0}$ to get a sparse recovery, which, due to the unknown size of $\alpha_0$, could be arbitrarily far away from the sample optimal $s\log(d)$. This question should at least be discussed.

However, I suspect that theorem D.2. can be made stronger. Would it not suffice to just set $X= UA$ where $U\in \mathbb{R}^{n,m}$ is in the Stiefel manifold (with $m=cn$) and $A\in \mathbb{R}^{m,d}$ Gaussian with $m\sim s\log(d)$?

**Questions:**

- In the beginning of section D.2, is $O(s\log(d)/n)$ a typo (should there not be a square root there)?
- Can the matrix in Theorem D.2 be constructed as I have outlined above, i.e. $X=UA$ with $U$ in the Stiefel manifold (and hence isometric) and $A$ a standard RIP matrix?
- In the formulation of Theorem 3.2, it is stated that $n\ll d$ and $\log(d) \ll n^\alpha_0$ -- it is unclear what this means and how it is used later in the proof.
- Can the authors provide some more details in their proof of their main theorem?

**Details Of Ethics Concerns:**

I think that the concerns of reviewer X44j bring up needs some attention. The paper the reviewer brings up indeed shares *large* similarities to the manuscript at hand. There are large sections of the text that are more or less verbatim copies of the version of [YangLi2021]. The ideas of the paper are also more or less identical, with the exception that Yang and Li do not propose to make a new sketch in each iteration. I however also do not see where the new draws of the matrix are used in the theoretical analysis in [YangLi2021]. In particular, if the authors of this article have applied a different strategy to draw the random sketches compared to [YangLi2021] to generate Figure 1, I find it highly unlikely that they arrive at the exact same error bounds. Since [YangLi2021] was not cited in the submitted version of the paper, this raises ethical concerns in my view.

In my personal opinion, the extent of these concerns depend very much on whether this is a case of self-plagiarism or 'bona-fide'-plagiarism. Note that the former could to some extent be justified: In certain academic disciplines, particularly in this type of applied mathematics, it is customary to first write a 'conference version' of a paper, which does not contain details of proofs etc., and send it to a more prestigious venue for 'actual' publication. I very much suspect that this is what has happened here. Due to the double blind policy of ICLR, it has been impossible to acknowledge the previous version, since that would reveal the identities of the authors. I am however not sure of this -- I have of course not tried to verify this, since this would break the double-blind review process from my perspective.

---

> ### Author Response · Authors · 2024-11-29
> **Response to Reviewer Reviewer Y2Sn Part 1**
>
> We appreciate the review and the suggestions in this review. The raised issues are addressed below. In the following text, the line numbers are for the revised paper.
>
> **(1) Improved Presentation**
>
> **"It takes some effort to sort out the relations between the problems (2), (5) and (6)…"**
>
> Herein we provide a detailed description of problems (2), (5) and (6). Problem (2) is the sketched problem corresponding to the original problem (1), where the data matrix $\mathbf X$ is replaced by its sketched version $\tilde {\mathbf X}$ in problem (2).
>
> Problem (5) is the intermediate problem at every iteration of the Iterative SRO described in Algorithm 1. The solution to problem (5), $\mathbf \beta^{(t)}$,  is supposed to approximate the original solution $\mathbf \beta^*$ better than its predecessor, $\mathbf \beta^{(t-1)}$ obtained at the previous iteration of the Iterative SRO. Theorem 3.1 shows that   $\mathbf \beta^{(t)}$ is geometrically close to $\mathbf \beta^*$ (by Eq. (8)).
>
> Problem (6)-(7) are used to explain the proof of Theorem 3.1, in particular, why  $\mathbf \beta^{(t)}$ can be geometrically close to $\mathbf \beta^*$. In particular, at the $t$-th iteration of Iterative SRO, the solution to problem (6) is in fact the gap between $\mathbf \beta^*$ and $\mathbf \beta^{(t)}$, or
> $\mathbf \beta^* -  \mathbf \beta^{(t)}$ . We then solve the sketched version of problem (6), which is problem (7), so that the solution $\hat {\mathbf \beta}$ to problem (7) is an approximation to $\mathbf \beta^* -  \mathbf \beta^{(t)}$. If such approximation can have a relative-error approximation error in Eq. (3), that is,
> $\hat {\mathbf \beta} – (\mathbf \beta^* -  \mathbf \beta^{(t)}) \le \rho (\mathbf \beta^* -  \mathbf \beta^{(t)})$,  then since $\mathbf \beta^{(t)}  = \mathbf \beta^{(t-1)} + \hat {\mathbf \beta} $, we have
> $|| \mathbf \beta^{(t)} - \mathbf \beta^*|| _ {\mathbf X}  =
> || \hat {\mathbf \beta} – (\mathbf \beta^* -\mathbf \beta^{(t-1)}) || _ {\mathbf X}  \le \rho || \mathbf \beta^* -\mathbf \beta^{(t-1)} || _ {\mathbf X} $. As a result, by mathematical induction, we have $|| \mathbf \beta^{(t)} - \mathbf \beta^*|| _ {\mathbf X} \le \rho^{t}  ||\mathbf \beta^*|| _ {\mathbf X} $ for all $t \ge 0$.
> We will put the above detailed description in the final version of this paper.
>
> **"…proof of Theorem 5.2 are written down in an unnecessarily complicated…"**,
> **"$\kappa$ is still present … set to $0$"**.
>
> We have revised the proof of this Theorem following your suggestions (now it becomes Theorem D. 2 of the revised paper). Please kindly refer to the “Proof of Theorem D.2” in Section D of the appendix of the revised paper. In addition, we have removed the term with $\kappa$ in this equation (now Eq. (37) of the revised paper).
>
> **"There are many steps in the proof of the main result that are only sketched, in particular for $L_h$-smooth $h$..."**.
>
> The detailed steps for  $L_h$-smooth $h$ are provided in the revised paper, and please kindly refer to line 1080-1100 in the proof of Theorem D.3, and line 1121-1133 in the proof of Theorem 3.1 for these detailed steps.
>
> **"The term JLT in Theorem D.4 is undefined…"**.
>
> In Theorem D.6 of the revised paper, it is now clearly mentioned that JLT refers to the Johnson–Lindenstrauss Transform, and it is defined in line 1307-1308.
>
> We would also like to mention that Theorem D.2 and Theorem D.3, as the intermediate results for the proof of Theorem 3.1, have been moved to Section D of the revised paper, following the suggestion of Reviewer 2raj for the improved clarity of this paper.  All the other presentation issues have  either been fixed in the revised paper, such as $\mathcal O(\sqrt{\bar s \log d/n})$ instead of $\mathcal O(\bar s \log d/n)$ in the beginning of Section D.3, or will be fixed in the final version of this paper (such as the reference to Eq. (51) instead of (52) in line 1253).

---

> ### Author Response · Authors · 2024-11-29
> **Response to Reviewer Reviewer Y2Sn Part 2**
>
> **(2) Improved Theorem D.2 (now Theorem D.4 in the revised paper)**.
>
> We have revised Theorem D.4 and provided more details in its proof in the revised paper.  Now $n$ should satisfy $n \ge (\Theta(s\log d))^{1/\alpha_0}$ where $\alpha_0$ is is an arbitrary
> positive constant such that $\alpha_0 \in (0,1)$. It is noted that we need $n \ge (\Theta(s\log d))^{1/\alpha_0}$ so that $\textup{RIP}(\delta,s)$ in Theorem D.4 happens with probability arbitrarily close to $1$ as $n \to \infty$.
>
> **We have also discussed the impact of $\alpha_0$ and the alterative construction of the low-rank matrix suggested in this review, and the trade-off between our construction and the suggested construction** in line 1209-1223 of the revised paper. **In particular, we show that the low-rank matrix constructed by the suggested way in fact satisfies $\textup{RIP}(\delta,s)$ with the optimal size $n \asymp \Theta(s \log d) $**.
>
> First, by letting the constant $\alpha_0 \to 1$, the lower bound for $n$ is $(\Theta(s \log d))^{1/\alpha_0}$ in Theorem D. 4, which can be close to the sample optimal $\Theta(s \log d)$. Furthermore, one can also construct the low-rank matrix $\mathbf X = \mathbf U \mathbf A$ where $\mathbf U \in {\mathbb R}^{n \times m}$ is sampled  from the Stiefel manifold $V_{m}({\mathbb R}^{n})$ comprising  all the $n \times m$ matrices of orthogonal columns with $n \ge m$, and all the elements of $\mathbf A \in {\mathbb R}^{m \times d}$ are i.i.d. Gaussian random variables with $\mathbf A_{ij} \sim \mathcal N(0,1/m)$ for $i \in [m], j \in [d]$. Then $\textup{rank}(\mathbf X) \le m$, and it follows by  Theorem 5.2 of [Baraniuk2008] that when $m \ge \Theta(s \log d)$, then $\mathbf A$ satisfies $\textup{RIP}(\delta,s)$. Since $\mathbf X^{\top} \mathbf X =  \mathbf A^{\top} \mathbf A$, it follows that w.h.p. (with high probability) $\mathbf X$ also satisfies $\textup{RIP}(\delta,s)$. The latter construction can admit the optimal size $n \asymp \Theta(s \log d) $. On the other hand, $\textup{rank}(\mathbf X) \to \infty$ as $d \to \infty$ in latter construction, while the construction in Theorem D.4 allows for arbitrarily specified rank of the constructed $\mathbf X$.
>
> **References**
>
> [Baraniuk2008] Baraniuk et al. A simple proof of the restricted isometry property for random matrices. Constructive Approximation, 2008.

---

> ### Comment · Reviewer_Y2Sn · 2024-12-02
> **Response to the update.**
>
> I thank the reviewers for the update. It in large addresses the issues of the presentation I had with the first version of the paper - with the exception of the two below concerns.
>
> First, the improved theorem D.2 is still not entirely clear to me -- the $\alpha_0$-constant now seems to be a constant that can be chosen freely. However, \alpha_0 only appears (without specification) when concentration of $\xi_2$-variables is applied. This makes it look like alpha_0 is the dimensions-of-freedom parameter of the $\chi_2$-variable? Is this the case? Then, $\alpha_0$ can not be arbitrariliy chosen. Also, in the proof, $n$ is still required to be larger than a threshold with unclear dependence of $\alpha_0$. This can and should still be made clearer in my opinion.
>
> Also, reviewer X44j makes a good point that the paper has a significant overlap with [YangLi2021]. I am not entirely convinced that the 'only-one-sketch' aspect of the work at hand is significant enough to claim novelty - while the algorithm in [YangLi2021] specifies that the random projections should be redrawn, it does not appear that this is used in their analysis. In my opinion, there are however still enough new theoretical results and proofs in the paper at hand compared to [YanLi2021] to justify a publication. However, I cannot fairly judge how large of an issue the overlap is given the information I have, and have therefore flagged the paper for an ethical review to make sure a fair judgement is made.
>
> All in all, the authors have made an effort to increase the readability of the paper, and addressed my concerns. Leaving the potential issue with the overlap aside, I will therefore raise my score to a 6.

---

> > ### Author Response · Authors · 2024-12-02
> > **Clarification that $\alpha_0$ does not depend on the dimension-of-freedom parameter of the chi-squared variable, and the justification that the raised research integrity flag does not reflect a factually existing issue.**
> >
> > Thank you for your feedback. Below are our further clarifications about $\alpha_0$ and the raised overlap issue, and we respectfully point out that the raised research integrity flag does not reflect a factually existing issue due to factual misunderstandings.
> >
> > **(1) Clarification that $\alpha_0$ does not depend on the dimension-of-freedom parameter of the chi-squared variable.**
> >
> > We would like to mention that the parameter $\alpha_0$ is chosen so that the chi-squared variable, $v \coloneqq ||\mathbf A \mathbf v||^2 \sim \chi^2(n)$ (where $||\mathbf A \mathbf v||^2$ is introduced in line 1182-1183 of the revised paper) satisfies the standard concentration equality
> > $\| \frac vn - 1 \| \le \Theta(n^{(\alpha_0-1)/2}+n^{\alpha_0-1})$ which holds with probability $1-\exp(n^{-\alpha_0})$, and we need
> > $\Theta(n^{(\alpha_0-1)/2}+n^{\alpha_0-1}) \overset{n \to \infty}{\to} 0$ so that $v = ||\mathbf A \mathbf v||^2 \overset{n \to \infty}{\to} 1$ with probability approaching to $1$ as $n \to \infty$. Please note that $n$ is the dimension-of-freedom parameter of the chi-squared variable $v$, and **$\alpha_0$ can be chosen as an arbitrary constant in $(0,1)$ independent of the dimension-of-freedom parameter $n$**.
> >
> >
> > **(2) Justification that this work is significantly different from  [YangLi2021]**
> >
> > We would like to emphasize that the focus of this paper is completely different from [YangLi2021] with more details about the novelty of our results with their significant difference from [YangLi2021].
> >
> > **First of all, we respectfully point that the proof of the iterative algorithm in [YangLi2021] needs to resample a new projection matrix $\mathbf P$ and compute a new sketched matrix $\tilde {\mathbf X} = \mathbf P \mathbf X$ at every iteration, and the proof of Theorem 3 of [YangLi2021], which shows the theoretical guarantee of their iterative algorithm, in fact depends on a separate projection matrix and a new sketched matrix at every iteration**. The fundamental reason of such resampling in [YangLi2021] is that every iteration of  the [YangLi2021]'s iterative algorithm needs to apply the approximation error bound in Theorem 1 of [YangLi2021] which is not based on oblivious $\ell^2$-subspace embedding. In a strong contrast, our Theorem D.2 exhibits the approximation error bound which is based on oblivious $\ell^2$-subspace embedding, so that all the iterations of our Iterative SRO can use the same projection matrix and the same sketched matrix taking advantage of the low-rankness of the data matrix. Such difference constitutes significant advantages over [YangLi2021] both empirically (better efficiency of Iterative SRO in Table 1) and theoretically (Theorem D.2 with the general approximation error bound for oblivious $\ell^2$-subspace embedding and a clear specification of the sketch size $\tilde n$ for different subspace embeddings, which are not offered by [YangLi2021]).
> >
> > **Second, we respectfully point out that the focus of this work is to establish minimax optimal rates for sparse convex and nonconvex learning problems by sketching, which has not been addressed by existing works in the literature including  [YangLi2021]**. While [YangLi2021] only focuses on the optimization perspective, that is, approximating the solution to the original optimization problem by the solution to the sketched problem, the focus of this work needs much more efforts beyond the efforts made in [YangLi2021] for optimization only: we need to show that the solution to the sketched problem can still enjoy the minimax optimal rates for estimation of the sparse parameter vector for both sparse convex and nonconvex learning problems. Such efforts and results in  minimax optimal rates for sparse convex and nonconvex learning problems by sketching have not been offered by previous works including [YangLi2021], which are provided in Section 4 of this paper. Such minimax optimal results are established in a highly non-trivial manner. For example, to the best of our knowledge, Theorem 4.1 is among the first in the literature which uses an iteratively sketching algorithm to achieve the minimax optimal rate for sparse convex learning. Furthermore, Theorem 4.5 shows that sketching can also lead to the minimax optimal rate even for sparse nonconvex problems, while sketching for nonconvex problems is still considered difficult and open in the literature.
> >
> > We will further emphasize such difference from [YangLi2021]  in the introduction section of this paper so that readers can more easily understand such difference and the novelty and significance of this work. **At this point, we sincerely request this reviewer to remove the research integrity flag as it does not factually reflect an existing issue. Please also kindly note that Reviewer X44j, who originally raised the same overlap issue, has clearly indicated that the overlap issue has been solved**.
> >
> > Thank you for your time and efforts carefully reviewing this paper, and we look forward to your response.

---

> > > ### Comment · Reviewer_Y2Sn · 2024-12-02
> > > **Clarifications**
> > >
> > > *The constant in Theorem D.2*
> > >
> > > Regarding the $\alpha_0$, I think now understand the issue. First, let me apologize for misunderstanding. I simply did not understand where the $\alpha_0$ came from. The reason for this is that I probably do not know the concrete concentration inequality the authors are referring to. The one I know (from e.g Vershynin) is $\mathbb{P}(|\frac{1}{n}\Vert{g}\Vert^2-1|\geq u) \leq 2\exp(-cn\max(u,u^2))$, which in this setting would mean that $|F(v)-1|\leq \Theta(\delta^2)$ with a probability smaller than $2\exp(-\tilde{c}n\delta^2)$. I know realize that there is a multiplicative 2 in front of the exponential term which is not present in the authors bound -- this can of course be removed by bringing the $\alpha_0$ into the game and then argue that $n$ is large enough. This is not a huge issue - it is ultimately a question of readability and taste.
> > >
> > > *The ethical flagging*
> > >
> > > I completely agree that there is a lot of theory contained in this paper which is not present in [YangLi2021]. As I have stated, I think that the difference between the papers points to enough novelty!
> > >
> > > To explain my wording 'highly unlikely', I was referring to the exactly equal experimental results in the first version of the paper, not to the theorems - and I only used the wording due to the randomized nature of the algorithms. Using the word 'error bounds' here is a genuine mistake - I apologize.
> > >
> > > The reason I still think this issue should be looked on by someone else (hence the flag) is that in in [the version of [YangLi2021] that I can find](https://ieeexplore.ieee.org/stamp/stamp.jsp?tp=&arnumber=9517998),  there are long passages that are verbatim the same as in the paper at hand. As I have explained, this may or may not be a problem depending on circumstances that I cannot check unless I try to circumvent the double blind reviewing process, I will keep the flag. I will refrain from commenting any more on this.
> > >
> > > *Final word*
> > >
> > > I reiterate that my score is a 6, and I hence think that the quality of the paper is good enough to be accepted.

---

> > > > ### Author Response · Authors · 2024-12-02
> > > > **Thank you for your clarification, and our further response**
> > > >
> > > > Dear Reviewer  Y2Sn,
> > > >
> > > > We really appreciate your timely response. We apologize for not addressing your concern for "exactly the same experimental results" in our previous response. It has been addressed in our updated comments, which is copied here for your convenience: "We would like to respectfully point out that we obtained the same Figure 1 because we used the same data as that in [YangLi2021] in the original Figure 1, which has already been mentioned in our response to Reviewer X44j. However, Figure 1 has been updated with different data in the experiment in the revised paper."
> > > >
> > > > **To solve the issue about text overlap with [YangLi2021] in the abstract and the introduction section, we have provided an updated "Abstract" and "Introduction" which will be used to replace the existing "Abstract" and the existing parts of the Introduction section of the current version of this paper, and they are provided in our previous response** titled "Clarification regarding the factually wrong statements in the "Ethics Concerns" Part 2". Please kindly let us know if you have remaining concerns for such text overlap issue, and **we would like to mention that so far [YangLi2021] has been cited sufficiently in both Introduction and the discussion in Section 5 of the revised paper and there is now no overlap in the "Abstract" and the "Introduction" sections except for commonly used standard mathematical description in this literature**. As you already kindly mentioned in your comment, a description and introduction to such applied mathematical results are standard. **We have tried our best to avoid any text overlap and there is in fact no intention to copy any text from  [YangLi2021]**.
> > > >
> > > > Thank you again for your time!
> > > >
> > > > Best Regards,
> > > >
> > > > The Authors

---

> ### Author Response · Authors · 2024-12-02
> **Clarification regarding the factually wrong statements in the "Ethics Concerns" Part 1**
>
> We would like to provide further clarification regarding the "Ethics Concerns" provided in the updated review, and we respectfully point out that the  "Ethics Concerns" contain several factually wrong statements which result in the wrong conclusion for the raised research integrity issue.
>
>  **"The ideas of the paper are also more or less identical..."** This statement is factually and completely wrong. We would like to emphasize that while [YangLi2021] also studies sketching for regularized optimization problem with an iterative sketching algorithm, the focus and results of this work are completely different from that in [YangLi2021], with a detailed discussion in Section 5 of the revised paper. In particular, due to our novel results in the approximation error bounds in Theorem D.2 and Theorem D.3, the proposed iterative sketching algorithm, Iterative SRO, does not need to sample a new projection matrix and compute the sketched matrix at every iteration, in a strong contrast to [YangLi2021]. Moreover, the focus of this work is to establish minimax optimal rates for sparse convex and nonconvex learning problems by sketching, which has not been addressed by existing works in the literature including [YangLi2021]. While [YangLi2021] only focuses on the optimization perspective, that is, approximating the solution to the original optimization problem by the solution to the sketched problem, the focus of this work needs much more efforts beyond the efforts made in [YangLi2021] for optimization only: we need to show that the solution to the sketched problem can still enjoy the minimax optimal rates for estimation of the sparse parameter vector for both sparse convex and nonconvex learning problems. Such minimax optimal results are established in a highly non-trivial manner. For example, to the best of our knowledge, Theorem 4.1 is among the first in the literature which uses an iteratively sketching algorithm to achieve the minimax optimal rate for sparse convex learning. Furthermore, Theorem 4.5 shows that sketching can also lead to the minimax optimal rate even for sparse nonconvex problems, while sketching for nonconvex problems is still considered difficult and open in the literature.
>
> **"In particular, if the authors of this article have applied a different strategy to draw the random sketches compared to [YangLi2021] to generate Figure 1, I find it highly unlikely that they arrive at the exact same error bounds."** This statement is factually and completely
>  wrong. As mentioned above and in Section 5, we rigorously prove in Theorem 3.1 that the new sampling strategy used in our Iterative SRO achieves the approximation error in Eq. (8), which is the same theoretical guarantee in Theorem 3 of [YangLi2021]. However, our new sampling strategy is novel and efficient as explained above and in Section 5 of the revised paper. We would like to respectfully point out that we obtained the same Figure 1 because we used the same data as that in [YangLi2021] in the original Figure 1, which has already been mentioned in our response to Reviewer X44j. However, Figure 1 has been updated with different data in the experiment in the revised paper.
>
> **"Since [YangLi2021] was not cited in the submitted version of the paper, this raises ethical concerns in my view"**. We would respectfully point out that [YangLi2021]  was not cited initially due to negligence, and we cited [YangLi2021] with a detailed discussion about the novelty of our results and their significant differences from [YangLi2021]. Such detailed discussion has been acknowledged by Reviewer X44j, but unfortunately missed in this review.
>
> Finally, as mentioned in this comment, applied mathematical results admit common and standard description. To this end, we provide as follows a revised abstract and revised introduction for this paper, which will replace the corresponding and existing parts in the abstract and introduction of the current version of this paper and solve the overlapping text issue. **We sincerely hope this reviewer would reconsider the raised "research integrity issue" which was based on all the factual misunderstandings described above**.

---

> ### Author Response · Authors · 2024-12-02
> **Clarification regarding the factually wrong statements in the "Ethics Concerns" Part 2**
>
> **New Abstract (to replace the current Abstract)**
>
> Randomized algorithms play a crucial role in efficiently solving large-scale optimization problems. In this paper, we introduce Sketching for Regularized Optimization (SRO), a fast sketching algorithm designed for least squares problems with convex or nonconvex regularization. SRO operates by first creating a sketch of the original data matrix and then solving the sketched problem. We establish minimax optimal rates for sparse signal estimation by addressing the sketched sparse convex and nonconvex learning problems. Furthermore, we propose a novel Iterative SRO algorithm, which significantly reduces the approximation error geometrically for sketched convex regularized problems. To the best of our knowledge, this work is among the first to provide a unified theoretical framework demonstrating minimax rates for convex and nonconvex sparse learning problems via sketching. Experimental results validate the efficiency and effectiveness of both the SRO and Iterative SRO algorithms.
>
>
> **New Introduction (to replace the corresponding parts in the current Introduction Section)**
>
> Randomized algorithms for efficient optimization are a critical area of research in machine learning and optimization, with wide-ranging applications in numerical linear algebra, data analysis, and scientific computing. Among these, matrix sketching and random projection techniques have gained significant attention for solving sketched problems at a much smaller scale  (Vempala, 2004; Bout-
> sidis & Drineas, 2009; Drineas et al., 2011; Mahoney, 2011; Kane & Nelson, 2014). These methods have been successfully applied to large-scale problems such as least squares regression, robust regression, low-rank approximation, singular value decomposition, and matrix factorization (Halko et al., 2011; Lu et al., 2013; Alaoui & Mahoney, 2015; Raskutti & Mahoney, 2016; Yang et al., 2015;
> Drineas & Mahoney, 2016; Oymak et al., 2018; Oymak & Tropp, 2017; Tropp et al., 2017). Regularized optimization problems with convex or nonconvex regularization, such as the widely used in $\ell^1$ or $\ell^2$-norm regularized least squares commonly known as Lasso and ridge regression, play a fundamental role in machine learning and statistics. While prior research has extensively explored random projection and sketching methods for problems with standard convex regularization  (Zhang
> et al., 2016b) or convex constraints (Pilanci & Wainwright, 2016), there has been limited focus on analyzing regularized problems with general convex or nonconvex regularization frameworks.
>
>
> We would like to emphasize that while [YangLi2021] also studies sketching for regularized optimization problem, the focus and results of this work are completely different  from that in [YangLi2021], with a detailed discussion deferred to Section 5. In particular, due to our novel result in the approximation error bound (Theorem D.2-Theorem D.3), the proposed iterative sketching algorithm, Iterative SRO, does not need to sample a new projection matrix and compute the sketched matrix at every iteration, in a strong contrast to [YangLi2021]. Moreover, the focus of this work is to establish minimax optimal rates for sparse convex and nonconvex learning problems by sketching, which has not been addressed by existing works in the literature including [YangLi2021]. While [YangLi2021] only focuses on the optimization perspective, that is, approximating the solution to the original optimization problem by the solution to the sketched problem, the focus of this work needs much more efforts beyond the efforts made in [YangLi2021] for optimization only: we need to show that the solution to the sketched problem can still enjoy the minimax optimal rates for estimation of the sparse parameter vector for both sparse convex and nonconvex learning problems. Such efforts and results in minimax optimal rates for sparse convex and nonconvex learning problems by sketching have not been offered by previous works including [YangLi2021], which are provided in Section 4 of this paper. Such minimax optimal results are established in a highly non-trivial manner. For example, to the best of our knowledge, Theorem 4.1 is among the first in the literature which uses an iteratively sketching algorithm to achieve the minimax optimal rate for sparse convex learning. Furthermore, Theorem 4.5 shows that sketching can also lead to the minimax optimal rate even for sparse nonconvex problems, while sketching for nonconvex problems is still considered difficult and open in the literature.

---

### Official Review · Reviewer_X44j · 2024-11-04

**Soundness:** 3
**Presentation:** 3
**Contribution:** 3
**Rating:** 6
**Confidence:** 4

**Summary:**

The paper presents an approach called Sketching for Regularized Optimization (SRO), which can tackle a broad class of regularized optimization problems by employing sketching techniques. This method is applicable to both convex and nonconvex forms of regularization. To address potential approximation errors in SRO, this paper introduces an enhanced variant, that is, Iterative SRO, which systematically reduces these errors at a geometric rate.  A unified theoretical framework is also proposed to establish the minimax rates for sparse signal estimation in both convex and nonconvex settings.

**Strengths:**

1. This work addresses a wide range of optimization problems using sketching techniques, effectively handling both convex and nonconvex regularizations.
2. The introduction of SRO and its refined version demonstrate a well-thought-out approach that balances efficiency and accuracy in solving complex optimization tasks.
3. This work presents a unified framework that derives minimax rates for sparse signal estimations, solidifying the method's effectiveness for both convex and nonconvex settings.

**Weaknesses:**

1. This work appears to be a specific case of the work proposed by Yang and Li, ISIT (2021). The similarities include the theoretical framework (such as the general bound in Theorem 5.2),  the algorithmic approach (the SRO and Iterative SRO), experimental design and results (the findings presented in Figure 1 is exactly the same), and even the writing style. Please clarify the differences between these two works.
2. Please provide more explanation regarding Definition 5.2. For convex functions, the value of theta is less than or equal to 0. So what about the impact of nonconvex functions on this value? For example, what range might it fall within? Additionally, could you provide some examples of nonconvex functions?
3. It seems that the assumption of full column rank in the explanation below Theorem 3.1 is not realistic for practical applications. In other words, if the smallest singular value is relatively small, would the smoothness constant become very small or even approach zero?
4. Some numerical experiments on nonconvex regularization should be conducted.

References
[1] Yang, Y., & Li, P.  FROS: Fast Regularized Optimization by Sketching. In 2021 IEEE International Symposium on Information Theory (ISIT), pp. 2780-2785.

**Questions:**

Please see the weeknesses part.

---

> ### Author Response · Authors · 2024-11-29
> **Response to Reviewer X44j Part 1**
>
> We appreciate the review and the suggestions in this review. The raised issues are addressed below. In the following text, the line numbers are for the revised paper.
>
> **(1) Novelty of Our Results and Their Significant Difference from [YangLi2021]**
>
> In Line 437-495 of the revised paper, we explain the novelty of our Results and their significant difference from [YangLi2021], which is copied below for your convenience.
>
> It is remarked that our results, including the Iterative SRO algorithm in Algorithm 1and its theoretical guarantee in Theorem 3.1, Theorem D.2-Theorem D.3, and the minimax optimal rates by sketching for sparse convex learning in Theorem 3.1 and sparse nonconvex learning in Theorem 4.5, are all novel and significantly different from [YangLi2021] in the following two aspects, although [YangLi2021] also presents an iterative sketching algorithm for regularized optimization problems. First, Iterative SRO does not need to sample a projection matrix $\mathbf P \in {\mathbb R}^{\tilde n \times n}$ and compute the sketched matrix $\tilde {\mathbf X} = \mathbf P \mathbf X$ at every iteration, while the iterative sketching algorithm in [YangLi2021]  samples a different projection matrix and computes the sketched matrix at every iteration which incurs considerable computational cost for large-scale problem with large data size $n$. Such advantage of Iterative SRO over [YangLi2021] is attributed to the novel  theoretical results in Theorem D.2-Theorem D.3 and Theorem 3.1. In contrast with Theorem 1 in [YangLi2021], the approximation error bound Theorem D.2 is derived for sketching low-rank data matrix by oblivious $\ell^2$-subspace embedding with the sketched size $\tilde n$ clearly specified. As a result, Theorem D.3 presents the approximation error bounds for convex and certain nonconvex regularization by sketching with oblivious $\ell^2$-subspace embedding. Based on such results, Theorem 3.1 shows that a single projection matrix suffices for the iterative sketching process. Second, minimax optimal rates for convex and nonconvex sparse learning problems by sketching are established by our results, while there are no such minimax optimal rates by a sketching algorithm in [YangLi2021].  Theorem 4.1, to the best of our knowledge, is among the first in the literature which uses an iteratively sketching algorithm to achieve the minimax optimal rate for sparse convex learning. Furthermore, Theorem 4.5 shows that sketching can also lead to the minimax optimal rate even for sparse nonconvex problems, while sketching for nonconvex problems is still considered difficult and open in the literature.
>
> We would also like to mention that Theorem D.2 and Theorem D.3, as the intermediate steps for the proof of Theorem 3.1, have been moved to Section D of the revised paper, following the suggestion of Reviewer 2raj for the improved clarity of this paper. The presentation style of Theorem D.2 and Theorem D.3 has also been revised. Figure 1 in the original paper coincided with that of [YangLi2021]  because we used the same data for the experiment with Generalized Lasso as [YangLi2021]. In the revised paper, new experimental results are reported with different generated data and different $\gamma$ (recall that $\gamma$ decides the sketch size $\tilde n$) so that both Figure 1 and Table 1 have been updated in the revised paper.
>
> **(2) More Explanation regarding Definition 5.2**
>
> In Remark 5.1 of the revised paper, we provide more explanation regarding the degree of nonconvexity defined in Definition 5.2. In particular, the impact of a nonconvex function $h$ on the degree of nonconvexity is explained as follows. The degree of nonconvexity of a  second-order differentiable and nonconvex function $h$ satisfies $\theta_{h}(\mathbf t,\kappa) \le 0$ if $h$ is “more PSD”  than $-\kappa || \cdot ||^2$, that is, the smallest eigenvalue of its Hessian is not less than $-\kappa$. In other words, when the smallest eigenvalue of its Hessian of the nonconvex function $h$ is not less than $-\kappa$, then its degree of nonconvexity with $\kappa$, $\theta_{h}(\mathbf t,\kappa)$, is always not greater than $0$.
>
> Two commonly used nonconvex functions for sparse learning, the smoothly clipped absolute deviation (SCAD)  and the minimax concave penalty (MCP), are introduced in Section A of both the original paper and the revised paper. These two nonconvex functions satisfy Assumption 2, so there exists a positive concavity parameter $\zeta_{-} > 0$ such that the smallest eigenvalue of its Hessian is not less than $-\zeta_{-} > 0$. As a result, the degree of nonconvexity of both SCAD and MCP satisfies $\theta_{h}(\mathbf t, -\zeta_{-}) \le 0$.

---

> ### Author Response · Authors · 2024-11-29
> **Response to Reviewer X44j Part 2**
>
> **(3) Assumption of full column rank in Theorem 3.1**
>
> We would like to emphasize that Theorem 3.1 holds for either of the two cases: (1) the regularizer $h = h_{\lambda}$ is convex, or (2) $\mathbf X$ is of full rank. Throughout this paper, we only apply Theorem 3.1 for convex regularizer $h$, such as the proof of the minimax optimal rates by sketching for sparse convex learning in Theorem 4.1, and in such applications of  Theorem 3.1 we do not require $\mathbf X$ to have full rank. **In summary, all the theoretical and empirical results of this paper using Theorem 3.1 do not need $\mathbf X$ to have full rank**.
>
> On the other hand, Theorem 3.1 can also be applied to nonconvex regularizer $h$. In order to ensure that the Iterative ROS algorithm can also enjoy the exponential decay of the approximation error (In Eq. (8) of Theorem 3.1), we need a full rank $\mathbf X$. It is still open in the sketching literature that how iterative sketching can render the same guarantee for nonconvex functions as that for convex functions, and our Theorem 3.1 gives a partial solution to this open problem with the condition that $\mathbf X$ is of full rank.
>
> **(4) Numerical Results for Nonconvex Regularization**
>
> Thank you for your suggestion, and we present the empirical study of SRO for sparse nonconvex learning with capped-$\ell^1$ regularization in Section C.5 of the revised paper.
>
>
> **References**
>
> [YangLi2021] FROS: fast regularized optimization by sketching. In IEEE Interna tional Symposium on Information Theory, 2021.

---

> > ### Comment · Reviewer_X44j · 2024-12-02
> >
> > Actually,  the authors have addressed most of my concerns. As such, I would like to raise my score to 6.

---

### Official Review · Reviewer_2raj · 2024-11-12

**Soundness:** 3
**Presentation:** 2
**Contribution:** 3
**Rating:** 8
**Confidence:** 3

**Summary:**

The paper introduces an extension of the Iterative Hessian Sketch from Pilanci and Wainwright to the regularized setting. The authors first show convergence of the algorithm for sparse signal estimation (i.e. the LASSO) for both the traditional l1 norm as well as for a non convex penalty. In both cases, under various assumptions,  they are able to derive a minimax error  bound on the deviation between the N^th iterate of the single step algorithm (which they call SRO and which amounts to a minimization of the sketched formulation) and the groundtruth. They then extend their result to a general convex setting and show how to improve convergence by means of an iterative version of the SRO algorithm which is essentially an extension of the IHS algorithm from P and W to regularized formulations.

**Strengths:**

The paper is well written on the whole. The beginning of the paper remains very close to the paper “Iterative Hessian Sketch: Fast and Accurate Solution Approximation for Constrained Least-Squares” from Pilanci and Wainwright. In particular compare Equation (5) with Equation (25) in this last paper, or (24) with (7).  From what I understand, the main difference lies in the use of a regularizer. An incremental result is not necessarily a bad result but I’m still not fully decided about whether it is sufficiently original to be accepted.

**Weaknesses:**

The paper is not bad but it should clearly be reorganized. The distinction between the solution of the sketched program and the final iterate of the SRO/iterative SRO algorithms is not very clear. Does \tilde{\beta} represent a critical point of (2) or the N^th iterate of Iterative SRO ? This is especially unclear for section 4.2. where the meaning of \tilde{\beta} is not clarified in the statements of Corollary 4.2., and Theorem 4.5. but is mentioned as a critical point in the statement of Theorem 5.2.

Also your key contribution is the sketching step so the \tilde{n} should appear more clearly in all your statements (see my comments below)

**Questions:**

- line 30 “Sketching algorithms has been used to approximately” —>  “Sketching algorithms have been used to approximately”
- line 62, Equation (30) the lines 129-131 should appear way earlier. In particular, you want to better motivate the fact that the sketching is restricted to the quadratic term. I understand that restricting the sketching to the quadratic term might improve convergence but is that really necessary? If you still need to compute the product y^TX when do you the gradient descent, isn’t that equivalent to the X^TX*\beta ? I.e. in the latter you just have 2 more applications of a vector to the matrix X.
- line 68, you give the definition of the semi-norm ||u||_X twice
- line 116 I would use the cardinality of the support to denote the number of non zero elements of a matrix X instead of introducing the notation nnz. I.e. |supp(X)|. It is a detail though
- line 123: “It is comprised of ” —> “It consist of”
- lines 125-127 are quite obvious to me. I think you could just remove them
- line 136-140, in the Definition 2.1. is there any reason you use \ell^2 instead of \ell_2 ?
- line 142, Definition 2.2. looks more like a proposition (Lemma  perhaps) to me.
- line 151 : “Taking derivative with respect to” —> “taking the derivative with respect to”?
- line 189-190 “is a more accuracy” —> “is a more accurate”
- line 170 “consectively” —> “consecutively” or “iteratively/alternatively”?
- The structure of the proof sketch for Theorem 3.1. is good but your exposition of formulations (5) (6) and (7) is unclear. You should either keep the formulation from (6) to (7) (aside from the sketching) or provide at least a to (short) sentence indicating that (7) is obtained by moving the \beta^{(t-1)} out of the quadratic term to the linear term.
- Lines 295-296, Statement of Assumption 3, is the concavity parameter \zeta_- that you introduce in Assumption 3 the same as the zeta_- that you use in Assumption 2? if so it should be clarified in Assumption 2 (i.e. you could say something like “where zeta_- is known as the concavity parameter” or slightly modify Assumption 3 by saying “The concavity parameter \zeta_- introduced in Assumption 2…”)
- In the statements of Theorem 4.1, Corollary 4.2, Theorem 4.5 (this is especially true for Theorem 4.5), I would remove the explicit constants. The relation between the values you get for \tilde{n} and the constants is not clear anyways so those should go (keep the \varepsilon/1-\varepsilon for Corollary 4.2). If you really want you can add a one line sentence for Corollary 4.2. indicating how they depend on \rho_{\mathcal{L}, +} and \rho_{\mathcal{L}, -}.
- I recommend that you change the statement of Theorems 4.1, 4.5 and 5.2. so that they look more like the statement of Corollary 4.2. in terms of how you introduce the \tilde{n}. I.e. the lower bounds on \tilde{n} should not be introduced as a side comment. \tilde{n} should appear before you introduce \tilde{\beta}. E.g. Let \tilde{\beta} denote the solution to (5) with a sketching operator of size \tilde{n}\times n (or even better “for the sketching parameter \tilde{n}…”)
- I’m not sure Figure 1 is really meaningful. It only indicates a constant reduction in the relative error. What is more interesting is a bound such as (8)
- Lines 403-407, Definition 5.2. From what I understand, the degree of non convexity for a particular value of kappa gives you a measure of whether the function is more convex than the function \kappa \|x\|^2 ? I.e \theta_h will give you a negative value if the Hessian of the function is more psd than -\kappa I and positive otherwise? Remark 5.1. could be clearer
- Lines 416-420, in the statement of Theorem 5.2., what do you mean by an “oblivious” \ell^2 subspace embedding? Are you referring to an embedding that satisfies Definition 2.1. ? Then why not just say it like that? If you use space to introduce Definition 2.1, leverage it.
- line 424 - 426 : “by arbitrary critical point ” —> “by any arbitrary critical point”
- lines 428-431, from what I understand \theta_{h_\lambda} is lower bounded by \kappa? it might be good to add even a short clarifying sentence
- The sentence on line 430 does not make sense. Do you mean that “if the if the subdifferential is Lipschitz or if \kappa can be set to 0 then you get an admissible error bound?” If so, this is not what Theorem 5.4. states.
- Statement of Theorem 5.4., I would remove the sentence following Equation (17) as it refers to a result that does not appear?
- I think I would reorganize sections 5.1. and 5.2 . At the end of the day, the two Theorems 5.2. and 5.4 are really steps towards the proof of Theorem 3.1. I would add a one paragraph below this last theorem explaining how you use the degree of non convexity to prove it. This would make it possible to remove the statements of Theorems 5.1 and 5.2
- general question: wouldn’t it be more efficient to draw a new sketch at each iteration in the iterative SRO algorithm ?

---

> ### Author Response · Authors · 2024-11-30
> **Response to  Reviewer 2raj  Part 1**
>
> We appreciate the review and the suggestions in this review. The raised issues are addressed below. In the following text, the line numbers are for the revised paper without special notes.
>
> **(1) Novelty of Our Results and Their Significant Difference from [Pilanci2016]**
>
> **We respectfully point out that the claim “An incremental result is not necessarily a bad result…” in this review is a factual misunderstanding**.  Our results are novel and significantly different from those in [Pilanci2016], which is detailed in line 437-459 of the revied paper and copied below for your convenience.
> It is remarked that [Pilanci2016] only handles convex constrained least square problems of the form $\min _ {\mathbf X \in \mathcal C} || \mathbf X \mathbf \beta-\mathbf y ||^2$ where the constraint set $\mathcal C$ is a convex set, while our results cover regularized convex and nonconvex problems with minimax optimal rates. It is emphasized that the techniques in [Pilanci2016] can never be applied to the regularized problems considered in this paper. [Pilanci2016] heavily relies on certain complexity measure of the constraint set $\mathcal C$, such as the Gaussian width. It shows that the complexity of such constraint set $\mathcal C$ is bounded, so that sketching with such constraint set $\mathcal C$ of limited complexity only incurs a relatively small approximation error. However, there is never such constraint set in the original problem (Eq. (1)) or the sketched problem (Eq. (2)), so that such complexity based analysis for sketching cannot be applied to this work. Furthermore, as mentioned in Section 1.1, Iterative SRO does not need to sample the projection matrix and compute the sketched matrix at each iteration, in contrast with IHS [Pilanci2016] where a separate projection matrix is sampled and the sketched matrix is computed at each iteration. As evidenced by Table 1 in Section 7.1, Iterative SRO is more efficient than its “HIS” counterpart where sketching is performed at every iteration while enjoying comparable approximation error.
>
> **(2) Improved Presentation**
>
> The meaning of …$\tilde {\mathbf \beta}^*$ has been clearly indicated in every theoretical result in the revised paper. In particular, when using Iterative SRO for sparse convex learning in Section 4.1, it is clearly stated in Theorem 4.1 that $\tilde {\mathbf \beta}^* = \mathbf \beta^{(N)}$ (line 262-263 of the revised paper or line 254-255 of the original submission). When using SRO for sparse nonconvex learning in Section 4.2, it is clearly stated in Theorem 4.2 that $\tilde {\mathbf \beta}^*$ is the optimization result of the sketched problem (Eq. (2)) in line 368-369 of the revised paper.  To make the meaning of $\tilde {\mathbf \beta}^*$ even clearer, it is stated in line 237-238 of the revised paper that “In Section 4.1, $\tilde {\mathbf \beta}^*$ is obtained by Algorithm 1 through $\tilde {\mathbf \beta}^* = \mathbf \beta^{(N)}$. In Section 4.2, $\tilde {\mathbf \beta}^*$ is the optimization result of the sketched problem (2)” (Section 4.1 in line 238 should be Section 4.2).
>
> We have improved the organization of the theoretical results, and Theorem 5.2 and Theorem 5.4 (which now become Theorem D.2 and Theorem D.3 of the revised paper), as the intermediate result for the proof of Theorem 3.1, has been moved to Section D of the revised paper, following the suggestion of this review for the improved clarity of this paper. We have also fixed the presentation issues or typos mentioned in all the questions in this review.
>
> **”…wouldn’t it be more efficient to draw a new sketch at each iteration in the iterative SRO algorithm ?”**
>
> As explained in line 443-447 of the revised paper, if we draw a new sketch at each iteration of Iterative SRO, then at each iteration of Iterative SRO a new projection matrix $\mathbf P \in {\mathbb R}^{\tilde n \times n}$ is sampled and a new sketched matrix $\tilde {\mathbf X} = \mathbf P \mathbf X$ is computed, which incurs considerably more computational cost for large-scale problem with large data size $n$ compared to the proposed Iterative SRO described in Algorithm 1 where only a single projection matrix is sampled and a single sketch matrix is computed throughout all the iterations of the Iterative SRO.
>
> **References**
>
> [Pilanci2016] Pilanci et al. Iterative hessian sketch: Fast and accurate solution approximation for constrained least-squares. Journal of Machine Learning Research, 2016.

---

### Author Response · Authors · 2024-12-04
**Explanation and remedies for the unintended, coincidental and the very limited text overlap with [YangLi2021] (only in the introduction to the basic background)**

Dear AC and Reviewers,

Thank you for your time reviewing and handling this paper. All the technical concerns about this paper have been solved in the rebuttal phase.

We would like to let you know that there is truly unintended, coincidental, and very limited text overlap with [YangLi2021] when we present an introduction to the basic background about sketching for convex and nonconvex optimization problems in this paper. We thank reviewer Y2Sn who mentioned such similarities to [YangLi2021]. We would like to emphasize again that such coincidental text overlap is limited to basic introduction to the general background in sketching for convex and nonconvex optimization (in the abstract and the beginning part of the introduction section), and it never affects the novelty of our results and their significant differences from [YangLi2021]. In fact, reviewer X44j  mentioned that the issue about the difference from [YangLi2021] has been solved. Importantly, reviewer Y2Sn mentioned that "I completely agree that there is a lot of theory contained in this paper which is not present in [YangLi2021]. As I have stated, I think that the difference between the papers points to enough novelty!" and "I reiterate that my score is a 6, and I hence think that the quality of the paper is good enough to be accepted."


**Remedies.** **As a fundamentally important remedy,  [YangLi2021] has already been cited sufficiently in the discussion in Section 5 of the revised paper where the differences of our results from [YangLi2021] are sufficiently discussed and acknowledged by reviewer X44j and reviewer Y2Sn, and also in the new Introduction attached below**. Moreover, we have presented a new revised version for both the Abstract and the Introduction section (attached below), and now there is no overlap with [YangLi2021]  in the Abstract and the Introduction section except for commonly used standard mathematical description in this literature. We will update the Abstract and the Introduction section of the final paper accordingly using their revised version.


**New Abstract (to replace the current Abstract)**

Randomized algorithms play a crucial role in efficiently solving large-scale optimization problems. In this paper, we introduce Sketching for Regularized Optimization (SRO), a fast sketching algorithm designed for least squares problems with convex or nonconvex regularization. SRO operates by first creating a sketch of the original data matrix and then solving the sketched problem. We establish minimax optimal rates for sparse signal estimation by addressing the sketched sparse convex and nonconvex learning problems. Furthermore, we propose a novel Iterative SRO algorithm, which significantly reduces the approximation error geometrically for sketched convex regularized problems. To the best of our knowledge, this work is among the first to provide a unified theoretical framework demonstrating minimax rates for convex and nonconvex sparse learning problems via sketching. Experimental results validate the efficiency and effectiveness of both the SRO and Iterative SRO algorithms.

**New Introduction (to replace the corresponding parts in the current Introduction Section)**

Randomized algorithms for efficient optimization are a critical area of research in machine learning and optimization, with wide-ranging applications in numerical linear algebra, data analysis, and scientific computing. Among these, matrix sketching and random projection techniques have gained significant attention for solving sketched problems at a much smaller scale (Vempala, 2004; Bout- sidis & Drineas, 2009; Drineas et al., 2011; Mahoney, 2011; Kane & Nelson, 2014). These methods have been successfully applied to large-scale problems such as least squares regression, robust regression, low-rank approximation, singular value decomposition, and matrix factorization (Halko et al., 2011; Lu et al., 2013; Alaoui & Mahoney, 2015; Raskutti & Mahoney, 2016; Yang et al., 2015; Drineas & Mahoney, 2016; Oymak et al., 2018; Oymak & Tropp, 2017; Tropp et al., 2017). Regularized optimization problems with convex or nonconvex regularization, such as the widely used in regularized least squares such as Lasso and ridge regression, play a fundamental role in machine learning and statistics. While prior research has extensively explored random projection and sketching methods for problems with standard convex regularization (Zhang et al., 2016b) or convex constraints (Pilanci & Wainwright, 2016), there has been limited focus on analyzing regularized problems with general convex or nonconvex regularization frameworks.

We would like to emphasize that while [YangLi2021] also studies sketching for regularized optimization problem, the focus and results of this work are completely different from that in [YangLi2021], with a detailed discussion deferred to Section 5. (to be continued)

---

> ### Author Response · Authors · 2024-12-04
> **The New Introduction (cont'd)**
>
> (cont'd) In particular, due to our novel result in the approximation error bound (Theorem D.2-Theorem D.3), the proposed iterative sketching algorithm, Iterative SRO, does not need to sample a new projection matrix and compute the sketched matrix at every iteration, in a strong contrast to [YangLi2021]. Moreover, the focus of this work is to establish minimax optimal rates for sparse convex and nonconvex learning problems by sketching, which has not been addressed by existing works in the literature including [YangLi2021]. While [YangLi2021] only focuses on the optimization perspective, that is, approximating the solution to the original optimization problem by the solution to the sketched problem, the focus of this work needs much more efforts beyond the efforts made in [YangLi2021] for optimization only: we need to show that the solution to the sketched problem can still enjoy the minimax optimal rates for estimation of the sparse parameter vector for both sparse convex and nonconvex learning problems. Such efforts and results in minimax optimal rates for sparse convex and nonconvex learning problems by sketching have not been offered by previous works including [YangLi2021], which are provided in Section 4 of this paper. Such minimax optimal results are established in a highly non-trivial manner. For example, to the best of our knowledge, Theorem 4.1 is among the first in the literature which uses an iteratively sketching algorithm to achieve the minimax optimal rate for sparse convex learning. Furthermore, Theorem 4.5 shows that sketching can also lead to the minimax optimal rate even for sparse nonconvex problems, while sketching for nonconvex problems is still considered difficult and open in the literature. (end of the new Introduction)
>
> Finally, we will be happy to work with the ethical reviewer and the AC to solve any remaining text overlap issues (if there are any).
>
> Best Regards,
>
> The Authors

---

### Meta-Review · Area_Chair_p96u · 2024-12-18

**Metareview:**

Dear Authors,

Thank you for your valuable contribution to the ICLR and the ML community. Your submitted paper has undergone a rigorous review process, and I have carefully read and considered the feedback provided by the reviewers.

This paper introduces an extension of the Iterative Hessian Sketch to the regularized setting. The methods Sketching for Regularized Optimization (SRO) and its iterative version can handle convex and non-convex regularizers. A theoretical framework is also presented to establish minimax rates. Overall, the paper received mostly positive response from the reviewers (8,6,6,6) scores.

Given this positive assessment, I am willing to recommend the acceptance of your paper for publication.

I would like to remind you to carefully review the reviewer feedback and the resulting discussion. While most reviews were positive, the reviewers have offered valuable suggestions that can further strengthen the quality of the paper. Please take another careful look a the 'weaknesses' section of each reviewer comment. I encourage you to use this feedback to make any necessary improvements and refinements before submitting the final version of your paper.

Once again, thank you for submitting your work to ICLR.

Best,
Area Chair

**Additional Comments On Reviewer Discussion:**

Reviewers pointed out issues in presentation and writing. They also questioned the improvements with respect to the Iterative Hessian Sketch work. The rebuttal helped clarify these issues and led to uniformly positive scores.

---

### Decision · Program_Chairs · 2025-01-22

Accept (Poster)